# Low Rank Matrix Completion via Robust Alternating Minimization in Nearly Linear Time

**Yuzhou Gu**
Institute of Advanced Study
yuzhougu@ias.edu

**Zhao Song**
Adobe Research
zsong@adobe.com

**Junze Yin**
Boston University
junze@bu.edu

**Lichen Zhang**
MIT CSAIL
lichenz@csail.mit.edu

## Abstract

Given a matrix $M \in \mathbb{R}^{m \times n}$, the low rank matrix completion problem asks us to find a rank-$k$ approximation of $M$ as $UV^\top$ for $U \in \mathbb{R}^{m \times k}$ and $V \in \mathbb{R}^{n \times k}$ by only observing a few entries specified by a set of entries $\Omega \subseteq [m] \times [n]$. In particular, we examine an approach that is widely used in practice — the alternating minimization framework. Jain, Netrapalli, and Sanghavi Jain et al. (2013) showed that if $M$ has incoherent rows and columns, then alternating minimization provably recovers the matrix $M$ by observing a nearly linear in $n$ number of entries. While the sample complexity has been subsequently improved Gamarnik et al. (2017), alternating minimization steps are required to be computed exactly. This hinders the development of more efficient algorithms and fails to depict the practical implementation of alternating minimization, where the updates are usually performed approximately in favor of efficiency.

In this paper, we take a major step towards a more efficient and error-robust alternating minimization framework. To this end, we develop an analytical framework for alternating minimization that can tolerate a moderate amount of errors caused by approximate updates. Moreover, our algorithm runs in time $\widetilde{O}(|\Omega|k)$, which is nearly linear in the time to verify the solution while preserving the sample complexity. This improves upon all prior known alternating minimization approaches which require $\widetilde{O}(|\Omega|k^2)$ time.

## 1 Introduction

Matrix completion is a well-studied problem both in theory and practice of computer science and machine learning. Given a matrix $M \in \mathbb{R}^{m \times n}$, the matrix completion problem asks to recover the matrix by observing only a few (random) entries of $M$. This problem originally appears in the context of collaborative filtering Rennie & Srebro (2005) with the most notable example being the Netflix Challenge. Since then, it has found various applications in signal processing Linial et al. (1994); So & Ye (2005) and traffic engineering Gürsun & Crovella (2012). In theory, one requires additional structural assumptions on the matrix $M$ in order to obtain provable guarantees, with the most natural and practical assumption being the matrix $M$ is a rank-$k$ low rank matrix. In tasks such as collaborative filtering, the matrix one faces is often low rank Candès & Recht (2012). Another popular assumption is the $M$ has incoherent rows and columns. This intuitively eliminates the degenerate case where $M$ only has a few large entries, making it imperative to observe them. Matrices in practice are usually incoherent as well Avron et al. (2010); Mohri & Talwalkar (2011); Kumar et al. (2012).

Under these assumptions, a variety of algorithms based on convex relaxation have been derived Candès & Tao (2010); Recht (2011); Candès & Recht (2012). These algorithms relax the problem as a trace-norm minimization problem that can be solved with semidefinite programs (SDP). Unfortunately, solving SDP is inherently slow and rarely used in practice: even the state-of-the-art SDP solver would

require $O(n^\omega)$ time[1] to solve the program Jiang et al. (2020); Huang et al. (2022). Heuristics such as alternating minimization and gradient descent are much preferred due to their efficiency. Jain et al. (2013) first showed that under the standard low rank and incoherent settings, as long as we are allowed to observe $\widetilde{O}(\kappa^4 n k^{4.5} \log(1/\epsilon))$ entries[2] (where $\kappa$ is the condition number of matrix $M$), then alternating minimization provably recovers the matrix $M$ up to $\epsilon$ error in terms of Frobenius norm. Subsequent works further unify different approaches by treating matrix completion as a non-convex optimization problem Zhao et al. (2015) and improve the sample complexity Gamarnik et al. (2017). A remarkable advantage of alternating minimization is its efficiency, as each iteration, the two alternating updates can be implemented in $O(|\Omega|k^2)$ time.

Despite various advantages, the series of theoretical papers analyzing alternating minimization fail to capture a crucial component of many practical implementations as the updates are usually computed approximately to further speed up the process. In contrast, the analysis pioneered in Jain et al. (2013) and followups Zhao et al. (2015); Gamarnik et al. (2017); Li et al. (2016) crucially relies on the exact formulation of the updates, making it difficult to adapt and generalize to the approximate updates setting. On the other hand, developing an alternating minimization framework would also enable us to utilize faster, approximate solvers to implement updates that leads to *theoretical speedup* of the algorithm.

In this paper, we take the first major step towards an error-robust[3] analysis for alternating minimization. Specifically, we show that the alternating updates can be formulated as two multiple response regressions and fast, high accuracy solvers can be utilized to solve them in nearly linear time. Coupling our robust alternating minimization framework with our faster multiple response regression solver, we derive an algorithm that solves the matrix completion problem in time $\widetilde{O}(|\Omega|k)$. We note that this runtime is nearly linear in terms of the time to *verify the solution*: Given matrices $U \in \mathbb{R}^{m \times k}$ and $V \in \mathbb{R}^{n \times k}$, it takes $O(k)$ time to verify a single entry and a total of $O(|\Omega|k)$ time to verify all entries in $\Omega$. To the best of our knowledge, this is *the first algorithm based on alternating minimization, that achieves a nearly linear time in verification.* We remark that a recent work Kelner et al. (2023b) achieves a similar runtime behavior of $\widetilde{O}(|\Omega|k)$: their algorithm has an improved sample complexity of $|\Omega| = \widetilde{O}(nk^{2+o(1)})$ and runtime of $\widetilde{O}(nk^{3+o(1)})$. However, their algorithm heavily relies on the tools developed in Kelner et al. (2023a) and mainly serves as a proof-of-concept for matrix completion rather than efficient practical implementation.[4] In contrast, our algorithm adopts an off the shelf regression solver that is fast and high accuracy with good practical performances Avron et al. (2010); Meng et al. (2014).

We state an informal version of our main result as follows:

**Theorem 1.1** (Informal version of Theorem 4.2)**.** *Let $M \in \mathbb{R}^{m \times n}$ be a matrix that is rank-$k$, has incoherent rows and columns and entries can be sampled independently. Then, there exists a randomzied algorithm that samples $|\Omega| = \widetilde{O}(n \operatorname{poly}(k))$ entries, and with high probability, outputs a pair of matrices $\widehat{U} \in \mathbb{R}^{m \times k}, \widehat{V} \in \mathbb{R}^{n \times k}$ such that*

$$\|M - \widehat{U}\widehat{V}^\top\|_F \le \epsilon,$$

*and the algorithm runs in time $\widetilde{O}(|\Omega|k)$.*

**Roadmap.** In Section 2, we introduce the related works which include matrix completion and applying sketching matrices to solve optimization problems. In Section 3, we present the notations, definitions, and lemmas that we use for the later sections. In Section 4, we present the significant findings and discuss the techniques utilized. In Section 5, we provide some concluding remarks and future directions.

---

[1] $\omega$ is the exponent of matrix multiplication. Currently, $\omega \approx 2.37$ Duan et al. (2023).

[2] In this paper, we assume $n \ge m$ and we use $\widetilde{O}(\cdot)$ to hide polylogarithmic factors in $n$, $k$ and $1/\epsilon$.

[3] We note that in the context of matrix completion, robustness is often used with respect to noise — i.e., we observe a noisy, higher rank matrix $M = M^* + N$, where $M^*$ is the low rank ground truth and $N$ is the higher rank noise matrix. In this paper, we focus on the setting where the ground truth is noiseless, and we use robust referring to robustness against the errors caused by computing updates approximately.

[4] We note another work that also claims to obtain a runtime of $\widetilde{O}(|\Omega|k)$ Cherapanamjeri et al. (2017), but as pointed out in Kelner et al. (2023b) their runtime is only achievable if certain linear algebraic operations can be performed exactly in sublinear time, which is unknown to this day.

## 2 RELATED WORK

Low rank matrix completion is a fundamental problem in machine learning. Practical applications including recommender systems, with the most notable one being collaborative filtering Rennie & Srebro (2005) and the Netflix Challenge Koren (2009); Koren et al. (2009). It also has a wide range of applications in computer visions Candès & Recht (2012) and signal processing Linial et al. (1994); So & Ye (2005); Candès et al. (2011). For more comprehensive surveys, we refer readers to Johnson (1990); Nguyen et al. (2019). Algorithms for matrix completion can be roughly divided into two categories — convex relaxation and non-convex heuristics. Candes and Recht Candès & Recht (2012) prove the first sample complexity for low rank matrix completion under convex relaxation and the bound is subsequently improved by Candès & Tao (2010). In practice, heuristic methods based on non-convex optimizations are often preferred due to their simplicity and efficiency. One notable approach is gradient descent Keshavan et al. (2009); Zhao et al. (2015). Alternating minimization is also a popular alternative that is widely applied in practice. Jain et al. (2013) provides a provable guarantee on the convergence of alternating minimization when the matrix-to-recover is low rank and incoherent. Subsequently, there have been a long line of works analyzing the performance of non-convex heuristics under standard matrix completion setting and under the noise or corrupted-entries setting Gunasekar et al. (2013); Hardt (2014); Hardt et al. (2014); Hardt & Wootters (2014); Jain & Netrapalli (2015); Sun & Luo (2016); Gamarnik et al. (2017); Cherapanamjeri et al. (2017). Recently, the work Kelner et al. (2023b) provides an algorithm with improved sample complexity and runtime when stronger subspace regularity assumptions are imposed.

In addition to the high accuracy randomized solver we utilize in this paper, sketching has a variety of applications in numerical linear algebra and machine learning, such as linear regression, low rank approximation Clarkson & Woodruff (2013); Nelson & Nguyên (2013), matrix CUR decomposition Boutsidis & Woodruff (2014); Song et al. (2017; 2019c), weighted low rank approximation Razenshteyn et al. (2016), entrywise $\ell_1$ norm low-rank approximation Song et al. (2017; 2019b), general norm column subset selection Song et al. (2019a), tensor low-rank approximation Song et al. (2019c), and tensor regression Diao et al. (2018; 2019); Reddy et al. (2022); Song et al. (2021).

A recent trend in the sketching community is to apply them as a means to reduce the iteration cost of optimization algorithms. Applications such as linear programming Song & Yu (2021), empirical risk minimization Lee et al. (2019); Qin et al. (2023), approximating the John ellipsoid Cohen et al. (2019), Frank-Wolfe algorithm Xu et al. (2021) and linear MDPs Xu et al. (2023).

## 3 PRELIMINARY

In this section, we provide necessary background on matrix completion and assumptions.

**Notations.** We use $[n]$ for a positive integer $n$ to denote the set $\{1, 2, \ldots, n\}$. Given a vector $x$, we use $\|x\|_2$ to denote its $\ell_2$ norm. Given a matrix $A$, we use $\|A\|$ to denote its spectral norm and $\|A\|_F$ to denote its Frobenious norm. Given a matrix $A$, we use $A_{i,*}$ to denote its $i$-th row and $A_{*,j}$ to denote its $j$-th column. We use $\|A\|_0$ to denote the $\ell_0$ semi-norm of $A$ that measures the number of nonzero entries in $A$.

Given an orthonormal basis $A \in \mathbb{R}^{n \times k}$, we use $A_\perp \in \mathbb{R}^{n \times (n-k)}$ to denote an orthonormal basis to $A$'s orthogonal complement, such that $AA^\top + A_\perp A_\perp^\top = I_n$.

Given a rank-$k$, $m \times n$ real matrix $A$, we use $\kappa(A)$ to denote its condition number: $\kappa(A) = \frac{\sigma_1(A)}{\sigma_k(A)}$ where $\sigma_1(A), \ldots, \sigma_k(A)$ are singular values of $A$ sorted in magnitude. When $A$ is clear from context, we often use $\kappa$ directly.

### 3.1 ANGLES AND DISTANCES BETWEEN SUBSPACES

A key quantity that captures the closeness of the subspaces by two matrices is the distance, defined as follows:

**Definition 3.1** (Distance between general matrices). *Given two matrices $\widehat{U}, \widehat{W} \in \mathbb{R}^{m \times k}$, the (principal angle) distance between the subspaces spanned by the columns of $\widehat{U}$ and $\widehat{W}$ is given by:*

$$\text{dist}(\widehat{U}, \widehat{W}) := \|U_\perp^\top W\| = \|W_\perp^\top U\|$$

*where $U$ and $W$ are orthonormal bases of the spaces $\text{span}(\widehat{U})$ and $\text{span}(\widehat{W})$, respectively. Similarly, $U_\perp$ and $W_\perp$ are any orthonormal bases of the orthogonal spaces $\text{span}(U)^\perp$ and $\text{span}(W)^\perp$, respectively.*

One can further quantify the geometry of two subspaces by their principal angles:

**Definition 3.2.** *Let $U \in \mathbb{R}^{n \times k}$ and $V \in \mathbb{R}^{n \times k}$.*

*For any matrix $U$, and for orthogonal matrix $V$ ($V^\top V = I_k$) we define*

- $\tan \theta(V, U) := \|V_\perp^\top U(V^\top U)^{-1}\|.$

*For orthogonal matrices $V$ and $U$ ($V^\top V = I_k$ and $U^\top U = I_k$), we define*

- $\cos \theta(V, U) := \sigma_{\min}(V^\top U).$
  - $\cos \theta(V, U) = 1/\|(V^\top U)^{-1}\|$ *and* $\cos \theta(V, U) \leq 1.$
- $\sin \theta(V, U) := \|(I - VV^\top)U\|.$
  - $\sin \theta(V, U) = \|V_\perp V_\perp^\top U\| = \|V_\perp^\top U\|$ *and* $\sin \theta(V, U) \leq 1.$

Standard trigonometry equalities and inequalities hold for above definitions, see, e.g., Zhu & Knyazev (2013).

## 3.2 BACKGROUND ON MATRIX COMPLETION

We introduce necessary definitions and assumptions for low rank matrix completion. Given a set of indices $\Omega$, we define a linear operator that selects the corresponding entries:

**Definition 3.3.** *Let $S \in \mathbb{R}^{m \times n}$ denote a matrix. Let $\Omega \subset [m] \times [n]$. We define operator $P_\Omega$ as follows:*

$$P_\Omega(S)_{i,j} = \begin{cases} S_{i,j}, & \text{if } (i,j) \in \Omega; \\ 0, & \text{otherwise.} \end{cases}$$

A crucial assumption for low rank matrix completion is incoherence. We start by defining the notion below.

**Definition 3.4.** *For an orthonormal basis $U \in \mathbb{R}^{m \times k}$, we say $U$ is $\mu$-incoherent, if*

$$\|u_i\|_2 \leq \frac{\sqrt{k}}{\sqrt{m}} \cdot \mu.$$

*Here $u_i$ is the $i$-th row of $U$, for all $i \in [m]$.*

We say a general rank-$k$ matrix is incoherent if both its row and column spans are incoherent:

**Definition 3.5** (($\mu, k$)-incoherent). *A rank-k matrix $M \in \mathbb{R}^{m \times n}$ is ($\mu, k$)-incoherent if:*

- $\|u_i\|_2 \leq \frac{\sqrt{k}}{\sqrt{m}} \cdot \mu, \forall i \in [m]$

- $\|v_j\|_2 \leq \frac{\sqrt{k}}{\sqrt{n}} \cdot \mu, \forall j \in [n]$

*where $M = U\Sigma V^\top$ is the SVD of M and $u_i$, $v_j$ denote the $i$-th row of $U$ and the $j$-th row of $V$ respectively.*

We are now in the position to state the set of assumptions for low rank matrix completion problem.

**Assumption 3.6.** *Let $M \in \mathbb{R}^{m \times n}$ and we assume $M$ satisfies the following property:*

- *$M$ is a rank-$k$ matrix;*

- *$M$ is $(\mu, k)$-incoherent;*

- *Each entry of $M$ is sampled independently with equal probability. In other words, the set $\Omega \subseteq [m] \times [n]$ is formed by sampling each entry independently according to a certain distribution.*

We note that the above assumptions are all standard in the literature.

Now we are ready to state the low rank matrix completion problem.

**Problem 3.7.** *Let $M \in \mathbb{R}^{m \times n}$ satisfy Assumption 3.6. The low rank matrix completion problem asks to find a pair of matrices $U \in \mathbb{R}^{m \times k}$ and $V \in \mathbb{R}^{n \times k}$ such that*

$$\|M - UV^\top\|_F \le \epsilon.$$

## 4    TECHNIQUE OVERVIEW

Before diving into the details of our algorithm and analysis, let us review the alternating minimization proposed in Jain et al. (2013). The algorithm can be described pretty succinctly: given the sampled indices $\Omega$, the algorithm starts by partitioning $\Omega$ into $2T + 1$ groups, denoted by $\Omega_0, \ldots, \Omega_{2T}$. The algorithm first computes a top-$k$ SVD of the matrix $\frac{1}{p} P_{\Omega_0}(M)$ where $p$ is the sampling probability. It then proceeds to trim all rows of the left singular matrix $U$ with large row norms (this step is often referred to as clipping). It then optimizes the factors $U$ and $V$ alternatively. At iteration $t$, the algorithm first fixes $U$ and solves for $V$ with a multiple response regression using entries in $\Omega_{t+1}$, then it fixes the newly obtained $V$ and solves for $U$ with entries in $\Omega_{T+t+1}$. After iterating over all groups in the partition, the algorithm outputs the final factors $U$ and $V$ as its output. Here, we use independent samples across different iterations to ensure that each iteration is independent of priors (in terms of randomness used). If one would like to drop the uniform and independent samples across iterations (see, e.g., Liu et al. (2017)), the convergence has only been shown under additional assumptions and in terms of critical points to certain non-convex program.

From a runtime complexity perspective, the most expensive steps are solving two multiple response regressions per iteration, which would take a total of $O(|\Omega|k^2)$ time. On the other hand, if the sum of the size of all the regressions across $T$ iterations is only $O(|\Omega|k)$, inspiring us to consider efficient, randomized solvers that can solve the regression in time nearly linear in the instance size.[5] This in turn produces an algorithm with a nearly linear runtime in terms of verifying all entries in $\Omega$.

Two problems remain to complete our algorithm: 1). What kind of multiple response regression solvers will run in nearly linear time and produce high accuracy solutions and 2). How to prove the convergence of the alternating minimization in the presence of the errors caused by approximate solvers. While the second question seems rather intuitive that the Jain et al. (2013) algorithm should tolerate a moderate amount of errors, the analysis becomes highly nontrivial when one examines the argument in Jain et al. (2013) and followups, as all of them crucially rely on the fact that the factors $U$ and $V$ are solved exactly and hence admit closed form. This motivates us to develop an analytical framework for alternating minimization, that admits approximate solutions for each iteration.

Our argument is inductive in nature. Let $M = U^*(V^*)^\top$ be optimal rank-$k$ factors, we build up our induction by assuming the approximate regression solutions $\widehat{U}_{t-1}, \widehat{V}_{t-1}$ are close to $U^*, V^*$ and proving for $\widehat{U}_t, \widehat{V}_t$. The loss of structure due to solving the regressions approximately forces us to find alternatives to bound the closeness, as the argument of Jain et al. (2013) relies on the exact formulation of the closed-form solution. Our strategy is to instead consider a sequence of imaginary, exact solutions to the multiple response regressions and prove that our approximate

---

[5]We note that our solver would incur an extra total cost of $\widetilde{O}(nk^3)$. Under the standard low rank and incoherent assumptions, the state-of-the-art non-convex optimization-based approach Kelner et al. (2023b) would require $\widetilde{O}(nk^{2+o(1)})$ samples and the $\widetilde{O}(nk^3)$ time is subsumed by the $\widetilde{O}(|\Omega|k)$ portion. This term is hence ignored.

solutions are close to those exact solutions. In particular, we show that if $\widehat{U}_t, \widehat{V}_t$ are close to their exact counterpart in the $\ell_2$ row norm sense, then it suffices to prove additional structural conditions on $U_t$ and $V_t$ to progress the induction. Specifically, we show that as long as $U_{t-1}$ is incoherent and $\text{dist}(\widehat{U}_{t-1}, U^*) \leq \frac{1}{4} \text{dist}(\widehat{V}_{t-1}, V^*)$, we will have $\text{dist}(\widehat{V}_t, V^*) \leq \frac{1}{4} \text{dist}(\widehat{U}_{t-1}, U^*)$ and the *exact solution at time* $t$, $\widehat{V}_t$ is incoherent. We can then progress the induction by using the small distance between $\widehat{V}_t$ and $V^*$ in conjunction with the incoherence of $V_t$ to advance on $\text{dist}(\widehat{U}_t, U^*)$ and the incoherence of $U_t$. As the gap decreases geometrically, this leads to an overall $\log(1/\epsilon)$ iterations to converge, as desired.

It remains to develop a perturbation theory for alternating minimization and our key innovation is a perturbation argument on the incoherence bound under small row norm perturbation. Let $A_i$ denote the $i$-th row of $A$. We show that if two matrices $A$ and $B$ satisfying $\|(A - B)_i\|_2 \leq \epsilon$ and $\epsilon \leq \sigma_{\min}(A)$, then the change of incoherence between $A$ and $B$ is also very small. Our main strategy involves reducing incoherence to statistical leverage score, a critical numerical quantity that measures the importance of rows Spielman & Srivastava (2011). The problem then reduces to showing that if two matrices have similar rows, their leverage scores are also similar. By utilizing the leverage score formulation and a perturbation result from Wedin (1973), we prove that it is indeed the case they have similar leverage scores. This concludes the perturbation argument on matrix incoherence.

However, this alone is insufficient for our inductive argument to progress, as we must also quantify the proximity between the approximate and exact solutions. To accomplish this, we employ a novel approach based on the condition number of the matrices. Specifically, given the exact matrix $A$ and the approximate matrix $B$, we demonstrate that the distance between $A$ and $B$ is relatively small as long as $\kappa(B)$ is bounded by $\kappa(A)$. To obtain a good bound on $\kappa(B)$, we need to solve the approximate regression to high precision (inverse polynomial proportionally to $\kappa(A)$). We achieve this objective through a sketching-based preconditioner, which is the first problem we would like to address.

Sketching-based preconditioning is a widely used and lightweight approach to solving regression problems with high accuracy. The fundamental concept involves reducing the dimensionality of the regression problem by utilizing a sketching matrix and creating an approximate preconditioner based on the sketch. This preconditioner accelerates the convergence of an iterative solver, such as the conjugate gradient or Generalized Minimal RESidual method (GMRES), employed in the reduced system. With this preconditioner, we obtain an $O(\log(1/\epsilon))$ convergence for an $\epsilon$-approximate solution. Beyond its theoretical soundness and efficiency, these preconditioners also exhibit good performances when used in practice Avron et al. (2010); Meng et al. (2014).

An important feature of our robust alternating minimization framework coupled with the fast regression solvers is that we preserve the sample complexity of Jain et al. (2013): by picking the sampling probability $p = O(\kappa^4 \mu^2 k^{4.5} \log n \log(1/\epsilon)/m)$, we are required to sample $|\Omega| = O(\kappa^4 \mu^2 n k^{4.5} \log n \log(1/\epsilon))$ entries. By a clever thresholding approach, Gamarnik et al. (2017) further improves the sample complexity to $O(\kappa^2 \mu^2 n k^4 \log(1/\epsilon))$. As their algorithm is alternating minimization in nature, our robust framework and nearly linear time regression solvers can be adapted and hence obtain an improved sample complexity and runtime by a factor of $O(\kappa^2 k^{0.5} \log n)$.

In the rest of this section, we will review our approaches and clarify how we employ various strategies to achieve our goal.

## 4.1 SUBSPACE APPROACHING ARGUMENT

The basis of the analysis of both Jain et al. (2013) and ours is an inductive argument that shows the low rank factors obtained in each iteration approach the optimum as their distance shrinks by a constant factor and we call this *subspace approaching argument*, as we iteratively refine subspaces that approach the optimum. The major difference is that the induction of Jain et al. (2013) can be based on the exact solutions solely — they inductively prove the distances shrink by a constant factor, and the low rank factors are $\mu_2$-incoherent for $\mu_2$ to be specified. It is tempting to replace these exact solutions with the approximate solutions we obtain. Unfortunately, the Jain et al. (2013) heavily exploits the closed-form formulation of the solutions so that they can easily decompose the matrices into terms whose spectral norms can be bounded in a straightforward manner. Our approximate solutions on the other hand cannot be factored in such a fashion.

---

**Algorithm 1** Alternating minimization for matrix completion. The INIT procedure clips the rows with large norms, then performs a Gram-Schmidt process.

1: **procedure** FASTMATRIXCOMPLETION($\Omega \subset [m] \times [n]$, $P_\Omega(M)$)                      ▷ Theorem 4.2
2:                      ▷ Partition $\Omega$ into $2T + 1$ subsets $\Omega_0, \cdots, \Omega_{2T}$
3:           ▷ Each element of $\Omega$ belonging to one of the $\Omega_t$ with equal probability (sampling with replacement)
4:     $U_\phi = \text{SVD}(\frac{1}{p} P_{\Omega_0}(M), k)$
5:     $\widehat{U}_0 \leftarrow \text{INIT}(U_\phi)$                      ▷ Algorithm 4
6:     **for** $t = 0, \cdots, T - 1$ **do**
7:         Obtain $\widehat{V}_{t+1}$ with Lemma 4.1 by solving $\min_{V \in \mathbb{R}^{n \times k}} \|P_{\Omega_{t+1}}(\widehat{U}_t V^\top - M)\|_F^2$
8:         Obtain $\widehat{U}_{t+1}$ with Lemma 4.1 by solving $\min_{U \in \mathbb{R}^{m \times k}} \|P_{\Omega_{T+t+1}}(U \widehat{V}_{t+1}^\top - M)\|_F^2$
9:     **end for**
10:     **return** $\widehat{U}_T \widehat{V}_T^\top$
11: **end procedure**

---

To circumvent this issue, we instead keep the exact solutions $U_t, V_t$ as a sequence of references and inductively bound the incoherence of these exact solutions. More specifically, for any $t \in [T]$, we assume $U_t$ is $\mu_2$-incoherent and $\text{dist}(\widehat{U}_t, U^*) \leq \frac{1}{4} \text{dist}(\widehat{V}_t, V^*) \leq 1/10$, then we show that

- $\text{dist}(\widehat{V}_{t+1}, V^*) \leq \frac{1}{4} \text{dist}(\widehat{U}_t, U^*) \leq 1/10$ and
- $V_{t+1}$ is $\mu_2$-incoherent.

We can then proceed to show similar conditions hold for $\widehat{U}_{t+1}$ and $U_{t+1}$, therefore advance the induction. At the first glance, our induction conditions seem rather far-off from what we want — in order to prove $\widehat{V}_{t+1}$ and $V^*$ are close, one would need to show $\widehat{V}_{t+1}$ has small incoherence. We alternatively adapt a perturbation argument on incoherence, specifically, note that $\widehat{V}_{t+1}$ can be treated as blending in a small perturbation to $V_{t+1}$ in terms of row $\ell_2$ norms, if we can manage to prove under such perturbations, the incoherence of $\widehat{V}_{t+1}$ is still close to the incoherence of $V_{t+1}$, then our induction condition on the incoherence of $V_{t+1}$ effectively translates to the incoherence of $\widetilde{V}_{t+1}$. In next section, we illustrate how to develop such a perturbation theory.

### 4.2 A PERTURBATION THEORY FOR MATRIX INCOHERENCE

Our main goal is to understand how perturbations to rows of a matrix change the incoherence. Let us consider two matrices $A, B \in \mathbb{R}^{m \times k}$ that are close in the spectral norm: $\|A - B\| \leq \epsilon_0$ for some small error $\epsilon_0$, as the closeness in spectral norm implies *all* row norms of the difference matrix $A - B$ are small. Our next step will be putting the rows of $A$ and $B$ to isotropic position[6], by mapping $a_i \mapsto (A^\top A)^{-1/2} a_i$ and $b_i \mapsto (B^\top B)^{-1/2} b_i$. The main motivation is the equivalence between matrix incoherence and statistical leverage score Spielman & Srivastava (2011), which measures the $\ell_2$ importance of rows by putting the matrix into isotropic position. This provides us with an alternative way to measure the matrix incoherence directly instead of working with the singular value decomposition. It remains to bound the difference $|\|(A^\top A)^{-1/2} a_i\|_2 - \|(B^\top B)^{-1/2} b_i\|_2|$, without loss of generality, assume $\|(A^\top A)^{-1/2} a_i\|_2 \geq \|(B^\top B)^{-1/2} b_i\|_2$, then we can equivalently express it using difference of squares formula:

$$\|(A^\top A)^{-1/2} a_i\|_2 - \|(B^\top B)^{-1/2} b_i\|_2 = \frac{\|(A^\top A)^{-1/2} a_i\|_2^2 - \|(B^\top B)^{-1/2} b_i\|_2^2}{\|(A^\top A)^{-1/2} a_i\|_2 + \|(B^\top B)^{-1/2} b_i\|_2}$$

where the numerator is the difference between leverage scores, and we utilize tools that bound perturbation of pseudo-inverses Wedin (1973) to provide an upper bound. For the denominator, we can derive a lower bound rather straightforwardly.

Combining these bounds, we conclude that

$$|\|(A^\top A)^{-1/2} a_i\|_2 - \|(B^\top B)^{-1/2} b_i\|_2| \leq O(\epsilon_0 \, \text{poly}(\kappa(A))).$$

---

[6]We say a collection of vectors $\{v_1, \ldots, v_m\} \subseteq \mathbb{R}^k$ are in isotropic position if $\sum_{i=1}^m v_i v_i^\top = I_k$.

To put this bound into perspective, we will set $A$ to be the exact update matrix $U$ and $B$ to be the approximate update matrix $\widehat{U}$. In our induction framework, we further make sure that the condition number of $U$ is well-controlled, therefore we can safely suffer error up to $\mathrm{poly}(\kappa(U))$, as we can set our final precision to be inverse polynomially in $\kappa(U)$. Since our algorithm converges in $\log(1/\epsilon)$ iterations, and our regression solver has an error dependence $\log(1/\epsilon)$, this only blows up the runtime by a factor of $\log(\kappa/\epsilon)$, as desired.

### 4.3 NEARLY LINEAR TIME SOLVE VIA SKETCHING

Now that we have a framework that tolerates errors caused by approximate solves, we are ready to deploy regression solvers that can actually compute the factors in nearly linear time. A popular and standard approach is through the so-called sketch-and-solve paradigm — given an overconstrained regression problem $\min_x \|Ax - b\|_2$ with $A \in \mathbb{R}^{n \times k}$ and $n \gg k$, one samples a random sketching matrix $S \in \mathbb{R}^{m \times n}$ with $m \ll n$ and instead solves the sketched regression $\|SAx - Sb\|_2$. As the new regression problem has much fewer row count, solving it would take nearly linear time as long as $SA$ can be quickly applied. In addition to its simplicity, efficiency and good error guarantees, if one picks $S$ to be a dense subsampled randomized Hadamard transform (SRHT) Lu et al. (2013), then the regression *solution* is also close to the exact solution in an $\ell_\infty$ sense Price et al. (2017). While appealing, sketch-and-solve has the deficiency of *low accuracy* — if one wants a solution with regression cost at most $(1 + \epsilon)$ times the optimal cost, then the algorithm would take time proportional to $\epsilon^{-2}$. As we would set the $\epsilon$ to be polynomially small to incorporate the error blowups due to perturbations to incoherence and the error conversion from regression cost to solution, an inverse polynomial dependence on $\epsilon$ would significantly slow down the algorithm and ultimately lead to an even slower runtime than solving the regressions exactly.

We resort to a sketching-based approach that enjoys high accuracy as the algorithm converges in $\log(1/\epsilon)$ iterations and each iteration can be implemented in $\widetilde{O}(nk + k^3)$ time. The idea is to pick a sketching matrix that produces a subspace embedding of $O(1)$-distortion[7] and compute a QR decomposition of matrix $SA = QR^{-1}$. The matrix $R$ is then used as a preconditioner and we solve the preconditioned regression $\min_x \|ARx - b\|_2$ via gradient descent.

Even though the solver runs in nearly linear time in the instance size, we still need to make sure that the *sparsity* of $\Omega$ is utilized, as $|\Omega| \ll mn$. To demonstrate how to achieve a runtime of $\widetilde{O}(|\Omega|k)$ instead of $\widetilde{O}(mnk)$, let us consider the following multiple response regression: $\min_{V \in \mathbb{R}^{n \times k}} \|UV^\top - M\|_F$. Our strategy is to solve $n$ least-square regression. Consider the $i$-th regression problem: $\min_v \|P_\Omega(Uv) - P_\Omega(M_{*,i})\|_2$, note that $v$ only contributes to the $i$-th column of $UV^\top$ which correlates to the $i$-th column of the operator $P_\Omega$, therefore we only need to solve a regression with the rows of $U$ selected by the $i$-th column of $P_\Omega$. Let $\mathcal{K}_i$ be the corresponding set of nonzero indices, we select the rows of $U$ which is of size $U_{\mathcal{K}_i, *} \in \mathbb{R}^{|\mathcal{K}_i| \times k}$ and the resulting regression is $\min_{v \in \mathbb{R}^k} \|U_{\mathcal{K}_i, *}v - M_{\mathcal{K}_i, i}\|_2$. Sum over all $T$ iterations, the total size of the multiple response regression is $O(\sum_{i=1}^n |\mathcal{K}_i|k) = O(|\Omega|k)$, and our regression solver takes $\widetilde{O}(|\Omega|k + nk^3) = \widetilde{O}(|\Omega|k)$ time, as desired.

We summarize the result in the following lemma.

**Lemma 4.1.** *Let $\Omega \subseteq [m] \times [n]$, $M \in \mathbb{R}^{m \times n}$ and $U \in \mathbb{R}^{m \times k}$. Let $\epsilon, \delta \in (0, 1)$ be precision and failure probability, respectively. There exists an algorithm that takes*

$$O((|\Omega|k \log n + nk^3 \log^2(n/\delta)) \cdot \log(n/\epsilon))$$

*time to output a matrix $\widehat{V}$ such that $\|\widehat{V} - V^*\| \leq \epsilon$ where $V^* = \arg\min_{V \in \mathbb{R}^{n \times k}} \|P_\Omega(UV^\top) - P_\Omega(M)\|_F$. The algorithm succeeds with probability at least $1 - \delta$.*

### 4.4 PUTTING THINGS TOGETHER

Given our fast regression solver and robust analytical framework that effectively handles the perturbation error caused by approximate solve, we are in the position to deliver our main result.

---

[7]We say a matrix $S$ produces a subspace embedding of $\epsilon$-distortion if for any fixed orthonormal basis $U \in \mathbb{R}^{n \times k}$, the singular values of $SU$ lie in $[1 - \epsilon, 1 + \epsilon]$ with high probability.

**Theorem 4.2** (Main result, formal version of Theorem 1.1). *Let $M \in \mathbb{R}^{m \times n}$ be a matrix satisfying Assumption 3.6. Let operator $P_\Omega$ be defined as in Definition 3.3. Then, there exists a randomized algorithm (Algorithm 1) with the following properties:*

- *It samples $|\Omega| = O(\kappa^4 \mu^2 n k^{4.5} \log n \log(k\|M\|_F/\epsilon))$ entries;*

- *It runs in $\widetilde{O}(|\Omega|k)$ time.*

*The algorithm outputs a pair of matrices $\widehat{U} \in \mathbb{R}^{m \times k}$ and $\widehat{V} \in \mathbb{R}^{n \times k}$ such that*

$$\|M - \widehat{U}\widehat{V}^\top\|_F \leq \epsilon$$

*holds with high probability.*

Let us pause and make some remarks regarding the above result. On the sample complexity front, we attain the result achieved in Jain et al. (2013), but we would also like to point out that it can be further improved using the approach developed in Gamarnik et al. (2017). Our robust alternating minimization framework indicates that as long as the error caused by the approximate solver can be polynomially bounded, then the convergence of the algorithm is preserved (up to $\log(1/\epsilon)$ factors). Thus, our framework can be safely integrated into any alternating minimization-based algorithm for matrix completion.

**Comparison with Kelner et al. (2023b).** The runtime of our algorithm is nearly linear in the verification time — given a set of observed entries $\Omega$, it takes $O(k)$ time to verify an entry of $P_\Omega(\widehat{U}\widehat{V}^\top)$, hence requires a total of $O(|\Omega|k)$ time. Kelner et al. (2023b) achieves a similar runtime behavior with an improved sample complexity of $|\Omega| = \widetilde{O}(nk^{2+o(1)})$. It is worth noting that most popular *practical algorithms* for matrix completion are based on either alternating minimization or gradient descent, since they are easy to implement and certain steps can be sped up via fast solvers. In contrary, the machinery of Kelner et al. (2023b) is much more complicated. In short, they need to decompose the update into a "short" progress matrix and a "flat" noise component whose singular values are relatively close to each other. To achieve this goal, their algorithm requires complicated primitives, such as approximating singular values and spectral norms Musco & Musco (2015), Nesterov's accelerated gradient descent Nesterov (1983) (which is known to be hard to realize for practical applications) and a complicated post-process procedure. While it is totally possible that these subroutines can be made practically efficient, empirical studies seem to be necessary to justify its practical performance. In contrast, our algorithm could be interpreted as providing a theoretical foundation on why *fast* alternating minimization works so well in practice. As most of the fast alternating minimization implementations rely on quick, approximate solvers (for instance, Lai & Varghese (2017); Liu et al. (2020)) but most of their analyses assume every step of the algorithm is computed exactly. From this perspective, one can view our robust analytical framework as "completing the picture" for all these variants of alternating minimization. Moreover, if one can further sharpen the dependence on $k$ and condition number $\kappa$ in the sample complexity for alternating minimization, matching that of Kelner et al. (2023b), we automatically obtain an algorithm with the same (asymptotic) complexity of their algorithm. We leave improving the sample complexity of alternating minimization as a future direction.

## 5 CONCLUSION

In this paper, we develop a nearly linear time algorithm for low rank matrix completion via two ingredients: a sketching-based preconditioner for solving multiple response regressions, and a robust framework for alternating minimization.

Our robust alternating minimization framework effectively bridges the gap between theory and practice — as all prior theoretical analysis of alternating minimization requires exact solve of the multiple response regressions, but in practice, fast and cheap approximate solvers are preferred. Our algorithm also has the feature that it runs in time nearly linear in verifying the solution, as given a set of entries $\Omega$, it takes $O(|\Omega|k)$ time to compute the corresponding entries.

ACKNOWLEDGEMENT

We would like to thank Jonathan Kelner for helpful discussions and pointing out many references, as well as comments from anonymous reviewers that significantly improve the presentation of this paper. Yuzhou Gu is supported by the National Science Foundation under Grant No. DMS-1926686. Lichen Zhang is supported by NSF CCF-1955217 and NSF DMS-2022448.

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

APPENDIX

**Roadmap.** In Section A, we introduce more fundamental lemmas and facts. In Section B, we provide more details about our sketching-based solver. In Section C, we support the main init by giving and explaining more definitions and lemmas. In Section D, we explain our main induction hypothesis. In Section E, we analyze the general case of the update rules and notations. In Section F, we give the lemmas for distance shrinking and their proofs: we upper bound different terms by distance. In Section G, we analyze the distance and incoherence under row perturbations.

## A  MORE PRELIMINARY

In this section, we display more fundamental concepts. In Section A.1, we introduce algebraic properties for matrices. In Section A.2, we analyze the properties of angles and distances. In Section A.3, we provide the tools which are used in previous works. In Section A.4, we state several well-known probability tools.

### A.1  BASIC MATRIX ALGEBRA

We state several standard norm inequalities here.

**Fact A.1** (Norm inequalities)**.** *For any matrix $A, B$*

- *Part 1. $\|AB\| \le \|A\| \cdot \|B\|$.*

- *Part 2. $\|AB\| \ge \|A\| \cdot \sigma_{\min}(B)$.*

- *Part 3. $\|A + B\| \le \|A\| + \|B\|$.*

- *Part 4. if $B^\top A = 0$, $\|A + B\| \ge \|A\|$ .*

*For any matrix $A$ and vector $x$*

- *Part 5. $\|Ax\|_2 \le \|A\| \cdot \|x\|_2$.*

- *Part 6. $\|A\| \le \|A\|_F$.*

- *Part 7. $\|A\|_F \le \sqrt{k}\|A\|$ if $A$ is rank-$k$.*

*For any square matrix $A$*

- *Part 8. If $A$ is invertible, we have $\|A\| = 1/\sigma_{\min}(A^{-1})$.*

- *Part 9. For vector $x$ with $\|x\|_2 = 1$, we have $\|Ax\|_2 = \max_{y:\|y\|_2=1} y^\top Ax$.*

*For any matrix $A$, square invertible matrix $B$*

- *Part 10. $\|A\| \le \sigma_{\max}(B) \cdot \|AB^{-1}\|$.*

*For any matrix $A$, and square diagonal invertible matrix $B$*

- *Part 11. $\sigma_{\min}(BA) \ge \sigma_{\min}(B) \cdot \sigma_{\min}(A)$.*

We omit their proofs, as they are quite standard. We also state Weyl's inequality for singular values:

**Lemma A.2** (Weyl (1912))**.** *Let $A, B \in \mathbb{R}^{n \times k}$ where $n \ge k$, then we have for any $i \in [k]$,*

$$|\sigma_i(A) - \sigma_i(B)| \le \|A - B\|.$$

The next fact bounds the spectral norm of a matrix after applying a unitary transformation.

**Fact A.3.** *Let $m \ge k$. Let $U \in \mathbb{R}^{m \times k}$ denote a matrix that has an orthonormal basis.*

- *Part 1. For any matrix $A \in \mathbb{R}^{m \times n}$, we have*

$$\|U^\top A\| \le \|A\|.$$

- *Part 2. For any matrix $B \in \mathbb{R}^{k \times d}$, we have*

$$\|UB\| = \|B\|.$$

*Proof.* **Part 1.** By the property of $U$, we know that $UU^\top \preceq I_m$.

Thus, for any vector $x$, we have

$$x^\top A^\top U U^\top A x \leq x^\top A^\top A x.$$

Thus, $\|U^\top A\| \leq \|A\|$.

**Part 2.** By property of $U$, we have $U^\top U = I_k$.

Thus, for any vector $x$, we have

$$x^\top B^\top U^\top U B x = x^\top B^\top B x.$$

Thus, $\|UB\| = \|B\|$. $\qquad\square$

The following is a collection of simple algebraic facts.

**Fact A.4.** *Let $y \in (0, 0.1)$. We have*

- *Part 1. If $x \in (0, 1/2)$, then $\sqrt{1 - x^2} - yx \geq 1/2$.*

- *Part 2. If $x \geq 1/2$, then $x - y\sqrt{1 - x^2} \geq \frac{1}{2}x$.*

- *Part 3. If $x \in [0, 1]$, then $1 - \sqrt{1 - x^2} \leq x^2$.*

*Proof.* **Proof of Part 1.** We have

$$\begin{aligned}
\sqrt{1 - x^2} - yx &\geq \sqrt{1 - x^2} - \frac{1}{5}x \\
&\geq 1 - \frac{1}{2}x - \frac{1}{5}x \\
&\geq \frac{1}{2},
\end{aligned}$$

where the first step follows from $y \in (0, 0.1)$, the second step follows from $\sqrt{1 - x^2} \geq 1 - \frac{1}{2}x$ for all $x \in [0, 4/3]$, and the last step follows from $x \leq 1/2$.

**Proof of Part 2.** We know that

$$y\sqrt{1 - x^2} \leq y \leq 0.1 \leq x/2, \tag{1}$$

where the first step follows from $0 \leq \sqrt{1 - x^2} \leq 1$, the second step follows from $y \in (0, 0.1)$, and the last step follows from $x \geq 1/2$.

Thus,

$$\begin{aligned}
x - y\sqrt{1 - x^2} &\geq x - x/2 \\
&= x/2,
\end{aligned}$$

where the first step follows from Eq. (1) and the second step follows from simple algebra.

**Proof of Part 3.** We know that

$$1 - x^2 \leq \sqrt{1 - x^2}$$

for all $x$ in the domain of $\sqrt{1 - x^2}$.

Then, the statement is true. $\qquad\square$

## A.2 PROPERTIES OF ANGLES AND DISTANCES

We explore some more relations for angles and distances between subspaces.

Let $U, V \in \mathbb{R}^{n \times k}$ be two matrices with orthonormal columns, we use $\mathrm{dist}_c(V, U)$ to denote the following minimization problem:

$$\mathrm{dist}_c(V, U) = \min_{Q \in O_k} \|VQ - U\|,$$

where $O_k \subset \mathbb{R}^{k \times k}$ is the set of $k \times k$ orthogonal matrices.

The next lemma is a simple application of fundamental subspace decomposition.

**Lemma A.5.** *Let $A \in \mathbb{R}^{n \times k}$ be an orthonormal basis, then there exists a matrix $A_\perp \in \mathbb{R}^{n \times (n-k)}$ with $AA^\top + A_\perp A_\perp^\top = I_n$.*

*Proof.* We know that the column space of $A$ is orthogonal to the null space of $A^\top$, and the null space of $A^\top$ has dimension $n - k$.

Let $A_\perp$ be an orthonormal basis of the null space of $A^\top$.

We have

$$A^\top A_\perp = 0. \tag{2}$$

It remains to show that

$$AA^\top + A_\perp A_\perp^\top = I_n.$$

Let $z \in \mathbb{R}^n$ be any vector.

We know that $z$ either in the column space of $A$ or the null space of $A^\top$.

**Case 1: $z$ in the column space of $A$.** In this case, we can write $z = Ay$.

Then,

$$
\begin{aligned}
AA^\top z + A_\perp A_\perp^\top z &= AA^\top Ay + A_\perp A_\perp^\top Ay \\
&= Ay + A_\perp A_\perp^\top Ay \\
&= Ay + A_\perp (A^\top A_\perp)^\top y \\
&= Ay + 0 \\
&= z,
\end{aligned}
$$

where the first step follows from $z = Ay$, the second step follows from $A$ is orthogonal that $A^\top = A^{-1}$, the third step follows from $(AB)^\top = B^\top A^\top$ for all matrices $A$ and $B$, the fourth step follows from Eq. (2), and the last step follows from $z = Ay$.

**Case 2: $z$ in the null space of $A^\top$.** In this case, we know that $A^\top z = 0$ and $z = A_\perp y$, so

$$
\begin{aligned}
AA^\top z + A_\perp A_\perp^\top z &= 0 + A_\perp A_\perp^\top A_\perp y \\
&= A_\perp y \\
&= z,
\end{aligned}
$$

where the first step follows from $A^\top z = 0$ and $z = A_\perp y$, the second step follows from $A_\perp^\top A_\perp = I$, and the third step follows from $z = A_\perp y$.

Thus, we have shown $AA^\top + A_\perp A_\perp^\top = I_n$. $\qquad\square$

The following lemma presents a simple inequality for orthogonal complements.

**Lemma A.6** (Structural lemma for matrices with orthonormal columns)**.** *Let $U, V \in \mathbb{R}^{n \times k}$ be matrices with orthonormal columns. Then*

$$(V^\top U)_\perp = V_\perp^\top U.$$

*Proof.* Let us first compute the Gram matrix of $V^\top U$, which is

$$
\begin{aligned}
U^\top V V^\top U &= U^\top (I - V_\perp V_\perp^\top) U \\
&= U^\top U - U^\top V_\perp V_\perp^\top U \\
&= I_k - U^\top V_\perp V_\perp^\top U,
\end{aligned}
$$

where the first step follows from $V_\perp V_\perp^\top + V V^\top = I$, the second step follows from simple algebra, and the last step follows from $U$ has orthonormal columns.

This means that $(V^\top U)_\perp = V_\perp^\top U$. $\qquad\square$

The singular vectors can be parametrized by orthogonal complement and inverse, solely.

**Lemma A.7** (Orthogonal and inverse share singular vectors). *Let $A \in \mathbb{R}^{k \times k}$ be non-singular such that there exists $A_\perp \in \mathbb{R}^{(n-k) \times k}$ with $A^\top A + A_\perp^\top A_\perp = I$, then the following holds:*

- *Part 1. $A_\perp$ and $A^{-1}$ have the same set of singular vectors.*

- *Part 2. Let $u$ be a singular vector of $A$, if $u$ corresponds to $\sigma_i(A)$, then it corresponds to $\sigma_{k-i}(A_\perp)$.*

- *Part 3. $\|A_\perp A^{-1}\| = \|A_\perp\| \|A^{-1}\|$.*

*Proof.* **Proof of Part 1.** Let $x \in \mathbb{R}^k$ be the unit eigenvector of $A$ that realizes the spectral norm.

Note that

$$
\|A_\perp x\|_2^2 = 1 - \|A\|^2,
$$

we argue that $x$ corresponds to the smallest singular value of $A_\perp$ via contradiction. Suppose there exists some unit vector $y$ with

$$
\|A_\perp y\|_2 < \|A_\perp x\|_2.
$$

By definition, we know that

$$
\|A_\perp y\|_2^2 + \|Ay\|_2^2 = 1,
$$

which means

$$
\|Ay\|_2 > \|Ax\|_2 = \|A\|,
$$

and it contradicts the definition of spectral norm.

Similarly, if $z$ is the unit vector that realizes the spectral norm of $A_\perp$, then it is also a singular vector corresponds to the smallest singular value of $A$, or equivalently, the spectral norm of $A^{-1}$. Our above argument essentially implies that $A_\perp$ and $A^{-1}$ have the same set of singular vectors.

Our above argument is choosing $x$ to the eigenvector corresponding to the largest singular values. Similarly, we can choose to 2nd, 3rd, and then prove entire sets.

**Proof of Part 2.** The key of the proof is that for any unit vector $u \in \mathbb{R}^k$, we have

$$
\|Au\|_2^2 + \|A_\perp u\|_2^2 = 1,
$$

let $\sigma_1(A), \ldots, \sigma_k(A)$ be the singular values of $A$ and $u_1, \ldots, u_k$ be corresponding singular vectors.

By above definition, we know

$$
\|Au_i\|_2 = \sigma_i(A)
$$

Then, we know

$$
\sigma_i^2(A) + \|A_\perp u_i\|_2^2 = 1,
$$

which means that $A_\perp u_i = \sqrt{1 - \sigma_i^2(A)} u_i$. Note that all singular values of $A_\perp$ are in this form, i.e., we have its singular values being

$$
\sqrt{1 - \sigma_1^2(A)}, \ldots, \sqrt{1 - \sigma_k^2(A)},
$$

as $\sigma_1(A) \geq \ldots \geq \sigma_k(A)$, and the above singular values are in ascending order.

Thus, we have

$$\sigma_{k-i}(A_\perp) = \sigma_i(A).$$

**Proof of Part 3.** The proof is then straightforward.

Suppose that $(\lambda, z)$ is largest singular value and singular vector (e.g. $\|A_\perp\| = \lambda$). Then, we have

$$A_\perp z = \lambda z \tag{3}$$

Using Part 2 (by choosing $i = 1$), we know that $z$ is also the largest singular vector for $A^{-1}$. Assume that $A^{-1}$ largest singular value is $\mu$ (e.g. $\|A^{-1}\| = \mu$). Then we have

$$A^{-1}z = \mu z. \tag{4}$$

Then, we have

$$\begin{aligned}
A_\perp A^{-1}z &= A_\perp \mu z \\
&= \mu(A_\perp z) \\
&= \lambda \mu z,
\end{aligned} \tag{5}$$

where the first step follows from Eq. (4), the second step follows from $\mu$ is a real number and a real number multiplying a matrix is commutative and follows from the associative property, and the third step follows from Eq. (3).

From Eq. (5), we know that

$$\|A_\perp A^{-1}\| \geq \lambda \mu = \|A_\perp\| \cdot \|A^{-1}\|.$$

Using norm inequality, we have

$$\|A_\perp A^{-1}\| \leq \|A_\perp\| \cdot \|A^{-1}\|.$$

Combining the lower bound and upper bound, we have

$$\|A_\perp A^{-1}\| = \|A_\perp\|\|A^{-1}\|,$$

and we have proved the assertion. □

We prove several fundamental trigonometry equalities for subspace angles.

**Lemma A.8.** *Let $U, V \in \mathbb{R}^{n \times k}$ be orthonormal matrices, then*

$$\tan \theta(V, U) = \frac{\sin \theta(V, U)}{\cos \theta(V, U)}.$$

*Proof.* We have,

$$\begin{aligned}
\tan \theta(V, U) &= \|V_\perp^\top U (V^\top U)^{-1}\| \\
&= \|(V^\top U)_\perp (V^\top U)^{-1}\| \\
&= \|(V^\top U)_\perp\|\|(V^\top U)^{-1}\| \\
&= \frac{\|(V^\top U)_\perp\|}{1/\|(V^\top U)^{-1}\|} \\
&= \frac{\|V_\perp^\top U\|}{1/\|(V^\top U)^{-1}\|} \\
&= \frac{\sin \theta(V, U)}{\cos \theta(V, U)},
\end{aligned}$$

where the first step follows from the definition of $\tan \theta(V, U)$ (see Definition 3.2), the second step follows from Lemma A.6, the third step follows from the part 3 of Lemma A.7, the fourth step follows from simple algebra, the fifth step follows from Lemma A.6, and the last step follows from the definition of $\sin \theta(V, U)$ and $\cos \theta(V, U)$ (see Definition 3.2). □

**Lemma A.9.** *Let $U, V \in \mathbb{R}^{n \times k}$ be orthogonal matrices, then*

$$\sin^2 \theta(V, U) + \cos^2 \theta(V, U) = 1.$$

*Proof.* Recall that $\cos \theta(V, U) = \frac{1}{\|(V^\top U)^{-1}\|}$ and $\sin \theta(V, U) = \|V_\perp^\top U\|$.

By Lemma A.6, we know that

$$(V^\top U)_\perp = V_\perp^\top U,$$

so by the definition of $\sin \theta(V, U)$ (see Definition 3.2), we have

$$\sin \theta(V, U) = \|(V^\top U)_\perp\|.$$

We define matrix $A \in \mathbb{R}^{k \times k}$ as follows,

$$A := V^\top U. \tag{6}$$

By part 1 of Lemma A.7, we know that $A_\perp$ and $A^{-1}$ have the same set of singular vectors.

Let $z \in \mathbb{R}^k$ be the unit singular vector with singular value $\|A_\perp\|$.

This implies

$$\|A_\perp z\|_2 = \|A_\perp\|. \tag{7}$$

Note that $A_\perp$ and $A^{-1}$ have the same singular vectors implies that the singular vector realizing $\|A_\perp\|$ corresponds to the smallest singular value of $A$, i.e.,

$$\|Az\|_2 = \sigma_{\min}(A). \tag{8}$$

Then, we have

$$
\begin{aligned}
1 &= z^\top z \\
&= z^\top (A^\top A + A_\perp^\top A_\perp) z \\
&= z^\top A^\top A z + z^\top A_\perp^\top A_\perp z \\
&= \|Az\|_2^2 + \|A_\perp z\|_2^2 \\
&= \|Az\|_2^2 + \|A_\perp\|^2 \\
&= \sigma_{\min}^2(A) + \|A_\perp\|^2,
\end{aligned}
\tag{9}
$$

where the first step follows from $\|z\|_2^2 = 1$, the second step follows from $A^\top A + A_\perp^\top A_\perp = I$, the third step follows from simple algebra, the fourth step follows from $x^\top B^\top B x = \|Bx\|_2^2$, the fifth step follows from Eq. (7), and the sixth step follows from Eq. (8).

Also, we get

$$
\begin{aligned}
\|A_\perp\|^2 + \sigma_{\min}^2(A) &= \|(V^\top U)_\perp\|^2 + \sigma_{\min}^2(V^\top U) \\
&= \|V_\perp^\top U\|^2 + \sigma_{\min}^2(V^\top U) \\
&= \sin^2 \theta(V, U) + \cos^2 \theta(V, U),
\end{aligned}
$$

where the first step follows from Eq. (6), the second step follows from Lemma A.6, the third step follows from definitions of $\sin \theta$ and $\cos \theta$ (see Def. 3.2).

Therefore, by Eq. (9), we have $\sin^2 \theta(V, U) + \cos^2 \theta(V, U) = 1$. This completes the proof. $\qquad \square$

The next lemma demonstrates relationship between several trigonometry definitions.

**Lemma A.10.** *Let $V, U \in \mathbb{R}^{n \times k}$ be two orthogonal matrices, then*

- *Part 1.* $\sin \theta(V, U) \leq \tan \theta(V, U)$.

- *Part 2.* $\frac{1-\cos\theta(V,U)}{\cos\theta(V,U)} \leq \tan\theta(V,U)$.

- *Part 3.* $\sin\theta(V,U) \leq \mathrm{dist}_c(V,U)$.

- *Part 4.* $\mathrm{dist}_c(V,U) \leq \sin\theta(V,U) + \frac{1-\cos\theta(V,U)}{\cos\theta(V,U)}$.

- *Part 5.* $\mathrm{dist}_c(V,U) \leq 2\tan\theta(V,U)$.

*Proof.* We define

$$Q^* := \arg\min_{Q\in O^{k\times k}} \|VQ - U\|,$$

Next, we define matrix $R$ to be

$$R := U - VQ^*. \tag{10}$$

Then we have

$$\mathrm{dist}_c(V,U) = \|R\| \tag{11}$$

and

$$\begin{aligned}
\sin\theta(V,U) &= \|V_\perp^\top U\| \\
&= \|V_\perp^\top (VQ^* + R)\| \\
&= \|V_\perp^\top VQ^* + V_\perp^\top R\| \\
&= \|V_\perp^\top R\| \\
&\leq \|V_\perp^\top\|\|R\| \\
&\leq \|R\|,
\end{aligned} \tag{12}$$

where the first step follows from the definition of

$$\sin\theta(V,U),$$

the second step follows from $U = VQ^* + R$ (see Eq. (10)), the third step follows from simple algebra, the fourth step follows from $V_\perp^\top V = (V^\top V_\perp)^\top = 0$ (see Lemma A.5), the fifth step follows from $\|AB\| \leq \|A\|\|B\|$, and the last step follows from $\|V_\perp^\top\| \leq 1$.

**Proof of Part 1.** Note that

$$\begin{aligned}
\sin\theta(V,U) &= \tan\theta(V,U) \cdot \cos\theta(V,U) \\
&\leq \tan\theta(V,U),
\end{aligned}$$

where the first step follows from Lemma A.8, the last step follows from $\cos\theta(V,U) \leq 1$.

**Proof of Part 2.** For simplicity, we use $\theta$ to represent $\theta(V,U)$.

The statement is

$$\frac{1-\cos\theta}{\cos\theta} \leq \tan\theta,$$

which implies

$$1 - \cos\theta \leq \sin\theta.$$

Taking the square on both sides, we get

$$1 - 2\cos\theta + \cos^2\theta \leq \sin^2\theta.$$

Using Lemma A.9, the above equation is further equivalent to

$$2\cos\theta(\cos\theta - 1) \leq 0.$$

This is true forever, since $\cos\theta \in [0,1]$.

**Proof of Part 3.** Given $\text{dist}_c(V,U) = \|R\|$ (Eq. (11)) and $\sin\theta(V,U) \leq \|R\|$ (Eq. (12)), we have
$$\sin\theta(V,U) \leq \text{dist}_c(V,U).$$

**Proof of Part 4.** We define $A, D, B$ as the SVD of matrix $V^\top U$ as follows
$$ADB^\top = \text{SVD}(V^\top U),$$

then

$$A = V^\top UBD^{-1} \tag{13}$$

and

$$\sigma_{\min}(D) = \sigma_{\min}(V^\top U) = \cos\theta(V,U).$$

In addition,

$$
\begin{aligned}
(AB^\top)^\top &= BA^\top \\
&= (B^\top)^\top A^\top \\
&= (B^\top)^{-1} A^{-1} \\
&= (AB^\top)^{-1},
\end{aligned}
$$

where the first step follows from $(AB)^\top = B^\top A^\top$ for all matrices $A$ and $B$, the second step follows from the simple property of matrix, the third step follows from the fact that $B$ is orthogonal, and the last step follows from simple algebra, i.e., $AB^\top \in O^{k \times k}$.

For $\text{dist}_c(V,U)$, we have

$$
\begin{aligned}
\text{dist}_c(V,U) &\leq \|VAB^\top - U\| \\
&= \|VV^\top VBD^{-1}B^\top - U\| \\
&= \|VV^\top VBD^{-1}B^\top - VV^\top U + VV^\top U - U\| \\
&\leq \|VV^\top UBD^{-1}B^\top - VV^\top U\| + \|VV^\top U - U\|, \tag{14}
\end{aligned}
$$

where the first step follows from $AB^\top$ can not provide a better cost than minimizer, the second step follows from Eq. (13), the third step follows from simple algebra, and the last step follows from the triangle inequality.

For the second term of Eq. (14), namely $\|VV^\top U - U\|$, we have

$$
\begin{aligned}
\|VV^\top U - U\| &= \|(VV^\top - I)U\| \\
&= \sin\theta(V,U),
\end{aligned}
$$

where the first step follows from simple algebra and the second step follows from the definition of $\sin\theta(V,U)$ (see Definition 3.2).

For the first term of Eq. (14), namely $\|VV^\top UBD^{-1}B^\top - VV^\top U\|$, we have

$$
\begin{aligned}
\|VV^\top UBD^{-1}B^\top - VV^\top U\| &= \|VV^\top U(BD^{-1}B^\top - I)\| \\
&\leq \|BD^{-1}B^\top - I\| \\
&= \|D^{-1} - I\| \\
&= \frac{1 - \cos\theta(V,U)}{\cos\theta(V,U)},
\end{aligned}
$$

where the first step follows from simple algebra, the second step follows from $\|VV^\top U\| \leq 1$, the third step follows from $B$ is orthonormal basis, and the last step follows from $\|D^{-1} - I\| = |\frac{1}{\sigma_{\min}(D)} - 1| = |\frac{1}{\cos\theta} - 1| = \frac{1-\cos\theta}{\cos\theta}$.

**Proof of Part 5.** We have

$$
\begin{aligned}
\text{dist}_c(V,U) &\leq \sin\theta(V,U) + \frac{1 - \cos\theta(V,U)}{\cos\theta(V,U)} \\
&\leq \sin\theta(V,U) + \tan\theta(V,U) \\
&\leq 2\tan\theta(V,U),
\end{aligned}
$$

where the first step follows from Part 4, the second step follows from Part 2, and the last step follows from Part 1. $\qquad\square$

## A.3 Tools from Prior Works

The previous paper Jain et al. (2013) assumes the entries of $M$ are sampled independently with the following probability. Note that the choice of $p$ also determines the sample complexity of the algorithm.

**Definition A.11** (Sampling probability). *Let $C \geq 10$ denote a sufficiently large constant. We define sample probability to be*

$$p := C \cdot (\sigma_1^*/\sigma_k^*)^2 \cdot \mu^2 \cdot k^{2.5} \cdot \log n \cdot \log(k\|M\|_F/\epsilon)/(m\delta_{2k}^2).$$

*Then, we sample $\Omega \sim [m] \times [n]$ each coordinately independently according to that probability.*

The parameter $\delta_{2k}$ is a tuning parameter for sampling probability.

**Definition A.12.** *We choose $\delta_{2k}$ as follows*

$$\delta_{2k} := \frac{1}{100}\frac{1}{k}\sigma_k^*/\sigma_1^*.$$

*We can see that $\delta_{2k} \in (0, 0.01)$.*

Given the set of indices, we denote the row and column selection operator as follows.

**Definition A.13.** *For each $j \in [n]$, we define set $\Omega_{*,j} \subset [m]$*

$$\Omega_{*,j} := \{i \in [m] \ : \ (i,j) \in \Omega\}.$$

*For each $i \in [m]$, we define set $\Omega_{i,*} \subset [m]$,*

$$\Omega_{i,*} := \{j \in [n] \ : \ (i,j) \in \Omega\}.$$

The next structural lemma bounds the inner product between vectors inside set $\Omega$.

**Lemma A.14** (Lemma C.5 in Jain et al. (2013)). *Let $\Omega \subset [m] \times [n]$ be a set $f$ indices sampled uniformly at random from $[m] \times [n]$ with each element of $[m] \times [n]$ sampled independently with probability $p \geq C(\log n)/m$. Let $C > 1$ be a sufficiently large constant.*

*Then with probability $1 - 1/\operatorname{poly}(n)$, for all $x \in \mathbb{R}^m$ with $\sum_{i=1}^m x_i = 0$ and for all $y \in \mathbb{R}^n$, we have*

$$\sum_{(i,j)\in\Omega} x_i y_j \leq C(mn)^{1/4}p^{1/2}\|x\|_2\|y\|_2.$$

The following lemma bounds the spectral gap between two matrix inverses in terms of the larger of their spectral norm squared, times the spectral norm of their differences.

**Lemma A.15** (Wedin (1973), see Theorem 1.1 in Meng & Zheng (2010) as an example). *For two conforming real matrices $A$ and $B$,*

$$\|A^\dagger - B^\dagger\| \leq 2 \cdot \max\{\|A^\dagger\|^2, \|B^\dagger\|^2\} \cdot \|A - B\|$$

The next claim bounds the inner product between any unit vector and rows of incoherent matrix.

**Claim A.16.** *Suppose $U_t$ is $\mu_2$ incoherent, and $U_*$ is $\mu$ incoherent. For any unit vectors $x, y$ we have*

- *Part 1. $\sum_{i\in[m]}\langle x, U_{t,i}\rangle^4 \leq \mu_2^2 k/m$*

- *Part 2. $\sum_{i\in[m]}\langle y, U_{*,i}\rangle^2 \cdot \langle x, U_{t,i}\rangle^2 \leq \mu^2 k/m$.*

*Proof.* **Proof of Part 1.** Note that $\|U_{t,i}\|_2 \leq 1$, so

$$\begin{aligned}\langle x, U_{t,i}\rangle &= \|U_{t,i}\|_2 \cdot \langle x, U_{t,i}/\|U_{t,i}\|_2\rangle \\ &\leq \|U_{t,i}\|_2,\end{aligned} \tag{15}$$

where the second step follows from the inner product between two unit vectors is at most 1.

Thus, based on Eq. (15),

$$\langle x, U_{t,i}\rangle^2 \leq \|U_{t,i}\|_2^2$$

$$\leq \mu_2^2 k/m. \tag{16}$$

where the last step follows from $U_t$ is $\mu_2$ incoherent.

We have

$$\sum_{i=1}^{m} \langle x, U_{t,i} \rangle^4 \leq \max_{i \in [m]} \langle x, U_{t,i} \rangle^2 \sum_{i=1}^{m} \langle x, U_{t,i} \rangle^2$$

$$\leq \frac{\mu_2^2 k}{m} \sum_{i=1}^{m} \langle x, U_{t,i} \rangle^2$$

$$= \frac{\mu_2^2 k}{m} x^\top U_t^\top U_t x$$

$$= \frac{\mu_2^2 k}{m} x^\top x$$

$$= \frac{\mu_2^2 k}{m},$$

where the first step follows from $\sum_{i \in [m]} a_i b_i \leq \max_{i \in [m]} a_i \sum_{i \in [m]} b_i$, the second step follows from Eq. (16), the third step follows from $U_t^\top U_t = \sum_{i=1}^{m} U_{t,i}^\top U_{t,i}$, the fourth step follows from $U_t^\top U_t = I_k$, and the last step follows from the Lemma statement that $x$ is a unit vector.

**Proof of Part 2.** Similarly as the proof of part 1. We know that

$$\langle y, U_{*,i} \rangle^2 \leq \|U_{*,i}\|_2^2 \leq \mu^2 k/m. \tag{17}$$

We can show

$$\sum_{i=1}^{m} \langle x, U_{t,i} \rangle^2 \cdot \langle y, U_{*,i} \rangle^2 \leq \max_{i \in [m]} \langle y, U_{*,i} \rangle^2 \cdot \sum_{i=1}^{m} \langle x, U_{t,i} \rangle^2$$

$$\leq \frac{\mu^2 k}{m} \cdot \sum_{i=1}^{m} \langle x, U_{t,i} \rangle^2$$

$$= \frac{\mu^2 k}{m} \cdot x^\top U_t^\top U_t x$$

$$= \frac{\mu^2 k}{m} x^\top x$$

$$= \frac{\mu^2 k}{m},$$

where the first step follows from $\sum_{i \in [m]} a_i b_i \leq \max_{i \in [m]} a_i \sum_{i \in [m]} b_i$, the second step follows from Eq. (17), the third step follows from $U_t^\top U_t = \sum_{i=1}^{m} U_{t,i}^\top U_{t,i}$, the fourth step follows from $U_t^\top U_t = I_k$, and the last step follows from the Lemma statement that $x$ is a unit vector. □

## A.4 Probability Tools

We introduce several well-known probability inequalities.

**Lemma A.17** (Chernoff bound). *Let $X = \sum_{i=1}^{n} X_i$, where $X_i = 1$ with probability $p_i$ and $X_i = 0$ with probability $1 - p_i$, and all $X_i$ are independent. Let $\mu = \mathbb{E}[X] = \sum_{i=1}^{n} p_i$. Then*

- $\Pr[X \geq (1 + \delta)\mu] \leq \exp(-\delta^2 \mu/3), \forall \delta > 0;$

- $\Pr[X \leq (1 - \delta)\mu] \leq \exp(-\delta^2 \mu/2), \forall \delta \in (0, 1).$

**Lemma A.18** (Bernstein inequality). *Let $X_1, \cdots, X_n$ be independent zero-mean random variables (i.e., $\mathbb{E}[X_i] = 0$ for all $i \in [n]$). Suppose that $|X_i| \leq M$ almost surely, for all $i \in [n]$.*

*Then, for any positive $t$, we have*

$$\Pr[\sum_{i=1}^{n} X_i > t] \leq \exp(-\frac{t^2/2}{\sum_{j=1}^{n} \mathbb{E}[X_j^2] + Mt/3}).$$

# B  HIGH ACCURACY WEIGHTED REGRESSION SOLVER

We demonstrate our high accuracy, iterative solver for weighted multiple response regression in this section.

## B.1  DENSE, HIGH ACCURACY AND ITERATIVE SOLVER

Here, we mainly focus on the SRHT matrix, the algorithm that processes it, and its properties. We start by introducing the definition of the SRHT matrix Lu et al. (2013).

**Definition B.1** (Subsampled randomized Hadamard transform (SRHT)). *The SRHT matrix $S \in \mathbb{R}^{m \times n}$ is defined as $S := \frac{1}{\sqrt{m}} PHD$, where each row of matrix $P \in \{0, 1\}^{m \times n}$ contains exactly one 1 at a random position. $H$ is the $n \times n$ Hadamard matrix. $D$ is an $n \times n$ diagonal matrix with each diagonal entry being a value in $\{-1, +1\}$ with equal probability.*

**Remark B.2.** *For an $n \times d$ matrix $A$, $SA$ can be computed in time $O(nd \log n)$.*

The following lemma states that a suitable chosen SRHT provides the so-called subspace embedding property.

**Lemma B.3** (Tropp (2011)). *Let $S \in \mathbb{R}^{m \times n}$ be an SRHT matrix as in Def. B.1. Let $\epsilon_{\mathrm{ose}}, \delta_{\mathrm{ose}} \in (0, 1)$ be parameters. Then for any integer $d \leq n$, if $m_{\mathrm{sk}} = O(\epsilon_{\mathrm{ose}}^{-2} d \log(n/\delta_{\mathrm{ose}}))$, then matrix $S$ is an $(\epsilon_{\mathrm{ose}}, \delta_{\mathrm{ose}})$ oblivious subspace embedding, i.e., for any fixed orthonormal basis $U \in \mathbb{R}^{n \times d}$ with probability at least $1 - \delta_{\mathrm{ose}}$, the singular values of $SU$ lie in $[1 - \epsilon_{\mathrm{ose}}, 1 + \epsilon_{\mathrm{ose}}]$.*

SRHT can be further utilized for a high accuracy regression solver, as presented by the following lemma.

**Lemma B.4** (Dense and high accuracy regression, Avron et al. (2010)). *Given a matrix $A \in \mathbb{R}^{n \times d}$ and a vector $b \in \mathbb{R}^n$, let $\epsilon_1 \in (0, 0.1)$ and $\delta_1 \in (0, 0.1)$, there exists an algorithm that takes time*

$$O((nd \log n + d^3 \log^2(n/\delta_1)) \log(1/\epsilon_1))$$

*and outputs $x' \in \mathbb{R}^d$ such that*

$$\|Ax' - b\|_2 \leq (1 + \epsilon_1) \min_{x \in \mathbb{R}^d} \|Ax - b\|_2$$

*holds with probability $1 - \delta_1$.*

The above result also extends to weighted regression, measured in terms of the $\|v\|_w^2 = \sum_{i=1}^n w_i v_i^2$ norm. For the sake of completeness, we include the algorithm and a proof here.

---

**Algorithm 2** High precision solver

---

1: **procedure** HIGHPRECISIONREG($A \in \mathbb{R}^{n \times d}, b \in \mathbb{R}^n, n, d, \epsilon_1 \in (0, 1), \delta_1 \in (0, 1)$)  ▷ Lemma B.4
2:      $T_1 \leftarrow \Theta(\log(1/\epsilon_1))$
3:      $\epsilon_{\mathrm{ose}} \leftarrow 0.01$
4:      $\delta_{\mathrm{ose}} \leftarrow \delta_1$
5:      $m_{\mathrm{sk}} \leftarrow \Theta(\epsilon_{\mathrm{ose}}^{-2} \cdot d \cdot \log^2(n/\delta_{\mathrm{ose}}))$
6:      Let $S \in \mathbb{R}^{m_{\mathrm{sk}} \times n}$ be an SRHT matrix
7:      Compute QR decomposition of $SA = QR^{-1}$
8:      $x_0 \leftarrow \arg\min_{x \in \mathbb{R}^d} \|SARx - Sb\|_2$
9:      **for** $t = 0 \to T_1$ **do**
10:          $x_{t+1} \leftarrow x_t + R^\top A^\top (b - ARx_t)$
11:      **end for**
12:      **return** $Rx_{T_1}$
13: **end procedure**

---

*Proof of Lemma B.4.* Let us analyze Algorithm 2, first on its convergence then on its runtime.

Note that the $S$ we choose is an $(\epsilon_{\mathrm{ose}}, \delta_{\mathrm{ose}})$-oblivious subspace embedding. Since $SA = QR^{-1}$ where $Q$ is orthonormal, we know the singular values of $AR$ are between $[1 - \epsilon_{\mathrm{ose}}, 1 + \epsilon_{\mathrm{ose}}]$.

Let $AR = U\Sigma V^\top$ be the SVD of $AR$ and $x^*$ denote the optimal solution to the regression $\min_{x \in \mathbb{R}^d} \|ARx - b\|_2$.

Let us consider

$$
\begin{aligned}
&AR(x_{t+1} - x^*) \\
&= AR(x_t + R^\top A^\top(b - ARx_t) - x^*) \\
&= AR(x_t - x^*) + ARR^\top A^\top b - ARR^\top A^\top ARx_t \\
&= AR(x_t - x^*) + ARR^\top A^\top ARx^* - ARR^\top A^\top ARx_t \\
&= (AR - ARR^\top A^\top AR)(x_t - x^*) \\
&= (U\Sigma V^\top - U\Sigma^3 V^\top)(x_t - x^*),
\end{aligned}
\tag{18}
$$

where the first step follows from the definition of $x_{t+1}$ from Algorithm 2, the second step follows from simple algebra, the third step follows from $b = ARx^*$, the fourth step follows from simple algebra, the last step follows from the SVD, $AR = U\Sigma V^\top$.

Therefore,

$$
\begin{aligned}
\|AR(x_{t+1} - x^*)\|_2 &= \|(U\Sigma V^\top - U\Sigma^3 V^\top)(x_t - x^*)\|_2 \\
&= \|(\Sigma - \Sigma^3)V^\top(x_t - x^*)\|_2 \\
&\leq O(\epsilon_{\mathrm{ose}}) \cdot \|V^\top(x_t - x^*)\|_2 \\
&\leq \frac{O(\epsilon_{\mathrm{ose}})}{1 - \epsilon_{\mathrm{ose}}} \|\Sigma V^\top(x_t - x^*)\|_2 \\
&= O(\epsilon_{\mathrm{ose}}) \cdot \|\Sigma V^\top(x_t - x^*)\|_2 \\
&= O(\epsilon_{\mathrm{ose}}) \cdot \|U\Sigma V^\top(x_t - x^*)\|_2 \\
&= O(\epsilon_{\mathrm{ose}}) \cdot \|AR(x_t - x^*)\|_2,
\end{aligned}
$$

where the first step follows from Eq. (18), the second step follows from $U^\top U = I$, the third step follows from $\|AB\| \leq \|A\| \cdot \|B\|$, the fourth step follows from $(1 - \epsilon_{\mathrm{ose}}) \leq \|\Sigma\|$, the fifth step follows from $\epsilon_{\mathrm{ose}} \in (0, 0.1)$, the sixth step follows from $U^\top U = I$, and the last step follows from the SVD, $AR = U\Sigma V^\top$.

This means the error shrinks by a factor of $O(\epsilon_{\mathrm{ose}})$ per iteration. After $T = O(\log(1/\epsilon_1))$ iterations, we have

$$
\|AR(x_T - x^*)\|_2 \leq O(\epsilon_1) \cdot \|AR(x_0 - x^*)\|_2,
\tag{19}
$$

and recall for initial solution $x_0$, we have

$$
\|ARx_0 - b\|_2 \leq (1 + \epsilon_{\mathrm{ose}}) \cdot \|ARx^* - b\|_2.
$$

The above equation implies that

$$
\|ARx_0 - b\|_2^2 - \|ARx^* - b\|_2^2 \leq O(\epsilon_{\mathrm{ose}})\|ARx^* - b\|_2^2.
\tag{20}
$$

We can wrap up the proof as follows:

$$
\begin{aligned}
&\|ARx_T - b\|_2^2 \\
&= \|AR(x_T - x^*)\|_2^2 + \|ARx^* - b\|_2^2 \\
&\leq O(\epsilon_1^2) \cdot \|AR(x_0 - x^*)\|_2^2 + \|ARx^* - b\|_2^2 \\
&= O(\epsilon_1^2) \cdot (\|ARx_0 - b\|_2^2 - \|ARx^* - b\|_2^2) + \|ARx^* - b\|_2^2 \\
&\leq O(\epsilon_1^2) \cdot (O(\epsilon_{\mathrm{ose}})\|ARx^* - b\|_2^2) + \|ARx^* - b\|_2^2 \\
&= (1 + O(\epsilon_1^2)) \cdot \|ARx^* - b\|_2^2,
\end{aligned}
$$

where the first step follows from the Pythagorean theorem, the second step follows from Eq. (19), the third step follows from the Pythagorean theorem again, the fourth step follows from Eq. (20), and the fifth step follows from $\epsilon_{\mathrm{ose}} \leq 1$.

It remains to show the runtime. Applying $S$ to $A$ takes $O(nd \log n)$ time, the QR decomposition takes $O(m_{\mathrm{sk}} d^2) = O(d^3 \log^2(n/\delta_{\mathrm{ose}}))$ time.

Inverting $d \times d$ matrix $Q$ takes $O(d^3)$ time. To solve for $x_0$, we need to multiply $SA$ with $R$ in $O(m_{\mathrm{sk}} d^2)$ time and the solve takes $O(m_{\mathrm{sk}} d^2)$ time as well. To implement each iteration, we multiply from right to left which takes $O(nd)$ time. Putting things together gives the desired runtime. $\qquad\square$

## B.2 BACKWARD ERROR OF REGRESSION

We show how to translate the error from $\|Ax' - b\|_2$ to $\|x' - x_{\mathrm{OPT}}\|_2$ via high-accuracy solver. In the low accuracy sketch-and-solve paradigm, SRHT provides a stronger $\ell_\infty$ guarantee, but for high accuracy solver, we only attain an $\ell_2$ backward error bound.

**Lemma B.5** (Backward error of regression). *Given a matrix $A \in \mathbb{R}^{n \times d}$, a vector $b \in \mathbb{R}^n$. Suppose there exists a vector $x' \in \mathbb{R}^d$ such that*

$$\|Ax' - b\|_2 \le (1 + \epsilon_1) \min_{x \in \mathbb{R}^d} \|Ax - b\|_2.$$

*Let $x_{\mathrm{OPT}}$ denote the exact solution to the regression problem, then it holds that*

$$\|x' - x_{\mathrm{OPT}}\|_2 \le O(\sqrt{\epsilon_1}) \cdot \frac{1}{\sigma_{\min}(A)} \cdot \|Ax_{\mathrm{OPT}} - b\|_2.$$

*Proof.* Note that

$$\|Ax' - Ax_{\mathrm{OPT}}\|_2 = \|Ax' - b - (Ax_{\mathrm{OPT}} - b)\|_2,$$

so we can perform the following decomposition:

$$\begin{aligned}
\|A(x' - x_{\mathrm{OPT}})\|_2^2 &= \|Ax' - b - (Ax_{\mathrm{OPT}} - b)\|_2^2 \\
&= \|Ax' - b\|_2^2 - \|Ax_{\mathrm{OPT}} - b\|_2^2 \\
&\le (1 + \epsilon_1)^2 \|Ax_{\mathrm{OPT}} - b\|_2^2 - \|Ax_{\mathrm{OPT}} - b\|_2^2 \\
&\le 4\epsilon_1 \cdot \|Ax_{\mathrm{OPT}} - b\|_2^2,
\end{aligned} \qquad (21)$$

where the first step follows from simple algebra, the second step follows from the Pythagorean theorem, the third step follows from the assumption of Lemma B.5, and the fourth step follows from simple algebra.

We expand more and explain why the second step follows from Pythagorean theorem. Ultimately, we will show that $Ax_{\mathrm{OPT}} - b$ is orthogonal to any vector in the column space of $A$. Recall that $x_{\mathrm{OPT}}$ can be written in terms of the solution to the normal equation:

$$x_{\mathrm{OPT}} = (A^\top A)^\dagger A^\top b,$$

note that $A$ is not necessarily full column rank, hence we use pseudo-inverse. Let $A = U\Sigma V^\top$ be the SVD of $A$, then the above can be written as

$$\begin{aligned}
x_{\mathrm{OPT}} &= (V\Sigma^2 V^\top)^\dagger V\Sigma U^\top b \\
&= V(\Sigma^\dagger)^2 V^\top V\Sigma U^\top b \\
&= V(\Sigma^\dagger)^2 \Sigma U^\top b \\
&= V\Sigma^\dagger U^\top b,
\end{aligned}$$

where we crucially use the fact that for a diagonal matrix $\Sigma$, the pseudo-inverse is by taking the inverse of nonzero diagonal entries, and keeping zero for remaining zero diagonal entries. We can therefore express $Ax_{\mathrm{OPT}} - b$ as

$$\begin{aligned}
Ax_{\mathrm{OPT}} - b &= U\Sigma V^\top V\Sigma^\dagger U^\top b - b \\
&= (U I_{\mathrm{rank}(A)} U^\top - I)b,
\end{aligned}$$

where $I_{\mathrm{rank}(A)}$ is the matrix whose first $\mathrm{rank}(A)$ diagonal entries are 1, and remaining diagonal entries are 0. Finally, we compute $A^\top(Ax_{\mathrm{OPT}} - b)$:

$$A^\top(Ax_{\mathrm{OPT}} - b) = V\Sigma U^\top(U I_{\mathrm{rank}(A)} U^\top - I)b$$

$$= (V\Sigma U^\top - V\Sigma U^\top)b$$
$$= (A^\top - A^\top)b$$
$$= \mathbf{0}_n,$$

where for the second step, we use $U^\top U = I$, $\Sigma I_{\mathrm{rank}(A)} = \Sigma$ since $\Sigma$ only has its first $\mathrm{rank}(A)$ diagonal entries being nonzero. This justifies the use of Pythagorean theorem.

We can express the SVD of $A^\dagger$ in a similar manner:

$$A^\dagger = V\Sigma^\dagger U^\top,$$

therefore, we must have

$$A^\dagger A = V\Sigma^\dagger U^\top U\Sigma V^\top$$
$$= V I_{\mathrm{rank}(A)} V^\top$$

and it acts as a contracting mapping, in the sense that for any $y \in \mathbb{R}^d$,

$$\|A^\dagger A y\|_2 = \|V I_{\mathrm{rank}(A)} V^\top y\|_2$$
$$= \|I_{\mathrm{rank}(A)} V^\top y\|_2$$
$$\leq \|V^\top y\|_2$$
$$\leq \|V^\top\|\|y\|_2$$
$$\leq \|y\|_2,$$

where the third step is due to we only need to compute the $\ell_2$ norm on a sub-vector, and the last step is by $\|V^\top\| \leq 1$.

Therefore, we have

$$\|x' - x_{\mathrm{OPT}}\|_2 \leq \|A^\dagger A(x' - x_{\mathrm{OPT}})\|_2$$
$$\leq \|A(x' - x_{\mathrm{OPT}})\|_2 \cdot \|A^\dagger\|$$
$$\leq 2\sqrt{\epsilon_1} \cdot \|Ax_{\mathrm{OPT}} - b\|_2 \cdot \|A^\dagger\|$$
$$= \frac{2\sqrt{\epsilon_1}}{\sigma_{\min}(A)} \cdot \|Ax_{\mathrm{OPT}} - b\|_2,$$

where the first step follows from $A^\dagger A$ is a contracting map, the second step follows from $\|ABx\|_2 \leq \|Ax\|_2\|B\|$, the third step follows from Eq. (21), and the last step follows from $\|A^\dagger\| = \frac{1}{\sigma_{\min}(A)}$. $\square$

### B.3 REDUCING WEIGHTED LINEAR REGRESSION TO LINEAR REGRESSION

Solving the regression in matrix completion involves computing the Hadamard product with a binary matrix, we further generalize it to a nonnegative weight matrix, and show it's equivalent up to a rescaling.

**Lemma B.6** (Reducing weighted linear regression to linear regression). *Given a matrix $A \in \mathbb{R}^{n \times d}$, a vector $b \in \mathbb{R}^n$, and a weight vector $w \in \mathbb{R}^n_{\geq 0}$.*

*Let $\epsilon_1 \in (0, 0.1)$ and $\delta_1 \in (0, 0.1)$.*

*Suppose there exists a regression solver that runs in time $\mathcal{T}(n, d, \epsilon_1, \delta_1)$ and outputs a vector $x' \in \mathbb{R}^d$ such that*

$$\|Ax' - b\|_2 \leq (1 + \epsilon_1) \min_{x \in \mathbb{R}^d} \|Ax - b\|_2$$

*with probability at least $1 - \delta_1$, then there is an algorithm that runs in*

$$O(\mathrm{nnz}(A)) + \mathcal{T}(n, d, \epsilon_1, \delta_1)$$

*time outputs $x' \in \mathbb{R}^d$ such that*

$$\|Ax' - b\|_w \leq (1 + \epsilon_1) \min_{x \in \mathbb{R}^d} \|Ax - b\|_w$$

*holds with probability $1 - \delta_1$.*

*Proof.* Recall that for a vector $z \in \mathbb{R}^n$, $\|z\|_w^2 = \sum_{i=1}^n D_{W_i} x_i^2$, we can define the following simple transformation on the regression problem: let $\widetilde{b}_i := D_{\sqrt{W_i}} b_i$ and $\widetilde{A}_{i,:} := D_{\sqrt{W_i}} A_{i,:}$.

We can then solve the regression on the transformed problem $\min_{x \in \mathbb{R}^d} \|\widetilde{A}x - \widetilde{b}\|_2$.

Let $x' \in \mathbb{R}^d$ be the solution such that

$$\|\widetilde{A}x' - \widetilde{b}\|_2 \leq (1 + \epsilon_1) \min_{x \in \mathbb{R}^d} \|\widetilde{A}x - \widetilde{b}\|_2,$$

then for any $y \in \mathbb{R}^d$

$$\|\widetilde{A}y - \widetilde{b}\|_2^2 = \sum_{i=1}^n (\widetilde{A}_{i,:}^\top y - \widetilde{b}_i)^2$$

$$= \sum_{i=1}^n (D_{\sqrt{W_i}} A_{i,:}^\top y - D_{\sqrt{W_i}} b_i)^2$$

$$= \sum_{i=1}^n (D_{\sqrt{W_i}} (A_{i,:}^\top y - b_i))^2$$

$$= \sum_{i=1}^n D_{W_i} (A_{i,:}^\top y - b_i)^2$$

$$= \|Ay - b\|_w^2,$$

where the first step follows from the definition of $\|\cdot\|_2^2$, the second step follows from the definition of $\widetilde{A}_{i,:}$ and $\widetilde{b}_i$, the third step follows from simple algebra, the fourth step follows from $(xy)^2 = x^2 y^2$, and the last step follows from the definition of $\|Ay - b\|_w^2$. This finishes the proof. □

### B.4 SOLVING WEIGHTED MULTIPLE RESPONSE REGRESSION IN WEIGHT SPARSITY TIME

The main result of this section is an algorithm, together with a weighted multiple response analysis that runs in time proportional to the sparsity of weight matrix $W$.

---

**Algorithm 3** Fast, high precision solver for weighted multiple response regression. We will use this algorithm in Algorithm 5 (the formal version) and Algorithm 1 (the informal version).

---

1: **procedure** FASTMULTREG($A \in \mathbb{R}^{m \times n}, B \in \mathbb{R}^{m \times k}, W \in \mathbb{R}^{m \times n}, m, n, k, \epsilon_0, \delta_0$)  ▷ Lemma B.7
2:   ▷ $A_i$ is the $i$-th column of $A$
3:   ▷ $W_i$ is the $i$-th column of $W$
4:   ▷ $D_{W_i}$ is a diagonal matrix where we put $W_i$ on diagonal, other locations are zero
5:   $\epsilon_1 \leftarrow \epsilon_0 / \operatorname{poly}(n, \kappa)$
6:   $\delta_1 \leftarrow \delta_0 / \operatorname{poly}(n)$
7:   **for** $i = 1 \to n$ **do**
8:     $X_i \leftarrow$ HIGHPRECISIONREG($D_{W_i} B \in \mathbb{R}^{m \times k}, D_{W_i} A_i \in \mathbb{R}^k, m, k, \epsilon_1, \delta_1$)  ▷ Algorithm 2
9:   **end for**
10:   **return** $X$  ▷ $X \in \mathbb{R}^{n \times k}$
11: **end procedure**

---

**Lemma B.7.** *For any accuracy parameter $\epsilon_0 \in (0, 0.1)$, and failure probability $\delta_0 \in (0, 0.1)$. Assume that $m \leq n$. Given matrix $M \in \mathbb{R}^{m \times n}$, $Y \in \mathbb{R}^{m \times k}$ and $W \in \{0, 1\}^{m \times n}$, let $\kappa_0 := \sigma_{\max}(M) / \sigma_{\min}(Y)$ and $X = \arg\min_{X \in \mathbb{R}^{n \times k}} \|W \circ (M - YX^\top)\|_F$. There exists an algorithm (Algorithm 3) that takes time*

$$O((\|W\|_0 k \log n + nk^3 \log^2(n/\delta_0)) \cdot \log(n\kappa_0/\epsilon_0))$$

*and returns a matrix $Z \in \mathbb{R}^{n \times k}$ such that*

- $\|Z - X\| \leq \epsilon_0$

*holds with probability $1 - \delta_0$.*

*Proof.* We can rewrite multiple regression into $n$ linear regression in the following way,

$$\min_{X \in \mathbb{R}^{n \times k}} \|M - XY^\top\|_W^2 = \sum_{i=1}^n \min_{X_{i,:} \in \mathbb{R}^k} \|D_{\sqrt{W_i}} Y X_{i,:} - D_{\sqrt{W_i}} M_{i,:}\|_2^2.$$

Consider the $i$-th linear regression. We define

- $A := D_{\sqrt{W_i}} Y \in \mathbb{R}^{m \times k}$.

- $x := X_{i,:} \in \mathbb{R}^k$.

- $b := D_{\sqrt{W_i}} M_{:,i} \in \mathbb{R}^m$.

Recall that $Y \in \mathbb{R}^{m \times k}$.

For each $i \in [n]$, let $X_{i,:} \in \mathbb{R}^k$ denote $i$-th row of $X \in \mathbb{R}^{n \times k}$. For each $i \in [n]$, let $M_{:,i} \in \mathbb{R}^m$ denote $i$-th column of $M \in \mathbb{R}^{m \times n}$. We define $\mathrm{OPT}_i$

$$\mathrm{OPT}_i := \min_{X_{i,:} \in \mathbb{R}^k} \| \underbrace{D_{\sqrt{W_i}}}_{m \times m} \underbrace{Y}_{m \times k} \underbrace{X_{i,:}}_{k} - \underbrace{D_{\sqrt{W_i}}}_{m \times m} \underbrace{M_{:,i}}_{m} \|_2.$$

Here $D_{\sqrt{W_i}} \in \mathbb{R}^{m \times m}$ is a diagonal matrix where diagonal entries are from vector $\sqrt{W_i}$ (where $W_i$ is the $i$-th column of $W \in \{0,1\}^{m \times n}$. $\sqrt{W_i}$ is entry-wise sqaure root of $W_i$.).

The trivial solution of regression is choosing $X_{i,:} \in \mathbb{R}^k$ to be an all zero vector (length-$k$). In this case $\mathrm{OPT}_i \leq \|M_{:,i}\|_2$.

Let $C > 1$ be a sufficiently large constant (The running time of the algorithm is linear in $C$). Using backward error Lemma B.5, we know that

$$
\begin{aligned}
\|z - x\|_2 &\leq \epsilon_0 \cdot \frac{1}{(n\kappa_0)^C} \cdot \frac{\|M_{:,i}\|_2}{\sigma_{\min}(Y)} \\
&\leq \epsilon_0 \cdot \frac{1}{(n\kappa_0)^C} \cdot \frac{\sigma_{\max}(M)}{\sigma_{\min}(Y)} \\
&= \epsilon_0 \cdot \frac{1}{n^C \kappa_0^C} \cdot \kappa_0 \\
&\leq \epsilon_0,
\end{aligned}
\tag{22}
$$

where the second step follows from $\|M_{:,i}\|_2 \leq \sigma_{\max}(M)$, the third step follows from the definition of $\kappa_0$ (see Lemma B.7), and the last step follows from simple algebra.

Using Lemma B.6 $n$ times, we can obtain the running time. $\qquad\square$

## C   INITIALIZATION CONDITIONS

In this section, we deal with init conditions. In Section C.1, we provide the lemma which bound the distance between $U_\phi$ and $U_*$. In Section C.2, we show that if $U_\phi$ is close to $U_*$, then $U_0$ is close to $U_*$. In Section C.3, we give the formal definition of $U_0$, $U_\tau$, and $U_\phi$. In Section C.4, we combine the previous lemmas to form a new result.

### C.1   BOUNDING THE DISTANCE BETWEEN $U_\phi$ AND $U_*$

We prove our init condition lemma. This can be viewed as a variation of Lemma C.1 in Jain et al. (2013).

**Lemma C.1.** *Let $\frac{1}{p}P_{\Omega_0}(M) = U\Sigma V^\top$ be the SVD of the matrix $\frac{1}{p}P_{\Omega_0}(M)$, where $U \in \mathbb{R}^{m \times m}, \Sigma \in \mathbb{R}^{m \times n}$ and $V \in \mathbb{R}^{m \times n}$. Let $U_\phi \in \mathbb{R}^{m \times k}$ be the $k$ columns of $U$ corresponding to the top-$k$ left singular vectors of $U$. We can show that*

$$\mathrm{dist}(U_\phi, U_*) \leq \frac{1}{10^4 k}$$

*holds with probability at least $1 - 1/\mathrm{poly}(n)$.*

*Proof.* Let $M_\phi := U_\phi \Sigma V^\top$. Let $C > 1$ be some constant decided by Theorem 3.1 Keshavan et al. (2009).

We can show

$$
\begin{aligned}
\|M - M_\phi\| &\leq C \cdot \frac{k^{1/2}}{p^{1/2}(mn)^{1/4}} \cdot \|M\|_F \\
&\leq C \cdot \frac{k^{1/2}}{p^{1/2}(mn)^{1/4}} \cdot \sqrt{k}\sigma_1^* \\
&= C \cdot \frac{k}{p^{1/2}(mn)^{1/4}} \cdot \sigma_1^* \\
&\leq C \cdot \frac{k}{p^{1/2}m^{1/2}} \cdot \sigma_1^* \\
&\leq \frac{1}{10^4 k} \cdot \sigma_k^*,
\end{aligned}
\tag{23}
$$

where the first step follows from Theorem 3.1 in Keshavan et al. (2009), the second step follows from $\|M\|_F \leq \sqrt{k}\sigma_1^*$, the third step follows from simple algebra, namely $k^{\frac{1}{2}} \cdot k^{\frac{1}{2}} = k$, the fourth step follows from $n \geq m$, the last step follows from

$$p \geq 10^8 \cdot C^2 \cdot \frac{k^2}{m} \cdot \frac{(\sigma_1^*)^2}{(\sigma_k^*)^2}.$$

We also have

$$
\begin{aligned}
\|M - M_\phi\|^2 &= \|U_*\Sigma_* V_*^\top - U_\phi \Sigma V^\top\|^2 \\
&= \|U_*\Sigma_* V_*^\top - U_\phi U_\phi^\top U_*\Sigma_*(V_*)^\top + U_\phi U_\phi^\top U_*\Sigma_* V_*^\top - U_\phi \Sigma V^\top\|^2 \\
&= \|(I - U_\phi U_\phi^\top)U_*\Sigma_* V_*^\top + U_\phi(U_\phi^\top U_*\Sigma_* V_*^\top - \Sigma V^\top)\|^2 \\
&\geq \|(I - U_\phi U_\phi^\top)U_*\Sigma_* V_*^\top\|^2 \\
&= \|U_{\phi,\perp} U_{\phi,\perp}^\top U_*\Sigma_* V_*^\top\|^2 \\
&= \|U_{\phi,\perp}^\top U_*\Sigma_*\|^2 \\
&\geq \|U_{\phi,\perp}^\top U_*\| \cdot (\sigma_{\min}(\Sigma_*))^2 \\
&\geq (\sigma_k^*)^2 \|U_{\phi,\perp}^\top U_*\|^2,
\end{aligned}
\tag{24}
$$

where the first step follows from the SVD of $M$ (see Definition 3.5) and $M_k$, the second step follows from adding and subtracting the same thing, the third step follows from simple algebra, the fourth step follows from $\|A + B\| \geq \|A\|$ if $B^\top A = 0$ (Fact A.1), the fifth step follows from $U_\phi U_\phi^\top + U_{\phi,\perp} U_{\phi,\perp}^\top = I$, the sixth step follows from applying Part 2 of Fact A.3 twice (one for $U_{\phi,\perp}$ and one for $V_*$), the seventh step follows from $\|AB\| \geq \|A\| \cdot \sigma_{\min}(B)$ (Fact A.1), and the last step follows from $\sigma_{\min}(\Sigma_*) = \sigma_k^*$.

Combining Eq. (23) and Eq. (24), we get:

$$
\begin{aligned}
\|U_{\phi,\perp}^\top U_*\|_2 &\leq \frac{1}{\sigma_k^*} \cdot \|M - M_\phi\| \\
&\leq \frac{1}{10^4 k}.
\end{aligned}
$$

Thus, we complete the proof. □

## C.2 Closeness Between $U_\phi$ and $U_*$ To Closeness Between $U_0$ and $U_*$

We first define $\tau$, and then provide a lemma to show that whenever $U_\phi$ is close to $U_*$, $U_0$ is close to $U_*$.

**Definition C.2.** *We define $\tau$ as follows:*

$$\tau := 2\frac{\sqrt{k}}{\sqrt{n}} \cdot \mu.$$

The initialization condition proof is very standard in the literature, we follow from Jain et al. (2013).

**Lemma C.3.** *Let $U_* \in \mathbb{R}^{m \times k}$ is incoherent with parameter $\mu$.*

*We define*

$$\epsilon_\phi := \frac{1}{10^4 k}$$

*Let $U_\phi$ be an orthonormal column matrix such that*

$$\mathrm{dist}(U_\phi, U_*) \leq \epsilon_\phi.$$

*Let $U_\tau$ be obtained from $U_\phi \in \mathbb{R}^{n \times k}$ by setting all rows with norm greater than $\tau$ to zero. Let $U_0$ be an orthonormal basis of $U_\tau$.*

*Then, we have*

- *Part 1.* $\mathrm{dist}(U_0, U_*) \leq 1/2$

- *Part 2.* $U_0$ *is incoherent with parameter* $4\mu\sqrt{k}$.

**Remark C.4.** *For the convenience of matching later induction proof, when we use this lemma for the base in induction, we write $\widehat{U}_0$ to denote $U_\tau$.*

*Proof.* **Proof of Part 1.** For simplicity, in the proof, we use $\epsilon$ to denote $\epsilon_\phi$.

Since $\mathrm{dist}(U_\phi, U_*) \leq \epsilon$, we have that for every $i$, $\exists z_i \in \mathrm{span}(U_*)$, $\|z_i\|_2 = 1$ such that

$$\langle u_i, z_i \rangle \geq \sqrt{1 - \epsilon^2} \tag{25}$$

Also, since $z_i \in \mathrm{span}(U_*)$, we have that $z_i$ is incoherent with parameter $\mu\sqrt{k}$:

$$\|z_i\|_2 = 1 \tag{26}$$

and

$$\|z_i\|_\infty \leq \frac{\mu\sqrt{k}}{\sqrt{m}}. \tag{27}$$

Let $u_{\tau,i} \in \mathbb{R}^m$ be the vector obtained by setting all the elements of $u_i \in \mathbb{R}^m$ with a magnitude greater than $\tau$ to zero.

For each $i \in [k]$, let $u_i \in \mathbb{R}^m$ denote the $i$-th column of $U_\phi$.

By definition of $U_\phi$, we know that

$$\|u_i\|_2 = 1. \tag{28}$$

We define vector $u_{\bar{\tau},i} \in \mathbb{R}^m$

$$u_{\bar{\tau},i} := u_i - u_{\tau,i}. \tag{29}$$

Now, for each $j \in [n]$, we have

$$|(u_i)_j| > \tau = \frac{2\mu\sqrt{k}}{\sqrt{m}},$$

where $j$ represents the $j$-th entry of $u_i \in \mathbb{R}^m$.

Then, for each $j \in [m]$ with $|(u_i)_j| > \tau$,

$$
\begin{aligned}
|(u_{\tau,i})_j - (z_i)_j| &= |0 - (z_i)_j| \\
&= |(z_i)_j| \\
&\leq \frac{\mu\sqrt{k}}{\sqrt{m}} \\
&\leq 2\frac{\mu\sqrt{k}}{\sqrt{m}} - \frac{\mu\sqrt{k}}{\sqrt{m}} \\
&\leq |(u_i)_j| - |(z_i)_j| \\
&\leq |(u_i)_j - (z_i)_j|,
\end{aligned}
\tag{30}
$$

where the first step follows from truncation at $\tau$, the second step follows from simple algebra, the third step follows from Eq. (27), the fourth step follows from simple algebra, the fifth step follows from the definition of $|(u_i)_j|$ and $|(z_i)_j|$, and the last step follows from the triangle inequality.

For each $j \in [m]$ with $|(u_i)_j| \leq \tau$, we know that

$$
|(u_{\tau,i})_j - (z_i)_j| = |(u_i)_j - (z_i)_j|.
\tag{31}
$$

where the first step follows from $(u_{\tau,i})_j = (u_i)_j$ in this case.

Combining Eq. (30) and Eq. (31), we know for all $j \in [m]$

$$
|(u_{\tau,i})_j - (z_i)_j| \leq |(u_i)_j - (z_i)_j|.
$$

Hence,

$$
\begin{aligned}
\|u_{\tau,i} - z_i\|_2^2 &\leq \|u_i - z_i\|_2^2 \\
&= \|u_i\|_2^2 + \|z_i\|_2^2 - 2\langle u_i, z_i\rangle \\
&\leq \|u_i\|_2^2 + \|z_i\|_2^2 - 2\sqrt{1 - \epsilon^2} \\
&= 2 - 2\sqrt{1 - \epsilon^2} \\
&\leq 2\epsilon^2,
\end{aligned}
\tag{32}
$$

where the first step follows from taking the summation of square of Eq. (30), and the second step follows from simple algebra, the third step follows from Eq. (25), the fourth step follows from $u_i$ and $z_i$ are unit vectors, and the last step follows Part 3 of Fact A.4.

This also implies the following:

$$
\begin{aligned}
\|u_{\tau,i}\|_2 &\geq \|z_i\|_2 - \|u_{\tau,i} - z_i\|_2 \\
&\geq \|z_i\|_2 - \sqrt{2}\epsilon \\
&= 1 - \sqrt{2}\epsilon,
\end{aligned}
\tag{33}
$$

where the first step follows from triangle inequality, the second step follows from Eq. (32), and the third step follows from Eq. (26).

We can show

$$
\begin{aligned}
\|u_{\bar{\tau},i}\|_2 &= \|u_i - u_{\tau,i}\|_2 \\
&\leq \|u_i\|_2 - \|u_{\tau,i}\|_2 \\
&\leq 1 - \|u_{\tau,i}\|_2 \\
&\leq 1 - (1 - \sqrt{2}\epsilon) \\
&\leq 2\epsilon,
\end{aligned}
\tag{34}
$$

where the first step follows from Eq. (29), the second step follows from triangle inequality, the third step follows from Eq. (28), the fourth step follows from Eq. (33), and the last step follows from simple algebra.

We can show that

$$
\|U_{\bar{\tau}}\|^2 \leq \|U_{\bar{\tau}}\|_F^2
$$

$$
= \sum_{i=1}^{k} \|u_{\overline{\tau},i}\|_2^2
$$
$$
\leq 4\epsilon^2 k, \tag{35}
$$

where the first step follows from $\|\cdot\| \leq \|\cdot\|_F$, the second step follows from taking the summation of square of Eq. (34).

Let

$$
U_\tau = U_0 \Lambda^{-1} \tag{36}
$$

be the QR decomposition.

Then, for any $u_{*,\perp} \in \text{span}(U_{*,\perp})$, we have:

$$
\|u_{*,\perp}^\top U_0\|_2 = \|u_{*,\perp}^\top U_\tau \Lambda\|_2
$$
$$
\leq \|u_{*,\perp}^\top U_\tau\|_2 \|\Lambda\|, \tag{37}
$$

where the first step follows from Eq. (36) and the second step follows from $\|Ax\|_2 \leq \|A\| \cdot \|x\|_2$.

For the first term in the above equation

$$
\|u_{*,\perp}^\top U_\tau\|_2 \leq \|u_{*,\perp}^\top U_\phi\|_2 + \|u_{*,\perp}^\top U_{\overline{\tau}}\|_2
$$
$$
\leq \epsilon + \|u_{*,\perp}^\top U_{\overline{\tau}}\|_2
$$
$$
\leq \epsilon + \|U_{\overline{\tau}}\|
$$
$$
\leq \epsilon + 2\sqrt{k}\epsilon
$$
$$
\leq 3\sqrt{k}\epsilon, \tag{38}
$$

where the first step follows from triangle inequality, the second step follows from $\|u_{*,\perp}^\top\|_2 \leq d$, the third step follows from the definition of $\|\cdot\|$, the fourth step follows from Eq. (35), and the last step follows from $k \geq 1$.

We now bound $\|\Lambda\|$ as follow:

$$
\|\Lambda\|^2 = \frac{1}{\sigma_{\min}(\Lambda^{-1})^2}
$$
$$
= \frac{1}{\sigma_{\min}(U_0 \Lambda^{-1})^2}
$$
$$
= \frac{1}{\sigma_{\min}(U_\tau)^2}
$$
$$
= \frac{1}{1 - \|U_{\overline{\tau}}\|^2}
$$
$$
\leq \frac{1}{1 - 2\sqrt{k}\epsilon}
$$
$$
\leq 2, \tag{39}
$$

where the first step follows from Fact A.1, the second step follows from the fact that $U_0$ forms an orthonormal basis (see Part 2 of Fact A.3), the third step follows from the QR decomposition of $U_\tau$ (Eq. (36)), the fourth step follows from $\cos^2\theta + \sin^2\theta = 1$ (see Lemma A.9), the fifth step follows from Eq. (35), and the last step follows from using the fact that $\epsilon < \frac{1}{100k}$.

Thus, we have:

$$
\|u_{*,\perp}^\top U_0\|_2 \leq \|u_{*,\perp}^\top U_\tau\|_2 \|\Lambda\|
$$
$$
\leq 3\sqrt{k}\epsilon \cdot \|\Lambda\|
$$
$$
\leq 3\sqrt{k}\epsilon \cdot 2
$$
$$
= 6\sqrt{k}\epsilon
$$

$$\leq \frac{1}{10},$$

where the first step follows from Eq. (37), the second step follows from Eq. (38), the third step follows from Eq. (39), the fourth step follows from simple algebra, and the last step follows from the fact that $\epsilon \leq \frac{1}{10^4 k}$.

This proves the first part of the lemma.

**Proof of Part 2.** The incoherence of $U_0$ (see incoherence in Definition 3.4) is

$$
\begin{aligned}
\frac{\sqrt{m}}{\sqrt{k}} \max_{i \in [m]} \|e_i^\top U_0\|_2 &\leq \frac{\sqrt{m}}{\sqrt{k}} \max_{i \in [m]} \|e_i^\top U_\tau \Lambda\|_2 \\
&\leq \frac{\sqrt{m}}{\sqrt{k}} \max_{i \in [m]} \|e_i^\top U_\tau\|_2 \|\Lambda\| \\
&\leq \frac{2\sqrt{m}}{\sqrt{k}} \max_{i \in [m]} \|e_i^\top U_\tau\|_2 \\
&\leq \frac{2\sqrt{m}}{\sqrt{k}} \cdot \tau \\
&= \frac{2\sqrt{m}}{\sqrt{k}} \cdot 2\mu\sqrt{k}/\sqrt{n} \\
&\leq 4\mu,
\end{aligned}
$$

where the first step follows from the definition of incoherence, the second step follows from $\|Ax\|_2 \leq \|A\| \cdot \|x\|_2$, the third step follows from Eq (39), the fourth step follows from $U_\tau$ is truncating at $\tau$, the fifth step follows from $\tau = 2\mu\sqrt{k}/\sqrt{n}$ (see Definition C.2), and the last step follows from $n \geq m$. $\qquad\square$

## C.3 Definitions for $U_0$, $U_\tau$, $U_\phi$

We provide the definitions for $U_0$, $U_\tau$, and $U_\phi$. We also include a procedure that clips rows with large norm then perform Gram-Schmidt.

**Definition C.5.** *Let $\frac{1}{p} P_{\Omega_0}(M) = U\Sigma V^\top$ be the SVD of the matrix $\frac{1}{p} P_{\Omega_0}(M)$, where $U \in \mathbb{R}^{m \times m}, \Sigma \in \mathbb{R}^{m \times n}$ and $V \in \mathbb{R}^{n \times n}$.*

*Let $\tau := \frac{2\mu\sqrt{k}}{\sqrt{n}}$.*

*Let $U_\phi \in \mathbb{R}^{m \times k}$ be the $k$ columns of $U$ corresponding to the top-$k$ left singular vectors of $U$.*

*We define $U_\tau \in \mathbb{R}^{m \times k}$ as follows:*

$$
U_{\tau,i,*} := \begin{cases} U_{\tau,i,*}, & \text{if } \|\widehat{U}_{i,*}\|_2 \leq \tau, \\ 0, & \text{otherwise.} \end{cases}
$$

*Finally, we define $U_0 \in \mathbb{R}^{m \times k}$ to be the matrix formed from performing Gram-Schmidt on $U_\tau$, i.e., columns of $U_0$ are orthonormal.*

*To make convenient of base case of induction proof, we call $U_\tau$ to be $\widehat{U}_0$.*

## C.4 Initialization Condition: Main Result

We provide a Lemma which bounds $\mathrm{dist}(U_0, U_*)$ and shows that $U_0$ is incoherent with parameter $4\mu\sqrt{k}$.

**Lemma C.6.** *Let $M \in \mathbb{R}^{m \times n}$ be a $(\mu, k)$-incoherent matrix (see Definition 3.5).*

*Let $\Omega \subset [m] \times [n]$ and $p \in (0, 1)$ be defined as Definition A.11.*

---

**Algorithm 4** Clipping and Gram-Schmidt

---

1: **procedure** INIT($U_\phi \in \mathbb{R}^{m \times k}$, $\mu \in (0, 1)$)
2:   $\tau \leftarrow \frac{2\mu\sqrt{k}}{\sqrt{n}}$
3:   **for** $i = 1 \rightarrow m$ **do**
4:     $U_{\tau,i,*} = \begin{cases} U_{\phi,i,*}, & \text{if } \|U_{\phi,i,*}\|_2 \leq \tau, \\ 0, & \text{otherwise.} \end{cases}$
5:   **end for**
6:   $(Q, R) \leftarrow \text{QR}(U_\tau)$
7:   $U_0 \leftarrow Q$
8:   **return** $U_0, U_\tau$
9: **end procedure**

---

*Let $U_0 \in \mathbb{R}^{m \times k}$ be defined as Definition C.5.*

*Then, we have the following properties*

- $\text{dist}(U_0, U_*) \leq \frac{1}{2}$ *and*

- $U_0$ *is incoherent with parameter $4\mu\sqrt{k}$.*

*holds with probability at least $1 - 1/\text{poly}(n)$.*

*Proof.* From Lemma C.1, we see that $U_0$ satisfy that

$$\text{dist}(U_0, U_*) \leq \frac{1}{100k}$$

Using Lemma C.3, we finish the proof. □

## D MAIN INDUCTION HYPOTHESIS

In this section, we provide our main induction hypothesis: we inductively bound $\text{dist}(\widehat{V}_{t+1}, V^*), \text{dist}(\widehat{U}_{t+1}, U^*)$ and the incoherence of $U_{t+1}, V_{t+1}$. We start by presenting our formal algorithm.

**Lemma D.1** (Induction hypothesis)**.** *Let $M = U_*\Sigma_*V_*^\top$ be defined as Definition 3.5. Define $\epsilon_d := 1/10$. For all $t \in [T]$, we have the following results.*

*Part 1. If $U_t$ is $\mu_2$ incoherent and $\text{dist}(\widehat{U}_t, U_*) \leq \frac{1}{4}\text{dist}(\widehat{V}_t, V_*) \leq \epsilon_d$, then we have*

- $\text{dist}(\widehat{V}_{t+1}, V_*) \leq \frac{1}{4}\text{dist}(\widehat{U}_t, U^*) \leq \epsilon_d$

*Part 2. If $U_t$ is $\mu_2$ incoherent and $\text{dist}(\widehat{U}_t, U_*) \leq \frac{1}{4}\text{dist}(\widehat{V}_t, V_*) \leq \epsilon_d$, then we have*

- $V_{t+1}$ *is $\mu_2$ incoherent*

*Part 3. If $V_{t+1}$ is $\mu_2$ incoherent and $\text{dist}(\widehat{V}_{t+1}, V_*) \leq \frac{1}{4}\text{dist}(\widehat{U}_t, U_*) \leq \epsilon_d$, then we have*

- $\text{dist}(\widehat{U}_{t+1}, U_*) \leq \frac{1}{4}\text{dist}(\widehat{V}_{t+1}, V_*) \leq \epsilon_d$

*Part 4. If $V_{t+1}$ is $\mu_2$ incoherent and $\text{dist}(\widehat{V}_{t+1}, V_*) \leq \frac{1}{4}\text{dist}(\widehat{U}_t, U_*) \leq \epsilon_d$, then we have*

- $U_{t+1}$ *is $\mu_2$ incoherent.*

*The above results hold with high probability.*

---

**Algorithm 5** Alternating minimization for matrix completion, formal version of Algorithm 1.

---

1: **procedure** FASTMATRIXCOMPLETION($\Omega \subset [m] \times [n], P_\Omega(M), k, \epsilon, \delta$)     ▷ Theorem 4.2
2:     Let $\epsilon \in (0, 1)$ denote the final accuracy of algorithm.
3:     Let $\delta \in (0, 1)$ denote the final failure probability.
4:                                              ▷ Partition $\Omega$ into $2T + 1$ subsets $\Omega_0, \cdots, \Omega_{2T}$
5:         ▷ Each element of $\Omega$ belonging to one of the $\Omega_t$ with equal probability (sampling with replacement)
6:     $U_\phi = \text{SVD}(\frac{1}{p} P_{\Omega_0}(M), k)$                          ▷ $U_\phi \in \mathbb{R}^{m \times k}$
7:     $U_0, \widehat{U}_0 \leftarrow \text{INIT}(U_\phi)$                                ▷ Algorithm 4
8:     $T \leftarrow \Theta(\log(1/\epsilon))$
9:     $\epsilon_0 \leftarrow \epsilon/\text{poly}(n, \kappa)$
10:     $\delta_0 \leftarrow \delta/\text{poly}(n, T)$
11:     **for** $t = 0, \cdots, T - 1$ **do**
12:         $\widehat{V}_{t+1} \leftarrow \text{FASTMULTREG}(M \in \mathbb{R}^{m \times n}, \widehat{U}_t \in \mathbb{R}^{m \times k}, \Omega_{2t+1}, m, n, k, \epsilon_0, \delta_0)$
13:                     ▷ Alg. 3 , Lem. B.7. Here $\Omega_{2t+1}$ can be viewed as $m \times n$ weight matrix.
14:         $\widehat{V}_{t+1} \leftarrow \text{FASTMULTREG}(M^\top \in \mathbb{R}^{n \times m}, \widehat{V}_{t+1} \in \mathbb{R}^{n \times k}, \Omega_{2t+2}^\top, n, m, k, \epsilon_0, \delta_0)$
15:                     ▷ Alg. 3 , Lem. B.7. Here $\Omega_{2t+2}^\top$ can be viewed as $n \times m$ weight matrix.
16:     **end for**
17:     $X \leftarrow \widehat{U}_T \widehat{V}_T^\top$
18:     **return** $X$                                    ▷ $X \in \mathbb{R}^{m \times n}$
19: **end procedure**

---

*Proof.* **Proof of Part 1.** To prove this part, we need to use Lemma F.1, and Lemma F.3. To use Lemma F.1, we need the condition that $U_t$ is $\mu_2$ incoherent. To use Lemma F.3, we need two conditions: one is that $U_t$ be a $\mu_2$-incoherent and the other is $\text{dist}(U_t, U_*) \leq 1/2$.

We can show that

$$\begin{aligned}
\text{dist}(V_*, V_{t+1}) &= \|V_{*,\perp}^\top V_{t+1}\| \\
&= \|V_{*,\perp}^\top (V_* \Sigma_* U_*^\top U_t + F) \cdot R_{t+1}^{-1}\| \\
&= \|V_{*,\perp}^\top V_* \Sigma_* U_*^\top U_t R_{t+1}^{-1} + V_{*,\perp}^\top F R_{t+1}^{-1}\| \\
&= \|V_{*,\perp}^\top F \cdot R_{t+1}^{-1}\| \\
&\leq \|F \cdot R_{t+1}^{-1}\| \\
&= \|F \Sigma_*^{-1} \Sigma_* R_{t+1}^{-1}\| \\
&\leq \|F \Sigma_*^{-1}\| \cdot \|\Sigma_* R_{t+1}^{-1}\|,
\end{aligned} \tag{40}$$

where the first step follows from the definition of distance (Definition 3.1), the second step follows from the definition of $V_{t+1}$ ($V_{t+1} = (V_* \Sigma_* U_*^\top U_t + F) \cdot R_{t+1}^{-1}$, see Definition E.6), the third step follows from simple algebra, the fourth step follows from $V_{*,\perp}^\top V_* = 0$, the fifth step follows from Fact A.3, the sixth step follows from $\Sigma_*^{-1} \Sigma_* = I$, and the seventh step follows from $\|A \cdot B\| \leq \|A\| \cdot \|B\|$ (Fact A.1).

By Eq. (40), we can upper bound

$$\begin{aligned}
\text{dist}(V_*, V_{t+1}) &\leq \|F \Sigma_*^{-1}\| \cdot \|\Sigma_* R_{t+1}^{-1}\| \\
&\leq 2\delta_{2k} \cdot k \, \text{dist}(U_t, U_*) \cdot \|\Sigma^* R_{t+1}^{-1}\| \\
&\leq 2\delta_{2k} \cdot k \, \text{dist}(U_t, U_*) \cdot 2\sigma_1^*/\sigma_k^* \\
&\leq \frac{1}{4} \text{dist}(U_t, U_*),
\end{aligned}$$

where the second step follows from Lemma F.1, the third step follows from Lemma F.3, the last step follows from Definition A.12.

**Proof of Part 2.** In this proof, we need to use Lemma F.3, Lemma F.4, Lemma F.5. To use Lemma F.3, we need the condition that $U_t$ is $\mu_2$ incoherent and the other is $\text{dist}(U_t, U_*) \leq 1/2$. To

use Lemma F.4, we need the condition that $U_t$ is $\mu_2$ incoherent. To use Lemma F.5, we need the condition that $U_t$ is $\mu_2$ incoherent.

For each $j \in [n]$, according to Definition E.6, we have

$$V_{t+1,j} = R_{t+1}^{-1}(D_j - B_j^{-1}(B_j D_j - C_j))\Sigma_* V_{*,j}.$$

Therefore,

$$\|V_{t+1,j}\|_2 \leq \frac{\sigma_1^*}{\sigma_{\min}(R_{t+1})} \cdot \|V_{*,j}\|_2 \cdot (\|D_j\| + \|B_j^{-1}(B_j D_j - C_j)\|) \tag{41}$$

For the first term of Eq. (41), we have

$$\frac{\sigma_1}{\sigma_{\min}(R_{t+1})} \leq 2\sigma_1^*/\sigma_k^* \tag{42}$$

where it follows from Part 2 of Lemma F.3.

For the second term of Eq. (41), we have

$$\|V_{*,j}\|_2 \leq \mu\sqrt{k}/\sqrt{n} \tag{43}$$

For the third term of Eq. (41), we have

$$
\begin{aligned}
\|D_j\| + \|B_j^{-1}(B_j D_j - C_j)\| &\leq \|D_j\| + \|B_j^{-1}\| \cdot \|B_j D_j - C_j\| \\
&\leq 1 + \|B_j^{-1}\| \cdot \|B_j D_j - C_j\| \\
&\leq 1 + 2 \cdot \|B_j D_j - C_j\| \\
&\leq 1 + 2 \cdot (\|B_j D_j\| + \|C_j\|) \\
&\leq 1 + 2 \cdot (2 + 2) \\
&\leq 10,
\end{aligned}
\tag{44}
$$

where the first step follows from $\|AB\| \leq \|A\| \cdot \|B\|$, the second step follows from $\|D_j\| \leq 1$, the third step follows from Lemma F.4, the fourth step follows from $\|A + B\| \leq \|A\| + \|B\|$, the fifth step follows from Lemma F.5 (for $\|C_j\| \leq 2$), and the last step follows from simple algebra.

Combining Eq. (41), Eq. (42), Eq. (43), and Eq. (44), we have

$$
\begin{aligned}
\|V_{t+1,j}\|_2 &\leq 2\frac{\sigma_1^*}{\sigma_k^*} \cdot \mu\frac{\sqrt{k}}{\sqrt{n}} \cdot 10 \\
&\leq \mu_2 \cdot \frac{\sqrt{k}}{\sqrt{n}},
\end{aligned}
$$

where the second step follows from Definition E.2

**Proof of Part 3 and Part 4.** By a symmetric argument, we can also prove

$$\text{dist}(U_{t+1}, U_*) \leq \frac{1}{4}\text{dist}(V_{t+1}, V_*).$$

and $U_{t+1}$ is $\mu_2$ incoherent. □

We are now ready to prove the main theorem of this paper (Theorem 4.2)

*Proof of Theorem 4.2.* For the correctness part, it follows from combining the Init condition (Lemma C.6), perturbation lemma (Lemma G.5) Induction lemmas (Lemma D.1).

For the running time part, it follows from using Lemma B.7 for $2T$ times. For each iteration, we use twice, and there are $T$ iterations. □

## E  GENERAL CASE UPDATE RULES AND NOTATIONS

In this section, we review various properties of rank-$k$ updates due to our algorithm. In Section E.1, we provide the formal definition of the updated rule. In Section E.2, we give the definition of the error matrix and analyze its properties. In Section E.3, we focus on providing the definition for the update rule for $V$.

### E.1 UPDATE RULE

To begin with, we formally define the updated rule as follows.

**Definition E.1** (Update rule). *We define several updated rules here:*

- *Part 1. We define the QR factorization $\widehat{U}_t \in \mathbb{R}^{m \times k}$ as $\widehat{U}_t := U_t R_{t,U}$.*

  - *Here $U_t \in \mathbb{R}^{m \times k}$ and $R_{t,U} \in \mathbb{R}^{k \times k}$*

- *Part 2. We define $\widehat{V}_{t+1} \in \mathbb{R}^{n \times k}$ as follows $\widehat{V}_{t+1} := \arg\min_{\widehat{V} \in \mathbb{R}^{n \times k}} \|P_\Omega(U_t \widehat{V}^\top) - P_\Omega(M)\|_F^2$.*

- *Part 3. We define the QR decomposition of $\widehat{V}_{t+1} \in \mathbb{R}^{n \times k}$ as follows $\widehat{V}_{t+1} := V_{t+1} R_{t+1,V}$.*

  - *Here $V_{t+1} \in \mathbb{R}^{n \times k}$ and $R_{t+1,V} \in \mathbb{R}^{k \times k}$*

- *Part 4. We define $\widehat{U}_{t+1} \in \mathbb{R}^{m \times k}$ as follows $\widehat{U}_{t+1} := \arg\min_{\widehat{U} \in \mathbb{R}^{m \times k}} \|P_\Omega(\widehat{U} V_{t+1}^\top) - P_\Omega(M)\|_F^2$.*

We note that the update rule does not directly reflect our algorithm. Nevertheless, we start by analyzing this QR-based update and later connecting it to our algorithm.

**Definition E.2** ($\mu_2$). *We define $\mu_2$ as follows*

$$\mu_2 := 40 \cdot \frac{\sigma_1^*}{\sigma_k^*} \cdot \sqrt{k} \cdot \mu.$$

### E.2 ERROR MATRIX

Next, we provide several definitions related to the error matrix $F$. For most of the notations, we follow Jain et al. (2013) (Note that those definitions are implicitly presented on page 16 in Jain et al. (2013)). Note that we use $U_*, V_*$ to denote the optimal low rank factor instead of $U^*, V^*$ to ease notations.

**Definition E.3** (Error matrix). *For a given $V_* \in \mathbb{R}^{n \times k}$ matrix, for each $j \in [n]$, we use $v_{*,i}$ to denote the $i$-th row of $V_*$.*

*We define $v_* \in \mathbb{R}^{nk}$ as follows*

$$v_* := \mathrm{vec}(V_*),$$

*which can be written as*

$$v_* = [v_{*,1}^\top, v_{*,2}^\top, \ldots, v_{*,k}^\top]^\top.$$

*We use $u_{t,i}$ to denote the $i$-th row of $U_t \in \mathbb{R}^{m \times k}$.*

*We use $u_{*,i}$ to denote the $i$-th row of $U_* \in \mathbb{R}^{m \times k}$.*

- *We define $B \in \mathbb{R}^{nk \times nk}$ to be the matrix where the $j$-th block matrix is $B_j$, i.e.,*

$$B := \begin{bmatrix} B_1 & & & \\ & B_2 & & \\ & & \ddots & \\ & & & B_n \end{bmatrix}$$

  *For each $j \in [n]$, we define $B_j \in \mathbb{R}^{k \times k}$ as follows*

$$B_j := \frac{1}{p} \sum_{i \in \Omega_{*,j}} u_{t,i} u_{t,i}^\top.$$

- *We define matrix $C \in \mathbb{R}^{nk \times nk}$ to be a diagonal block matrix where the the $j$-th diagonal block is $C_j \in \mathbb{R}^{k \times k}$*

$$C_j := \frac{1}{p} \sum_{i \in \Omega_{*,j}} u_{t,i} u_{*,i}^\top$$

- *We define matrix $D \in \mathbb{R}^{nk \times nk}$ to be the matrix where the $j$-th diagonal block is $D_j \in \mathbb{R}^{k \times k}$*

$$D_j := U_t^\top U_*$$

- *We define matrix $S \in \mathbb{R}^{nk \times nk}$ as*

$$S = \begin{bmatrix} \sigma_1^* I_n & & & \\ & \sigma_2^* I_n & & \\ & & \ddots & \\ & & & \sigma_k^* I_n \end{bmatrix}$$

*For each $i \in [k]$, we define $F_i \in \mathbb{R}^n$ as follows*

$$F_i := \begin{bmatrix} (B^{-1}(BD - C)Sv_*)_{n(i-1)+1} \\ (B^{-1}(BD - C)Sv_*)_{n(i-1)+2} \\ \vdots \\ (B^{-1}(BD - C)Sv_*)_{n(i-1)+n} \end{bmatrix}$$

*We define $F \in \mathbb{R}^{n \times k}$ as follows:*

$$F := \begin{bmatrix} F_1 & F_2 & \cdots & F_k. \end{bmatrix}$$

**Claim E.4.** *Let $B, D, C$ be diagonal block matrices (where each block has size $k \times k$).*

*Let $S$ be block rescaled identity matrix (where each block has size $n \times n$).*

*Then, we have*

$$SB^{-1}(BD - C) = B^{-1}(BD - C)S.$$

*Proof.* Without loss of generality, we can assume that $n/k$ is an integer. Then, $S$ can also be viewed as a diagonal block matrix with size $k \times k$.

The $j$-th block of $SB^{-1}(BD - C)$ is

$$\begin{aligned} (SB^{-1}(BD - C))_j &= S_j B_j^{-1}(B_j D_j - C_j) \\ &= B_j^{-1}(B_j D_j - C_j) \cdot S_j \\ &= (B^{-1}(BD - C)S)_j, \end{aligned}$$

where the first step follows from all matrices are diagonal block matrix, the second step follows from $S_j$ is an identiy matrix with resacling ($AI = IA$), and the third step follows from all matrices ar diagonal block matrix.

Thus, we complete the proof. $\square$

**Claim E.5.** *We have the following identity:*

$$\|F\Sigma_*^{-1}\|_F^2 = \|B^{-1}(BD - C)v_*\|_2^2$$

*Proof.* We have

$$\|F\Sigma_*^{-1}\|_F^2 = \sum_{j=1}^n \sum_{i=1}^k (B^{-1}(BD - C)Sv_*)_{n(i-1)+j}^2 (\sigma_i^*)^{-2}$$

$$= \sum_{j=1}^{n} \sum_{i=1}^{k} (SB^{-1}(BD-C)v_*)^2_{n(i-1)+j} (\sigma_i^*)^{-2}$$

$$= \sum_{j=1}^{n} \sum_{i=1}^{k} S^2_{n(i-1)+j, n(i-1)+j} (B^{-1}(BD-C)v_*)^2_{n(i-1)+j} (\sigma_i^*)^{-2}$$

$$= \sum_{j=1}^{n} \sum_{i=1}^{k} (\sigma_i^*)^2 (B^{-1}(BD-C)v_*)^2_{n(i-1)+j} (\sigma_i^*)^{-2}$$

$$= \sum_{j=1}^{n} \sum_{i=1}^{k} (B^{-1}(BD-C)v_*)^2_{n(i-1)+j}$$

$$= \|B^{-1}(BD-C)v_*\|_2^2,$$

where the first step follows from the definition of $\| \cdot \|_F^2$ the second step follows from Claim E.4, the third step follows from that fact that $S$ is a diagonal matrix, the fourth step follows from $S_{n(i-1)+j, n(i-1)+j} = \sigma_i^*$, the fifth step follows from $(\sigma_i^*)^2$ and $(\sigma_i^*)^{-2}$ canceling out, and the last step follows from the definition of $\| \cdot \|_2^2$. $\qquad\square$

### E.3   UPDATE RULE FOR $V$

Now, we define the update rule for $V$.

**Definition E.6.** *We provide the definition of $\widehat{V}_{t+1} \in \mathbb{R}^{n \times k}$ and $V_{t+1} \in \mathbb{R}^{n \times k}$.*

- *We define $\widehat{V}_{t+1} \in \mathbb{R}^{n \times k}$ as follows*

$$\widehat{V}_{t+1} := V_* \Sigma_* U_*^\top U_t - F$$

  *Note that the first term can be treated as power-method update, and the second term is the error term. Here $F \in \mathbb{R}^{n \times k}$ is the error matrix defined in Definition E.3.*

- *We define $V_{t+1} \in \mathbb{R}^{n \times k}$ as follows*

$$V_{t+1} := \widehat{V}_{t+1} R_{t+1}^{-1}.$$

  *Here $R_{t+1} \in \mathbb{R}^{k \times k}$ is a upper-triangular matrix obtained using QR-decomposition of $\widehat{V}_{t+1} \in \mathbb{R}^{n \times k}$.*

*The above two definitions imply that*

$$V_{t+1} = (V_* \Sigma_* U_*^\top U_t - F) R_{t+1}^{-1}.$$

## F   TECHNICAL LEMMAS FOR DISTANCE SHRINKAGE

In this section, we prove a collection of technical lemmas that will facilitate us to eventually conclude our induction.

In Section F.1, we bound $\|F\Sigma_*^{-1}\|$ by distance. In Section F.2, we bound $\|(BD-C)v_*\|_2$ by distance. In Section F.3, we upper bound $\|\Sigma^* R_{t+1}^{-1}\|$. In Section F.4, we upper bound $\|B^{-1}\|$. In Section F.5, we upper bound $\|C_j\|$.

### F.1   UPPER BOUNDING $\|F\Sigma_*^{-1}\|$ BY DISTANCE

In this section, we upper bound $\|F\Sigma_*^{-1}\|$ by distance between $U_t$ and $U_*$. We generally follow the approach of Jain et al. (2013).

**Lemma F.1.** *Let $F$ be the error matrix defined by Definition E.6. Let $U_t$ be a $\mu_2$-incoherent matrix and $\mathrm{dist}(U_t, U_*) \le 1/2$. Let $M$ be a $(\mu, k)$-incoherent matrix (see Definition 3.5). Let $\Omega \subset [m] \times [n]$ and $p \in (0, 1)$ be defined as Definition A.11. Let $\delta_{2k} \in (0, 0.1)$ be defined as Definition A.12.*

*Then, we have*

$$\|F(\Sigma_*)^{-1}\| \leq 2\delta_{2k} \cdot k \cdot \mathrm{dist}(U_t, U_*)$$

*holds with probability least $1 - 1/\mathrm{poly}(n)$.*

*Proof.* We have

$$
\begin{aligned}
\|F\Sigma_*^{-1}\| &\leq \|F\Sigma_*^{-1}\|_F \\
&= \|B^{-1}(BD - C)v_*\|_2 \\
&\leq \|B^{-1}\| \cdot \|(BD - C)v_*\|_2 \\
&\leq 2 \cdot \|(BD - C)v_*\|_2 \\
&\leq 2 \cdot \delta_{2k} \cdot \mathrm{dist}(U_t, U_*),
\end{aligned}
$$

where the first step follows from $\|\cdot\| \leq \|\cdot\|_F$ (Fact A.1), the second step follows from Claim E.5, the third step follows from $\|Ax\|_2 \leq \|A\| \cdot \|x\|_2$, the fourth step follows from Lemma F.4, and the fifth step follows from Lemma F.2. $\square$

## F.2    UPPER BOUNDING $\|(BD - C)v_*\|_2$ BY DISTANCE

Now, we upper bound $\|(BD - C)v_*\|_2$. We follow similar ideas in literature Jain et al. (2013).

**Lemma F.2.** *Let $M \in \mathbb{R}^{m \times n}$ be a $(\mu, k)$-incoherent matrix (see Definition 3.5). Let $\Omega \subset [m] \times [n]$ and $p \in (0, 1)$ be defined as Definition A.11. Let $U_t \in \mathbb{R}^{m \times k}$ be $\mu_2$ incoherent matrix. For each $j \in [n]$, let $v_{*,j}$ denote the $j$-th row of $V_* \in \mathbb{R}^{n \times k}$.*

*We define $v_* \in \mathbb{R}^{n \times k}$ as follows*

$$v_* = [v_{*,1} \quad \cdots v_{*,n}]$$

*Then, we have:*

$$\|(BD - C)v^*\|_2 \leq \delta_{2k} \cdot \mathrm{dist}(V_{t+1}, V^*),$$

*holds with probability at least $1 - 1/\mathrm{poly}(n)$.*

*Proof.* Let $X \in \mathbb{R}^{n \times k}$ and $x = \mathrm{vec}(X) \in \mathbb{R}^{nk}$, where $\|x\|_2 = 1$. For each $j \in [n]$, let $x_j \in \mathbb{R}^k$ be the $j$-th row of $X \in \mathbb{R}^{n \times k}$. For each $j \in [n]$, we define $H_j \in \mathbb{R}^{k \times k}$ as

$$H_j := (B_j D_j - C_j).$$

Then, by the definition of $B_j$, $C_j$, and $D_j$ (see Definition E.3), we can write $H_j \in \mathbb{R}^{k \times k}$ as

$$H_j = \frac{1}{p} \sum_{i \in \Omega_{*,j}} u_{t,i} u_{t,i}^\top U_t^\top U_* - u_{t,i} u_{*,i}^\top = \frac{1}{p} \sum_{i \in \Omega_{*,j}} H_{j,i}.$$

For $j \in [n], i \in [m]$, we define $H_{j,i} \in \mathbb{R}^{k \times k}$ as follows

$$H_{j,i} := u_{t,i} u_{t,i}^\top U_t^\top U_* - u_{t,i} u_{*,i}^\top \tag{45}$$

Then, we have

$$H_j = \frac{1}{p} \sum_{i \in \Omega_{*,j}} H_{j,i}$$

Note that,

$$\sum_{i=1}^n H_{j,i} = U_t^\top U_t U_t^\top U^* - U_t^\top U^* = 0. \tag{46}$$

For each $j \in [n]$, let $V_{*,j}$ denote the $j$-th row of $V_* \in \mathbb{R}^{n \times k}$.

Now,

$$
\begin{aligned}
x^\top (BD - C) v_* &= \sum_{j=1}^n x_j^\top (B_j D - C_j) V_{*,j} \\
&= \frac{1}{p} \sum_{p=1}^k \sum_{q=1}^k \sum_{(i,j) \in \Omega} (x_j)_p \cdot (V_{*,j})_q \cdot (H_{j,i})_{p,q},
\end{aligned}
$$

where the first step follows from $j$ being defined as the $j$-th row of matrices and the second step follows from the definition of $B_j, C_j, D$ (see Definition E.3).

Also, using Eq. (46), we have $\forall (p,q) \in [k] \times [k]$:

$$
\sum_{i=1}^n (H_{j,i})_{p,q} = 0.
$$

Hence, we get with probability at least $1 - \frac{1}{n^3}$:

$$
\begin{aligned}
x^\top (BD - C) v_* &= \sum_{j=1}^n x_j^\top (B_j D - C_j) V_{*,j} \\
&\leq \frac{1}{p} \sum_{p=1}^k \sum_{q=1}^k (\sum_{j=1}^n (x_j)_p^2 \cdot (V_{*,j})_q^2)^{1/2} \cdot (\sum_{i=1}^m (H_{j,i})_{p,q}^2)^{1/2},
\end{aligned} \tag{47}
$$

where the first step follows from $j$ being defined as the $j$-th row of matrices and the second step follows from Lemma A.14.

For each $q \in [k]$, we use $U_{*,q}$ to denote the $q$-th column of $U_* \in \mathbb{R}^{m \times k}$.

Also,

$$
\begin{aligned}
\sum_{i=1}^m (H_{j,i})_{p,q}^2 &= \sum_{i=1}^m (u_{t,i})_p^2 \cdot (u_{t,i}^\top U_t^\top U_{*,q} - (U_*)_{i,q})^2 \\
&\leq \max_{i \in [m]} (u_{t,i})_p^2 \cdot \sum_{i=1}^m (u_{t,i}^\top U_t^\top U_{*,q} - (U_*)_{i,q})^2 \\
&= \max_{i \in [m]} (u_{t,i})_p^2 \cdot \|(U_t U_t^\top - I) U_{*,q}\|_2^2 \\
&\leq \frac{\mu_2^2 k}{m} \cdot \|(U_t U_t^\top - I) U_{*,q}\|_2^2 \\
&\leq \frac{\mu_2^2 k}{m} \cdot \|(U_t U_t^\top - I) U_*\|^2 \\
&\leq \frac{\mu_2^2 k}{m} \cdot \mathrm{dist}(U_t, U_*)^2,
\end{aligned} \tag{48}
$$

where the first step follows from Eq. (45), the second step follows from $\sum_i a_i b_i \leq \max_i a_i \sum_i b_i$, the third step follows from simple algebra, the fourth step follows from $u_t$ is $\mu_2$ incoherent, the fifth step follows from spectral norm definition, the last step follows from the definition of distance.

Using Eq. (47), Eq. (48), and incoherence of $V_* \in \mathbb{R}^{n \times k}$, we get (w.p. $1 - 1/n^3$),

$$
\begin{aligned}
\max_{x: \|x\|_2 = 1} x^\top (BD - C) v^* &\leq \sum_{p=1}^k \sum_{q=1}^k \frac{\mu_2^2 k}{mp} \mathrm{dist}(U_t, U_*) \|x_p\|_2 \\
&\leq \delta_{2k} \mathrm{dist}(U_t, U_*),
\end{aligned} \tag{49}
$$

where the last step follows from $\sum_{p=1}^k \|x_p\|_2 \leq \sqrt{k} \|x\|_2 = \sqrt{k}$.

Finally, we have

$$
\begin{aligned}
\|(BD - C) v^*\|_2 &= \max_{x, \|x\| = 1} x^\top (BD - C) v^* \\
&\leq \delta_{2k} \mathrm{dist}(U_t, U_*),
\end{aligned}
$$

where the first step follows from Fact A.1 and the second step follows from Eq. (49). $\qquad \square$

## F.3    UPPER BOUNDING $\|\Sigma^* R_{t+1}^{-1}\|$ BY CONDITION NUMBER

To bound $\|\Sigma^* R_{t+1}^{-1}\|$, we show that $\|\Sigma^* R_{t+1}^{-1}\| \leq 2 \cdot \sigma_1^*/\sigma_k^*$. Similarly, we also bound $\frac{\sigma_1^*}{\sigma_{\min}(R_{t+1})}$ by showing $\frac{\sigma_1^*}{\sigma_{\min}(R_{t+1})} \leq 2 \cdot \sigma_1^*/\sigma_k^*$.

**Lemma F.3.** *Let $R_{t+1}$ be the lower-triangular matrix obtained by QR decomposition of $\widehat{V}_{t+1}$ (see Definition E.6). Let $U_t$ be a $\mu_2$-incoherent and $\mathrm{dist}(U_t, U_*) \leq 1/2$. Let $M$ be a $(\mu, k)$-incoherent matrix (see Definition 3.5). Let $\Omega$ be defined as Definition A.11. Let $\delta_{2k} \in (0, 0.01)$ be defined as Definition A.12.*

*Then, we have*

- *Part 1.*

$$\|\Sigma^* R_{t+1}^{-1}\| \leq 2 \cdot \sigma_1^*/\sigma_k^*.$$

- *Part 2.*

$$\frac{\sigma_1^*}{\sigma_{\min}(R_{t+1})} \leq 2 \cdot \sigma_1^*/\sigma_k^*$$

*Proof.* Note that

$$\|\Sigma^* R_{t+1}^{-1}\| \leq \frac{\sigma_1^*}{\sigma_{\min}(R_{t+1})}$$

We can show that

$$
\begin{aligned}
\|F\| &= \|F\Sigma_*^{-1}\Sigma_*\| \\
&\leq \|F\Sigma_*^{-1}\| \cdot \|\Sigma_*\| \\
&\leq \|F\Sigma_*^{-1}\| \cdot \sigma_1^*,
\end{aligned}
\tag{50}
$$

where the first step follows from $\Sigma_*^{-1}\Sigma_* = I$, the second step follows from $\|AB\| \leq \|A\| \cdot \|B\|$, and the third step follows from $\|\Sigma_*\| \leq \sigma_1^*$.

We can show that

$$
\begin{aligned}
\min_{z:\|z\|_2=1} \|V_*\Sigma_* U_*^\top U_t z\|_2^2 &= \min_{z:\|z\|_2=1} \|\Sigma_* U_*^\top U_t z\|_2^2 \\
&= \sigma_{\min}^2(\Sigma_* U_*^\top U_t z) \\
&\geq \sigma_{\min}^2(\Sigma_*) \cdot \sigma_{\min}^2(U_*^\top U_t) \\
&= (\sigma_k^*)^2 \cdot \sigma_{\min}^2(U_*^\top U_t) \\
&\geq (\sigma_k^*)^2 \cdot (1 - \|U_{*,\perp}^\top U_t\|^2),
\end{aligned}
$$

where the first step follows from Part 2 of Fact A.3 that $V$ is a matrix of SVD that has an orthonormal basis, the second step follows from definition of $\sigma_{\min}$, the third step follows from Fact A.1, the fourth step follows from $\sigma_{\min}(\Sigma_*) = \sigma_k^*$, and the last step follows from $\cos^2\theta + \sin^2\theta = 1$ (see Lemma A.9).

Now, we have

$$
\begin{aligned}
\sigma_{\min}(R_{t+1}) &= \min_{z:\|z\|_2=1} \|R_{t+1}z\|_2 \\
&= \min_{z:\|z\|_2=1} \|V_{t+1}R_{t+1}z\|_2 \\
&= \min_{z:\|z\|_2=1} \|V_*\Sigma_* U_*^\top U_t z - Fz\|_2 \\
&\geq \min_{z:\|z\|_2=1} \|V_*\Sigma_* U_*^\top U_t z\|_2 - \|Fz\|_2 \\
&\geq \min_{z:\|z\|_2=1} \|V_*\Sigma_* U_*^\top U_t z\|_2 - \|F\|
\end{aligned}
$$

$$\geq \sigma_k^* \cdot \sqrt{1 - \|U_{*,\perp}^\top U_t\|_2^2} - \sigma_1^* \cdot \|F\Sigma_*^{-1}\|$$

$$= \sigma_k^* \cdot \sqrt{1 - \mathrm{dist}(U_*, U_t)^2} - \sigma_1^* \cdot \|F\Sigma_*^{-1}\|$$

$$= \sigma_k^* \cdot \sqrt{1 - \mathrm{dist}(U_*, U_t)^2} - 2\sigma_1^* \delta_{2k} k \, \mathrm{dist}(U_t, U_*),$$

where the first step follows from the definition of $\sigma_{\min}$, the second step follows from Part 2 of Fact A.3, the third step follows from Definition E.6, the fourth step follows from triangle inequality, the fifth step follows from the definition of $\|\cdot\|$, the sixth step follows from Eq. (50), the seventh step follows from the definition of $\mathrm{dist}(U_*, U_t)$ (see Definition 3.1), and the last step follows from Lemma F.1.

We can obtain that

$$\|\Sigma^* R_{t+1}^{-1}\| \leq \frac{\sigma_1^*/\sigma_k^*}{\sqrt{1 - \mathrm{dist}(U_t, U_*)^2} - 2 \cdot (\sigma_1^*/\sigma_k^*)\delta_{2k} k \, \mathrm{dist}(U_t, U_*)}.$$

For convenient, we define $x := \mathrm{dist}(U_t, U_*)$.

We define $y := 2(\sigma_1^*/\sigma_k^*)\delta_{2k} k$.

Then we now $x \in [0, 1/2]$ and $y \in (0, 0.1)$.

Then we obtain that

$$\|\Sigma_* R_{t+1}^{-1}\| \leq \frac{\sigma_1^*/\sigma_k^*}{\sqrt{1 - x^2} - y \cdot x}.$$

Using Part 1 of Fact A.4, we know that

$$\sqrt{1 - x^2} - y \cdot x \geq 1/2.$$

Thus, we have

$$\|\Sigma_* R_{t+1}^{-1}\| \leq 2\sigma_1^*/\sigma_k^*.$$

$\square$

## F.4 UPPER BOUNDING $\|B^{-1}\|$ BY CONSTANT

We analyze $\|B_j^{-1}\|$, for all $j \in [n]$, and $\|B^{-1}\|$ and show both of them are bounded by 2. We prove a variation of Lemma C.6 in Jain et al. (2013).

**Lemma F.4.** *Let $M$ be a $(\mu, k)$-incoherent matrix (see Definition 3.5). Let $\Omega, p$ be defined as Definition A.11. For each $j \in [n]$, we define $B_j \in \mathbb{R}^{k \times k}$ as follows*

$$B_j := \frac{1}{p} \sum_{i \in \Omega_{*,j}} U_{t,i} U_{t,i}^\top$$

*Let $U_t$ be $\mu_2$ incoherent. Then, we have:*

- *Part 1. For all $j \in [n]$,*

$$\|B_j^{-1}\| \leq 2$$

- *Part 2.*

$$\|B^{-1}\| \leq 2 \tag{51}$$

*both events succeed with a probability at least $1 - 1/\mathrm{poly}(n)$*

*Proof.* We have:

$$\|B^{-1}\|_2 = \frac{1}{\sigma_{\min}(B)}$$

$$= \frac{1}{\min_{x, \|x\|_2 = 1} \|Bx\|_2}$$

$$= \frac{1}{\min_{x, \|x\|_2 = 1} x^\top Bx},$$

where $x \in \mathbb{R}^{nk}$ and the first step follows from Fact A.1, the second step follows from definition of $\sigma_{\min}$, and the last step follows from $B$ is a symmetric matrix.

Let $x = \text{vec}(X)$, i.e., $x_p$ is the $p$-th column of $X$ and $x_j$ is the $j$-th row of $X$. Let $B_j \in \mathbb{R}^{k \times k}$. Let $x_j \in \mathbb{R}^k$. Now, $\forall x \in \mathbb{R}^{nk}$,

$$x^\top Bx = \sum_{j \in [n]} x_j^\top B_j x_j$$

$$\geq \min_{j \in [n]} \sigma_{\min}(B_j),$$

where the first step follows from $B$ is a diagonal block matrix, and the second step follows from the definition of $\sigma_{\min}$ for a symmetric matrix.

Results would follow using the bound on $\sigma_{\min}(B_j), \forall j \in [n]$ that we show below

**Lower bound on $\sigma_{\min}(B_j)$:**

Consider any $w \in \mathbb{R}^k$ such that $\|w\|_2 = 1$.

We have:

$$Z = w^\top B_j w$$

$$= w^\top (\frac{1}{p} \sum_{i \in \Omega_{*,j}} U_{t,i} U_{t,i}^\top) w$$

$$= \frac{1}{p} \sum_{i \in \Omega_{*,j}} \langle w, U_{t,i} \rangle^2$$

$$= \frac{1}{p} \sum_{i \in [m]} \delta_{i,j} \cdot \langle w, U_{t,i} \rangle^2,$$

where the first step follows from the definition of $Z$, the second step follows from the definition of $B$, the third step follows from the definition of inner product, and the last step follows from definition of $\delta_{i,j}$.

Note that,

$$\mathbb{E}[Z] = \mathbb{E}[\frac{1}{p} \sum_{i \in [m]} \delta_{i,j} \cdot \langle w, U_{t,i} \rangle^2]$$

$$= \frac{1}{p} \sum_{i \in [m]} \mathbb{E}[\delta_{i,j}] \cdot \langle w, U_{t,i} \rangle^2$$

$$= \frac{1}{p} \sum_{i \in [m]} p \cdot \langle w, U_{t,i} \rangle^2$$

$$= \sum_{i \in [m]} \langle w, U_{t,i} \rangle^2$$

$$= \sum_{i \in [m]} w^\top U_{t,i} U_{t,i}^\top w$$

$$= w^\top U_t^\top U_t w$$

$$= w^\top w$$

$$= 1, \tag{52}$$

where the first step follows from the definition of $Z$, the second step follows from $U$ being orthogonal, the third step follows from $\mathbb{E}[\delta_{i,j}] = p$, the fourth step follows from simple algebra, the fifth step

follows from the definition of the inner product, the sixth step follows from the fact that $U_t$ is a $m \times k$ matrix, the seventh step follows from $U_t U_t^\top = I_k$, and the last step follows from $\|w\|_2 = 1$.

For variance, we have

$$
\begin{aligned}
\mathbb{E}[Z^2] &= \mathbb{E}[(\frac{1}{p} \sum_{i \in [m]} \delta_{i,j} \cdot \langle w, U_{t,i} \rangle^2)^2] \\
&= \mathbb{E}[\frac{1}{p^2} (\sum_{i \in [m]} \delta_{i,j} \cdot \langle w, U_{t,i} \rangle^2)^2] \\
&= \frac{1}{p^2} \mathbb{E}[(\sum_{i \in [m]} \delta_{i,j} \cdot \langle w, U_{t,i} \rangle^2)^2] \\
&= \frac{1}{p^2} (\sum_{i \in [m]} \mathbb{E}[\delta_{i,j}^2](\langle w, U_{t,i} \rangle^2)^2 + \sum_{i_1 \neq i_2} \mathbb{E}[\delta_{i_1,j}] \langle w, U_{t,i_1} \rangle^2 \mathbb{E}[\delta_{i_2,j}] \langle w, U_{t,i_2} \rangle^2) \\
&= \frac{1}{p^2} (\sum_{i \in [m]} p \langle w, U_{t,i} \rangle^4 + \sum_{i_1 \neq i_2} p^2 \langle w, U_{t,i_1} \rangle^2 \langle w, U_{t,i_2} \rangle^2) \\
&= \frac{1}{p} (\sum_{i \in [m]} \langle w, U_{t,i} \rangle^4 - p \sum_{i \in [m]} \langle w, U_{t,i} \rangle^4 + p \sum_{i \in [m]} \langle w, U_{t,i} \rangle^4 + p \sum_{i_1 \neq i_2} \langle w, U_{t,i_1} \rangle^2 \langle w, U_{t,i_2} \rangle^2) \\
&= \frac{1}{p} \sum_{i \in [m]} \langle w, U_{t,i} \rangle^4 - \sum_{i \in [m]} \langle w, U_{t,i} \rangle^4 + \sum_{i \in [m]} \langle w, U_{t,i} \rangle^4 + \sum_{i_1 \neq i_2} \langle w, U_{t,i_1} \rangle^2 \langle w, U_{t,i_2} \rangle^2 \\
&= (\frac{1}{p} - 1) \sum_{i \in [m]} \langle w, U_{t,i} \rangle^4 + (\sum_{i \in [m]} \langle w, U_{t,i} \rangle^2)^2 \\
&= (\frac{1}{p} - 1) \sum_{i \in [m]} \langle w, U_{t,i} \rangle^4 + (\mathbb{E}[Z])^2 \\
&\leq \frac{1}{p} \sum_{i=1}^m \langle w, U_{t,i} \rangle^4 + (\mathbb{E}[Z])^2 \\
&= \frac{1}{p} \frac{\mu_2^2 k}{m} + (\mathbb{E}[Z])^2
\end{aligned}
\tag{53}
$$

where the first step follows from the definition of $Z$, the second step follows from $(ab)^2 = a^2 b^2$, the third step follows from $\mathbb{E}[ca] = c \mathbb{E}[a]$, where $c$ is a constant, the fourth step follows from $\mathbb{E}[a + b] = \mathbb{E}[a] + \mathbb{E}[b]$, the fifth step follows from $\mathbb{E}[\delta_{i,j}^2] = p$, the sixth step follows from adding and subtracting the same thing, the seventh step follows from simple algebra, the eighth step follows from simple algebra, the ninth step follows from Eq. (52), the tenth step follows from $(\frac{1}{p} - 1) \leq \frac{1}{p}$, the last step follows from Claim A.16.

Then, we can compute

$$
\begin{aligned}
\mathrm{Var}[Z] &= \mathbb{E}[Z^2] - (\mathbb{E}[Z])^2 \\
&= \frac{\mu_2^2 k}{mp} + (\mathbb{E}[Z])^2 - (\mathbb{E}[Z])^2 \\
&= \frac{\mu_2^2 k}{mp},
\end{aligned}
$$

where the first step follows from the definition of variance, the second step follows from Eq. (53), and the third step follows from simple algebra.

Similarly,

$$
\max_{i \in [m]} |\langle w, U_{t,i} \rangle^2| \leq \frac{\mu_2^2 k}{mp}.
$$

Hence, using Bernstein's inequality (Lemma A.18):

$$\Pr[|Z - \mathbb{E}[Z]| \geq \delta_{2k}] \leq \exp\left(-\frac{\delta_{2k}^2/2}{1 + \delta_{2k}/3}\frac{mp}{\mu_2^2 k}\right).$$

That is, by using $p$ as in the statement of the lemma with the above equation and using union bound, we get (w.p. $> 1 - 1/n^3$): $\forall w \in \mathbb{R}^k, j \in [n]$

$$w^\top B_j w \geq 1 - \delta_{2k}.$$

That is, $\forall j \in [n]$,

$$\sigma_{\min}(B_j) \geq (1 - \delta_{2k}) \geq 0.5.$$

$\square$

### F.5 UPPER BOUNDING $\|C_j\|$ BY $1 + \delta_{2k}$

Now, we analyze $\|C_j\|$ and show that it is bounded by $1 + \delta_{2k}$ for all $j \in [n]$. We prove a variation of Lemma C.7 in Jain et al. (2013).

**Lemma F.5.** *For each $j \in [m]$, we define $C_j \in \mathbb{R}^{k \times k}$ as follows*

$$C_j := \frac{1}{p} \sum_{i \in \Omega_{*,j}} U_{t,i} U_{*,i}^\top$$

*Then, we have: for all $j \in [n]$*

$$\|C_j\| \leq 1 + \delta_{2k}.$$

*Proof.* Let $x \in \mathbb{R}^k$ and $y \in \mathbb{R}^k$ be two arbitrary unit vectors. We define $Z := x^\top C_j y$. Then,

$$
\begin{aligned}
Z &= x^\top C_j y \\
&= \frac{1}{p} \sum_{i \in \Omega_{*,j}} x^\top U_{t,i} \cdot y^\top U_{*,i} \\
&= \frac{1}{p} \sum_{i=1}^m \delta_{i,j} x^\top U_{t,i} \cdot y^\top U_{*,i},
\end{aligned}
$$

where the first step follows from the definition of $Z$, the second step follows from the the notation $\Omega_{*,j} \subset [m]$ being defined in Definition A.13, and the third step follows from definition of $\delta_{i,j}$ which is 1 if $i \in \Omega_{*,j}$ and 0 if $i \notin \Omega_{*,j}$.

Note that,

$$
\begin{aligned}
\mathbb{E}[Z] &= \mathbb{E}\left[\frac{1}{p} \sum_{i=1}^m \delta_{i,j} x^\top U_{t,i} \cdot y^\top U_{*,i}\right] \\
&= \frac{1}{p} \sum_{i=1}^m \mathbb{E}[\delta_{i,j}] x^\top U_{t,i} \cdot y^\top U_{*,i} \\
&= \frac{1}{p} \sum_{i=1}^m p x^\top U_{t,i} \cdot y^\top U_{*,i} \\
&= \sum_{i=1}^m x^\top U_{t,i} \cdot y^\top U_{*,i} \\
&= x^\top U_t^\top U_* y, \quad\quad\quad\quad\quad\quad\quad\quad\quad\quad (54)
\end{aligned}
$$

where the first step follows from the definition of $Z$, the second step follows from the linearity of expectation ($\mathbb{E}[a + b] = \mathbb{E}[a] + \mathbb{E}[b]$), the third step follows from $\mathbb{E}[\delta_{i,j}^2] = p$, the fourth step follows from simple algebra, and the last step follows from the fact that $U$ is a $m \times k$ matrix.

We can compute the second moment

$$
\begin{aligned}
\mathbb{E}[Z^2] &= \mathbb{E}[(\frac{1}{p}\sum_{i=1}^{m}\delta_{i,j}x^\top U_{t,i}\cdot y^\top U_{*,i})^2] \\
&= \frac{1}{p^2}\mathbb{E}[(\sum_{i=1}^{m}\delta_{i,j}x^\top U_{t,i}\cdot y^\top U_{*,i})^2] \\
&= \frac{1}{p^2}(\sum_{i=1}^{m}\mathbb{E}[\delta_{i,j}^2](x^\top U_{t,i}\cdot y^\top U_{*,i})^2 + \sum_{i_1\neq i_2}\mathbb{E}[\delta_{i_1,j}]x^\top U_{t,i_1}\cdot y^\top U_{*,i_1}\mathbb{E}[\delta_{i_2,j}]x^\top U_{t,i_2}\cdot y^\top U_{*,i_2}) \\
&= \frac{1}{p^2}(\sum_{i=1}^{m}p(x^\top U_{t,i}\cdot y^\top U_{*,i})^2 + \sum_{i_1\neq i_2}p^2 x^\top U_{t,i_1}\cdot y^\top U_{*,i_1}x^\top U_{t,i_2}\cdot y^\top U_{*,i_2}) \\
&= \frac{1}{p}(\sum_{i=1}^{m}(x^\top U_{t,i}\cdot y^\top U_{*,i})^2 + p\sum_{i_1\neq i_2}x^\top U_{t,i_1}\cdot y^\top U_{*,i_1}x^\top U_{t,i_2}\cdot y^\top U_{*,i_2}) \\
&= \frac{1}{p}(\sum_{i=1}^{m}(x^\top U_{t,i}\cdot y^\top U_{*,i})^2 - p\sum_{i=1}^{m}(x^\top U_{t,i}\cdot y^\top U_{*,i})^2 + p\sum_{i=1}^{m}(x^\top U_{t,i}\cdot y^\top U_{*,i})^2 \\
&\quad + p\sum_{i_1\neq i_2}x^\top U_{t,i_1}\cdot y^\top U_{*,i_1}x^\top U_{t,i_2}\cdot y^\top U_{*,i_2}) \\
&= (\frac{1}{p}-1)\sum_{i=1}^{m}(x^\top U_{t,i}\cdot y^\top U_{*,i})^2 + (\sum_{i=1}^{m}x^\top U_{t,i}\cdot y^\top U_{*,i})^2 \\
&\leq \frac{1}{p}\sum_{i=1}^{m}(x^\top U_{t,i})^2\cdot(y^\top U_{*,i})^2 + (\sum_{i=1}^{m}x^\top U_{t,i}\cdot y^\top U_{*,i})^2 \\
&= \frac{1}{p}\sum_{i=1}^{m}(x^\top U_{t,i})^2\cdot(y^\top U_{*,i})^2 + (\mathbb{E}[Z])^2 \\
&= \frac{\mu^2 k}{mp} + (\mathbb{E}[Z])^2,
\end{aligned}
\tag{55}
$$

where the first step follows from the definition of $Z$, the second step follows from $\mathbb{E}[ca] = c\,\mathbb{E}[a]$, where $c$ is a constant, the third step follows from $\mathbb{E}[a+b] = \mathbb{E}[a] + \mathbb{E}[b]$, the fourth step follows from $\mathbb{E}[\delta_{i,j}^2] = p$, the fifth step follows from simple algebra, the sixth step follows from adding and subtracting the same thing, the seventh step follows from simple algebra, the eighth step follows from $(xy)^2 = x^2 y^2$, the ninth step follows Eq. (54), the 10th step follows from Claim A.16.

Then, we have

$$
\begin{aligned}
\mathrm{Var}[Z] &= \mathbb{E}[Z^2] - (\mathbb{E}[Z])^2 \\
&= \frac{\mu^2 k}{mp} + (\mathbb{E}[Z])^2 - (\mathbb{E}[Z])^2 \\
&= \frac{\mu^2 k}{mp},
\end{aligned}
$$

where the first step follows from the definition of variance, the second step follows from Eq. (55), and the last step follows from simple algebra.

We have

$$
\max_{i\in[m]}|x^\top U_{t,i}\cdot y^\top U_{*,i}| \leq \frac{\mu_2^2 k}{m}
$$

Lemma now follows from using Bernstein's inequality. □

# G  A PERTURBATION THEORY FOR DISTANCES AND INCOHERENCE

In this section, we present one of the main technical contributions of this paper — a perturbation theory for distances and matrix incoherence. Our main technology is a reduction from incoherence to leverage score, then utilize different parametrizations of leverage score to obtain a bound on incoherence.

In Section G.1, show to bound distance by the spectral norm. In Section G.2, we present the perturbation related lemma that is from right to left low rank factor. In Section G.3, we assert a toll from the previous work: leverage score changes under row perturbation. In Section G.4, we elucidate the perturbation related lemma that is from approximate to exact updates.

## G.1  BOUND DISTANCE BY SPECTRAL NORM

The following lemma establishes a relationship between $\mathrm{dist}$ and the spectral norm.

**Lemma G.1** (Bounded distance by spectral). *Let $X, Y \in \mathbb{R}^{n \times k}$ be two matrices (not necessarily orthogonal). Suppose $\|X - Y\|^2 \leq \sigma_{\min}(X)\sigma_{\min}(Y)$. Then*

$$\mathrm{dist}(X, Y) \leq 4\|X - Y\|\sigma_{\min}(X)^{-0.5}\sigma_{\min}(Y)^{-0.5}.$$

*Proof.*

$$
\begin{aligned}
\cos\theta(X, Y) &= \min_{u \in \mathrm{span}(X)} \max_{v \in \mathrm{span}(Y)} \frac{u^\top v}{\|u\|\|v\|} \\
&\geq \min_{\substack{w \in \mathbb{R}^k \\ u = Xw, v = Yw}} \frac{u^\top v}{\|u\|\|v\|} \\
&= \min_{\substack{w \in \mathbb{R}^k \\ u = Xw, v = Yw}} \frac{1}{2} \cdot \frac{\|u\|^2 + \|v\|^2 - \|u - v\|^2}{\|u\|\|v\|} \\
&\geq \min_{\substack{w \in \mathbb{R}^k \\ u = Xw, v = Yw}} 1 - \frac{1}{2} \cdot \frac{\|u - v\|^2}{\|u\|\|v\|} \\
&\geq \min_{w \in \mathbb{R}^k} 1 - \frac{1}{2} \cdot \frac{\|X - Y\|^2\|w\|^2}{\|u\|\|v\|} \\
&\geq \min_{w \in \mathbb{R}^k} 1 - \frac{1}{2} \cdot \frac{\|X - Y\|^2\|w\|^2}{\sigma_{\min}(X)\sigma_{\min}(Y)\|w\|^2} \\
&= 1 - \frac{1}{2}\|X - Y\|^2\sigma_{\min}(X)^{-1}\sigma_{\min}(Y)^{-1},
\end{aligned}
\tag{56}
$$

where the first step follows from the definition of $\cos\theta(X, Y)$, the second step follows from $\max_i X_i \geq X_i$, the third step follows from simple algebra, the fourth step follows from $a^2 + b^2 \geq 2ab$, the fifth step follows from $\|u - v\|_2^2 = \|Xw - Yw\|_2^2 \leq \|X - Y\|^2\|w\|_2^2$, the sixth step follows from $\|u\|_2 = \|Xw\|_2 \geq \sigma_{\min}(X)\|w\|_2$, and the last step follows from canceling the term $\|w\|_2^2$.

Thus,

$$
\begin{aligned}
\sin^2\theta(X, Y) &= 1 - \cos^2\theta(X, Y) \\
&\leq 1 - \left(1 - \frac{1}{2}\|X - Y\|^2\sigma_{\min}(X)^{-1}\sigma_{\min}(Y)^{-1}\right)^2 \\
&\leq 1 - \left(1 - \|X - Y\|^2\sigma_{\min}(X)^{-1}\sigma_{\min}(Y)^{-1}\right) \\
&= \|X - Y\|^2\sigma_{\min}(X)^{-1}\sigma_{\min}(Y)^{-1},
\end{aligned}
\tag{57}
$$

where the first step follows from $\cos^2\theta(X, Y) + \sin^2\theta(X, Y) = 1$ (see Lemma A.9) and the second steep follows from Eq. (56), the third step follows from $-(a - b)^2 = -a^2 - b^2 + 2ab \leq -a^2 + 2ab$, and the fourth step follows from simple algebra.

Note that $\mathrm{dist}(X, Y) = \sin\theta(X, Y)$, thus, we complete the proof. $\qquad\square$

## G.2 FROM RIGHT TO LEFT FACTOR

Given an orthonormal basis $Z$ and a target matrix $M$, suppose $M = XZ^\top$ for some matrix $X$. We prove some relations on singular values of $X$ and $M$ given some conditions on $Z$.

**Lemma G.2.** *Let $M = U_*\Sigma_*V_*^\top$.*

*Suppose the matrix $Z \in \mathbb{R}^{n \times k}$ satisfies that*

- $Z \in \mathbb{R}^{n \times k}$ *is an orthonormal basis*
- $\mathrm{dist}(Z, V_*) \leq \sqrt{1 - \alpha^2}$

*Let $X \in \mathbb{R}^{m \times k}$ denote the matrix that*

$$M = XZ^\top. \tag{58}$$

*Then, we have*

$$\sigma_{\min}(X) \geq \alpha \cdot \sigma_{\min}(M)$$

*and*

$$\sigma_{\max}(X) \leq \sigma_{\max}(M)$$

*Proof.* We have

$$\|V_{*,\perp}^\top Z\| = \mathrm{dist}(Z, V_*)$$
$$\leq \sqrt{1 - \alpha^2}. \tag{59}$$

where the first step follows from the definition of distance and the second step follows from the Lemma statement.

Using $\sin^2 \theta + \cos^2 \theta = 1$ (Lemma A.9), we have

$$\sigma_{\min}(V_*^\top Z)^2 + \|V_{*,\perp}^\top Z\|^2 = 1. \tag{60}$$

We can show

$$\sigma_{\min}(V_*^\top Z) = \sqrt{1 - \|V_{*,\perp}^\top Z\|^2}$$
$$\geq \sqrt{1 - (1 - \alpha^2)}$$
$$= \alpha, \tag{61}$$

where the first step follows from Eq. (59), the second step follows from Eq. (60), and the third step follows from simple algebra.

We have

$$\sigma_{\min}(X) = \sigma_{\min}(M \cdot (Z^\top)^\dagger)$$
$$= \sigma_{\min}(U_*\Sigma_*V_*^\top \cdot (Z^\top)^\dagger)$$
$$= \sigma_{\min}(\Sigma_*V_*^\top \cdot (Z^\top)^\dagger)$$
$$= \sigma_{\min}(\Sigma_*V_*^\top \cdot Z)$$
$$\geq \sigma_{\min}(\Sigma_*) \cdot \sigma_{\min}(V_*^\top \cdot Z)$$
$$= \sigma_{\min}(M) \cdot \sigma_{\min}(V_*^\top \cdot Z)$$
$$\geq \sigma_{\min}(M) \cdot \alpha,$$

where the first step follows from Eq. (58), the second step follows from $M = U_*\Sigma_*V_*^\top$, the third step follows from $U_*$ has orthonormal columns, the fourth step follows from $(Z^\top)^\dagger = Z$, the fifth step follows from part 11 of Fact A.1, the sixth step follows from definition of matrix $M \in \mathbb{R}^{m \times n}$, and the last step follows from Eq. (61). $\square$

### G.3   LEVERAGE SCORE CHANGE UNDER ROW PERTURBATIONS

We analyze the change to leverage scores when the rows are perturbed in a structured manner.

**Lemma G.3.** *Given two matrices $A \in \mathbb{R}^{m \times k}$ and $B \in \mathbb{R}^{m \times k}$. For each $i \in [m]$, let $a_i$ denote the $i$-th row of matrix $A$. For each $i \in [m]$, let $b_i$ denote the $i$-th row of matrix $B$.*

*If*

- $\|A - B\| \leq \epsilon_0$

- $\epsilon_0 \leq \frac{1}{2}\sigma_{\min}(A)$

*then we have*

$$\|\|(A^\top A)^{-1/2}a_i\|_2 - \|(B^\top B)^{-1/2}b_i\|_2\| \leq 75\epsilon_0\sigma_{\min}(A)^{-1}\kappa^4(A).$$

*Proof.* Without loss of generality assuming $\|(A^\top A)^{-1/2}a_i\|_2 \geq \|(B^\top B)^{-1/2}b_i\|_2$ as the other case is symmetric. Note that by the difference of squares formula, we know that

$$\|(A^\top A)^{-1/2}a_i\|_2 - \|(B^\top B)^{-1/2}b_i\|_2 = \frac{\|(A^\top A)^{-1/2}a_i\|_2^2 - \|(B^\top B)^{-1/2}b_i\|_2^2}{\|(A^\top A)^{-1/2}a_i\|_2 + \|(B^\top B)^{-1/2}b_i\|_2},$$

it suffices to provide an upper bound on the numerator and a lower bound on the denominator.

**Lower bound on denominator.** The lower bound is relatively straightforward:

$$\begin{aligned}
\|(A^\top A)^{-1/2}a_i\|_2 + \|(B^\top B)^{-1/2}b_i\|_2 &\geq \frac{1}{\sigma_{\max}(A)}\|a_i\|_2 + \frac{1}{\sigma_{\max}(B)}\|b_i\|_2 \\
&\geq \frac{1}{\sigma_{\max}(A)}\|a_i\|_2 + \frac{1}{\sigma_{\max}(B) + \epsilon_0}\|b_i\|_2 \\
&\geq \frac{\|a_i\|_2 + \|b_i\|_2}{\sigma_{\max}(A) + \epsilon_0} \\
&\geq \frac{2\sigma_{\min}(A) - \epsilon_0}{2\sigma_{\max}(A)} \\
&\geq \frac{3}{4}\kappa^{-1}(A)
\end{aligned}$$

where we use Weyl's inequality (Lemma A.2) to uppper bound $\sigma_{\max}(B)$ by $\sigma_{\max}(A) + \epsilon_0$, $\epsilon_0 \leq \frac{1}{2}\sigma_{\min}(A)$ and $\|a_i\|_2 \leq \|A\|$.

**Upper bound on the numerator.** The upper bound can be obtained via a chain of triangle inequalities.

We start by proving some auxiliary result: First, we can show

$$\begin{aligned}
\|A^\top A - B^\top B\| &= \|A^\top A - A^\top B + A^\top B - B^\top B\| \\
&\leq \|A^\top A - A^\top B\| + \|A^\top B - B^\top B\| \\
&\leq \sigma_{\max}(A)\epsilon_0 + \sigma_{\max}(B)\epsilon_0 \\
&\leq (2\sigma_{\max}(A) + \epsilon_0) \cdot \epsilon_0 \\
&\leq 3\sigma_{\max}(A) \cdot \epsilon_0
\end{aligned}$$

We can then prove the discrepancy between inverses:

$$\begin{aligned}
&\|(A^\top A)^{-1} - (B^\top B)^{-1}\| \\
&\leq 2\max\{\|(A^\top A)^{-1}\|^2, \|(B^\top B)^{-1}\|^2\} \cdot \|(A^\top A) - (B^\top B)\| \\
&\leq 2 \cdot (2/\sigma_{\min}(A)^2)^2 \cdot \|(A^\top A) - (B^\top B)\| \\
&\leq 2 \cdot (2/\sigma_{\min}(A)^2)^2 \cdot (3\sigma_{\max}(A) \cdot \epsilon_0) \\
&= 24\kappa(A)\sigma_{\min}^{-3}(A)\epsilon_0,
\end{aligned}$$

where the first step is by Lemma A.15.

Finally, note

$$|a_i^\top (A^\top A)^{-1} a_i - b_i^\top (B^\top B)^{-1} b_i|$$
$$\leq |a_i^\top (A^\top A)^{-1} a_i - a_i^\top (A^\top A)^{-1} b_i| + |a_i^\top (A^\top A)^{-1} b_i - a_i^\top (B^\top B)^{-1} b_i| + |a_i^\top (B^\top B)^{-1} b_i - b_i^\top (B^\top B)^{-1} b_i|$$
$$\leq \|a_i\|_2 \cdot \|a_i - b_i\|_2 \cdot \|(A^\top A)^{-1}\| + \|a_i\|_2 \cdot \|b_i\|_2 \cdot \|(A^\top A)^{-1} - (B^\top B)^{-1}\| + \|b_i\|_2 \cdot \|a_i - b_i\|_2 \cdot \|(B^\top B)^{-1}\|$$
$$\leq \epsilon_0 \sigma_{\min}^{-2}(A)\|a_i\|_2 + 24\epsilon_0 \kappa(A)\sigma_{\min}^{-3}(A)(\|a_i\|_2^2 + \epsilon_0\|a_i\|_2) + \epsilon_0(\sigma_{\min}(A) - \epsilon_0)^{-2}\|b_i\|_2$$
$$\leq \epsilon_0 \kappa(A)\sigma_{\min}^{-1}(A) + 48\epsilon_0 \kappa^3(A)\sigma_{\min}^{-1}(A) + \frac{5}{8}\epsilon_0 \kappa(A)\sigma_{\min}^{-1}(A)$$
$$\leq 50\epsilon_0 \kappa^3(A)\sigma_{\min}^{-1}(A).$$

**Put things together.** The final upper bound can be obtained by taking the ratio between these two terms:

$$\|(A^\top A)^{-1/2} a_i\|_2 - \|(B^\top B)^{-1/2} b_i\|_2 \leq 75\epsilon_0 \kappa^4(A)\sigma_{\min}^{-1}(A).$$

This completes the proof.

$\square$

### G.4 FROM APPROXIMATE TO EXACT DISTANCE AND INCOHERENCE

Given a matrix $X$ that is incoherent, we show that if a matrix $Y$ is close to $X$ enough in spectral norm, then it also preserves the distance and incoherence of $Y$.

**Lemma G.4.** *Let $X \in \mathbb{R}^{m \times k}$ and $X = U\Sigma V^\top$ be its SVD. Then*

$$\|u_i\|_2^2 = x_i^\top (X^\top X)^{-1} x_i.$$

*where $u_i, x_i$ denote the $i$-th row of $U$ and $X$ respectively.*

*Proof.* Let us form the projection matrix $X(X^\top X)^{-1}X^\top$, note that the quantity on the RHS is the $i$-th diagonal of the projection matrix.

$$X(X^\top X)^{-1}X^\top = U\Sigma V^\top (V\Sigma^2 V^\top)^{-1} V\Sigma U^\top$$
$$= U\Sigma V^\top V\Sigma^{-2} V^\top V\Sigma U^\top$$
$$= UU^\top,$$

where the first step follows from the Lemma statement: $X = U\Sigma V^\top$, the second step follows from simple algebra, the third step follows from simple algebra.

The $i$-th diagonal of $UU^\top$ is then $u_i^\top u_i = \|u_i\|_2^2$, as desired. $\square$

**Lemma G.5.** *Let $X, Y \in \mathbb{R}^{m \times k}$. Let $\mu_0 \geq 1$. Let $\epsilon_0 \in (0, 1)$.*

*Suppose that*

- *$X$ is $\mu_0$ incoherent, i.e., $\|(U_X)_i\|_2 \leq \mu_0 \sqrt{k}/\sqrt{m}$.*

- *Let $\|Y - X\| \leq \epsilon_0$.*

- *Let $\epsilon_0 \leq 0.1 \cdot \sigma_{\min}(X)$*

*For each $i \in [m]$, let $X_i$ denote the $i$-th row of matrix $X \in \mathbb{R}^{m \times k}$. Let $\kappa(X) := \sigma_{\max}(X)/\sigma_{\min}(X)$ be the condition number of $X$. Let $\epsilon_{\mathrm{sk}} := \Theta(\epsilon_0 \cdot (m \cdot \kappa^4(X) \cdot \sigma_{\min}(X)^{-1}))$.*

*Then, we have*

- *Part 1. $\mathrm{dist}(Y, U_*) \leq \mathrm{dist}(X, U^*) + \epsilon_{\mathrm{sk}}$.*

- *Part 2. $Y$ is $\mu_0 + \epsilon_{\mathrm{sk}}$ incoherent, i.e., $\|(U_Y)_i\|_2 \leq (\mu_0 + \epsilon_{\mathrm{sk}})\sqrt{k}/\sqrt{m}$.*

**Remark G.6.** *Eventually we choose $\epsilon_{\mathrm{sk}} = \epsilon/\Theta(T)$ where $\epsilon$ is the final accuracy parameter $(0, 1/10)$ and $T$ is the number of iterations. We will use this lemma for $\Theta(T)$ times, and each time we incur an extra $\epsilon/\Theta(T)$ error. Our ultimate goal is to shrink $\mathrm{dist}(Y, U_*)$ to $\Theta(\epsilon)$ after $T$ iterations. Since all iterations only incur extra $\Theta(\epsilon)$ error, the final $\mathrm{dist}(Y, U_*)$ is still bounded by $\Theta(\epsilon)$. When we use this Lemma, $\kappa(X)$ is always bounded by $O(\kappa(M))$ where $M$ is the ground truth matrix in matrix completion problem. Note that our running time blowup is only logarithmically in $1/\epsilon_0$, thus it's always fine to set $\epsilon$ to be $1/\mathrm{poly}(n, \kappa, T)$ smaller. Similar argument will hold for part 2 incoherent. Since $\mu_0 \geq 1$, over all the iterations, the incoherent parameter is within $2\mu_0$.*

*Proof.* We prove two parts separately.

**Proof of Part 1.** From lemma assumptions, we have

$$\|Y - X\| \leq \epsilon_0 \leq 0.1 \sigma_{\min}(X)$$

Then, we know that

$$\sigma_{\min}(Y) \geq 0.5 \sigma_{\min}(X). \tag{62}$$

We have

$$
\begin{aligned}
\mathrm{dist}(Y, U_*) &\leq \mathrm{dist}(X, U_*) + \mathrm{dist}(Y, X) \\
&\leq \mathrm{dist}(X, U_*) + \|Y - X\| \cdot \sigma_{\min}(X)^{-0.5} \cdot \sigma_{\min}(Y)^{-0.5} \\
&= \mathrm{dist}(X, U_*) + \epsilon_0 \cdot \sigma_{\min}(X)^{-0.5} \sigma_{\min}(Y)^{-0.5} \\
&\leq \mathrm{dist}(X, U_*) + 2\epsilon_0 \cdot \sigma_{\min}(X)^{-1} \\
&\leq \mathrm{dist}(X, U_*) + \epsilon_{\mathrm{sk}},
\end{aligned}
$$

where the first step follows from triangle inequality, the second step follows from Lemma G.1, the third step follows from lemma statement $\|Y - X\| \leq \epsilon_0$, the fourth step follows from Eq. (62), the last step follows from definition $\epsilon_{\mathrm{sk}}$ in the lemma statement.

**Proof of Part 2.** By Lemma G.3 and Lemma G.4, we know that for any $i \in [m]$

$$\left| \|(U_X)_i\|_2 - \|(U_Y)_i\|_2 \right| \leq 500 \cdot \epsilon_0 \cdot \sigma_{\min}(X)^{-1} \cdot \kappa^4(X).$$

Since $X \in \mathbb{R}^{m \times k}$ is $\mu_0$ incoherent, by triangle inequality, we have

$$
\begin{aligned}
\|(U_Y)_i\|_2 &\leq \mu_0 \sqrt{k}/\sqrt{m} + 500 \cdot \epsilon_0 \cdot \sigma_{\min}(X)^{-1} \cdot \kappa^4(X) \\
&\leq (\mu_0 + \epsilon_{\mathrm{sk}}) \cdot \sqrt{k}/\sqrt{m}
\end{aligned}
$$

where the last step follows from definition of $\epsilon_{\mathrm{sk}}$ in the lemma statement.

Thus, we complete the proof. $\qquad\square$

