# OpenReview forum: "Low Rank Matrix Completion via Robust Alternating Minimization in Nearly Linear Time"
_ICLR.cc/2024/Conference — ICLR 2024 poster_

### Official Review · Reviewer_kPm7 · 2023-10-31

**Soundness:** 3 good
**Presentation:** 2 fair
**Contribution:** 3 good
**Rating:** 6
**Confidence:** 3

**Summary:**

In this work, the authors proposed the analysis of a robust and fast matrix completion algorithm based on the alternating minimization method. The total running time is linear in terms of the complexity of verifying the correctness of a completion.

**Strengths:**

The work provides novel theoretical results on the alternating minimization approach for matrix completion. The results should be interesting to researchers in the optimization and machine learning fields.

**Weaknesses:**

In my opinion, the presentation of the paper can be improved. The current paper spends too much space on providing the intuition of the proposed results. This makes the paper too dry to understand. In addition, I think the authors need to include more technical details in the main body of the paper. Due to the time limit, I do not have time to check the appendix. So I cannot be sure about the correctness of the theoretical results given the limited information in the main manuscript.

**Questions:**

(1) Page 2, line 1: \epsilon is not defined.

(2) Page 2: "as the perturbation to incoherence itself is not enough the algorithm approaches the true optimum". It seems that the sentence is not complete.

(3) Section 2: it might be better to compare the running time and sample complexity of alternating minimization and (stochastic) gradient descent.

(4) Page 3: "weighted low rank approximation" -> "weighted low-rank approximation"

(5) Page 5: the (total) instance size for the multiple response regression is not defined.

(6) Section 4: I wonder if the partition of samples across iterations can be avoided? Namely, can we use all samples in different iterations? It would be better to clarify if the reuse of samples will fail the proposed algorithms, or only makes it technically difficult to prove the theoretical results.

(7) Algorithm 1, line 5: it seems that U_0 is not used later in the algorithm.

(8) Section 4.1: "but to conclude the desired guarantee on the output, we also need to show that..." It would be helpful to be more specific on the connection between the desired guarantee and the incoherence condition of \hat{U}_t, \hat{V}_t.

(9) Theorem 4.2: I think it might be better to provide the main results (Theorem 4.2) a little earlier in the paper. The preparation for the statement of the main results is too long. The contents of Section 4 is basically a more detailed version of Section 1. Given the page limit, I feel that it is more efficient to simplify the discussion of techniques in Section 1, but include more details in Sections 4.1-4.3.

---

> ### Author Response · Authors · 2023-11-17
> **Thank you for your review!**
>
> We appreciate your valuable and insightful comments on our manuscript, particularly regarding its structure. We have significantly condensed the discussion in Section 1 and now present our informal main result at its conclusion. Additionally, we have enriched Sections 4.1 to 4.3 with more detailed discussions, aiming to enhance clarity about our techniques and overall proof strategy.
>
> We would like to emphasize that our decision to focus on informal intuitions and technique overviews in the main body of the paper is a deliberate choice, given its technical nature. The paper's results are highly technical, and deriving a major result requires many intricate components. Including these in the main body could make the paper overly technical and challenging for readers. Therefore, we have opted for a format that conveys the core ideas behind the proofs while minimizing too complicated technical details.

---

> > ### Comment · Reviewer_kPm7 · 2023-11-21
> > **Response to authors' rebuttal**
> >
> > I would like to thank the authors for the response and the revision. I will keep my rating.

---

### Official Review · Reviewer_CTEU · 2023-11-02

**Soundness:** 2 fair
**Presentation:** 2 fair
**Contribution:** 3 good
**Rating:** 6
**Confidence:** 3

**Summary:**

The paper studies the low rank matrix completion problem using alternating minimization. Existing algorithms based on alternating minimization for this problem (Jain, Netrapalli and Sanghavi) takes time $\tilde{O}(|\Omega|k^2)$ time where $\Omega$ is the set of entries samples (nearly linear in $n$ for an $m \times n$ incoherent matrix $M$) and $k$ is the target rank $(k<<m,n)$. The main computational bottleneck in alternating minimization comes from solving two multiple response regressions per iteration (once for U and once for V). The algorithm presented in this paper proposes solving the regression problems approximately using off-the shelf sketching based solvers which take nearly linear time in input size per iteration and thus, the time for solving each regression problem reduces to $\tilde{O}(|\Omega|k)$ (with extra $\log(1/\epsilon)$ steps for convergence). However, this complicates the analysis of the algorithm as the solution at every iteration cannot be written exactly in factorized form (so the previous analysis doesn't carry through). This runtime is equal to verification time for the solution upto polylog factors. To analyze this, an inductive argument is presented which shows that at every step, the approximate solution for $U$ or $V$ is close to the optimal solution $U^*$ and $V^*$. Moreover, it is shown that the incoherence of the exact solutions to the regression problem is preserved. Finally, to show that the incoherence of the approximate solution is also preserved, some matrix perturbation bounds are developed which show that as long as any two matrices are very close in spectral norm and one matrix is incoherent, the other matrix will also be incoherent. The sample complexity for the proposed algorithm is the same as that of the old algorithm.

**Strengths:**

1) The algorithm presented improves upon the runtime to make it nearly linear in verification time of the solution (up to log factors) i.e. $\tilde{O}(|\Omega|k)$. Previous alternating minimization based algorithms try to solve the regressions exactly and hence incur $\tilde{O}(|\Omega|k^2)$ time. Moreover, the proposed algorithm is practical and easy to implement as different off-the-shelf sketching based solvers can be used for this regression step. Also, the sample complexity remains the same as previous algorithms.

2) Some interesting technical result are developed for the theoretical analysis of the algorithm. Specifically, some matrix perturbation bounds are proven which show that if a matrix with incoherent rows and columns is close in spectral norm to another matrix, that matrix will also be incoherent with the incoherence factor depending on the condition number of the first matrix. This seems to be an interesting result which could be of independent interest (though I have some questions related to the proof, please see the questions section).

Remark: I haven't checked all the proofs in the appendix closely (especially the proofs related to induction in Section E and F of the appendix).

**Weaknesses:**

1) Though the runtime of the proposed algorithm is nearly linear in verification time and improves on the runtime compared to previous algorithms, without any discussion on computational limitations or lower bounds, it is hard to judge if this is indeed a significant theoretical result for this problem. Some discussion on runtime or sample complexity lower bound could be useful to understand what is the runtime one should be aiming for this problem.

2) I'm unsure of certain key steps in the proofs for the forward error bound and the matrix perturbation results (please see the questions).

3) The proofs in the appendix seems some confusing notations and sometimes uses certain notations without defining first them which cause problems while reading the proofs:

  i) For example, in Lemma B.6, in some places $D_{W_i}$ seems to be indicate a matrix with $W_i$ on diagonal and on other places, a constant? When defining $\| z\|_w^2$ is should be just ||z ||_w^2=\sum_{i=1}^n  w_i^2   I think for a vector $w$?

  ii) Also, in definition 3.5, M is factorized as $U \Sigma V^T$ while in the appendix, it seems $U^* \Sigma^* V^*$ is used?


Though the paper has interesting results, I'm recommending a borderline reject with a major concern being some of the key steps in the proofs (please see the questions).

**Questions:**

I could be misunderstanding the following steps in the proofs:

1)  In forward error bound of Lemma B.5, I'm confused why the step ||Ax'-b-(Ax_{OPT}-b)||_2^2=||Ax'-b||_2^2-||Ax_{OPT}-b||_2^2 should be true. Why should the Pythagorean theorem hold in this case? Is Ax_{OPT}-b orthogonal to Ax'-b-(Ax_{OPT}-b) due to some condition?
Also, it seems A is assumed to have full column rank for the proof. Is A guaranteed to have full column rank whenever this result is applied i.e. in all iterations wheneer B.7 is invoked?

2) In Lemma G.3, I'm not able to understand how $\sigma_{min}(B) \geq 0.5\sigma_{min}(A)$ follows from $||A-B|| \leq \epsilon_o \leq \sigma_{min}(A)$.

---

> ### Author Response · Authors · 2023-11-17
> **Thank you for your review!**
>
> We appreciate your helpful and insightful comments that improve the presentation and clarity of our paper. We would like to address your questions and comments.
>
> 1. Sample complexity lower bound. For this problem, it is known that $\Omega(nk)$ samples are needed by a simple parameter counting argument. Beyond that, Hardt et al., 2014 has shown that even if the ground truth is rank-4, has incoherent rows and columns, we can observe 90\% of entries and in additional, we are allowed to output a higher constant rank matrix, then this problem is still NP-hard (via a reduction from 4-coloring). From an upper bound side, it is known that methods based on convex relaxation (Candes and Tao, 2010; Recht, 2011) provide a sample complexity upper bound $\widetilde O(nk)$, but these methods require $O(n^\omega)$ time to solve a semidefinite program. For non-convex programming methods, current best algorithm is due to Kelner et al., 2023 which requires $O(nk^{2+o(1)})$ samples. However, as we have discussed in our paper, it is not clear whether their algorithm can provide a good practical performance as their method involves using many techniques with good theoretical runtime guarantees but lack practical validations. See the end of Section 4 of our revised draft for a more detailed comparison.
>
> 2. Proof notations: for a non-negative vector $w\in \mathbb{R}^n$, we use  $\\|v\\|_w^2=\sum_i w_iv_i^2=\sum_i (D_W)_i v_i^2$ to denote the weighted norm, where the second equality follows from we can rewrite a vector by a diagonal matrix that puts the vector on the diagonal. In definition 3.5, the factorization $M=U\Sigma V^\top$ is a *local* definition, as we will use the incoherent assumptions on other matrices. The factorization $M=U^* \Sigma^* (V^*)^\top$ is a *global* definition of the ground truth.
>
> 3. Lemma B.5. We have revised the lemma statement and provided an alternative proof without assuming $A$ has full column rank. Further, we have provided a comprehensive explanation why using the Pythagorean theorem is valid. For more details, see page 25-26 in our revised draft.
>
> 4. Lemma G.3. We have added an additional assumption $\epsilon_0 \leq \frac{1}{2}\sigma_{\min}(A)$, note this won’t affect the overall analysis. We also significantly simplify and streamline the proof so that it’s easier to understand. For more details, see page 51-52 in our revised draft.
>
> Hardt et al., 2014. Hardt, Meka, Raghavendra and Weitz. Computational Limits for Matrix Completion. COLT'2014.
>
> Candes and Tao, 2010. The power of convex relaxation: Near-optimal matrix
> completion. IEEE Trans. Inf. Theory'2010.
>
> Recht, 2011. A simpler approach to matrix completion. JMLR'2011.
>
> Kelner et al., 2023. Kelner, Li, Liu, Sidford and Tian. Matrix Completion in Almost-Verification Time. FOCS'2023.

---

> > ### Comment · Reviewer_CTEU · 2023-11-21
> >
> > I thank the authors for their detailed response to my queries. Most of my concerns have been addressed. I am increasing my score to a weak accept. This is primarily a theoretical paper and has some interesting theoretical results on alternating minimization and matrix perturbation. The only lingering concern I have is the lack of any experimental results to show how the proposed algorithm compares to other algorithms particularly Kelner et.al. Even though the authors claim that their method is more numerically stable and thus more practical than Kelner et al., having at least some experimental comparison would have been beneficial especially considering that the venue is ICLR and the broader ML community might be interested in practical implementations.

---

### Official Review · Reviewer_QBsW · 2023-11-03

**Soundness:** 3 good
**Presentation:** 3 good
**Contribution:** 2 fair
**Rating:** 6
**Confidence:** 3

**Summary:**

The authors give a nearly linear time algorithm (in the number of samples) for low-rank matrix completion. Specifically, they give a $O(|\Omega| k)$ time alternating-minimization based algorithm that converges to the original underlying rank-$k$, $\mu$-incoherent matrix $M \in \mathbb{R}^{m \times n}$ when $|\Omega| = \tilde{O}(\kappa^2 \mu^2 n k^{4.5})$ samples are drawn from it.

The running time guarantee improves on a line of works, starting with that of Jain et. al (2013), on the efficiency of each step in AM framework -- going from $O(|\Omega| k^2)$ running time to $O(|\Omega| k)$ time. This compares however to a recent paper (that uses a different approach altogether) by Kelner et. al (2023) also achieving a $O(|\Omega| k)$ running time with significantly fewer samples $|\Omega| = O(n k^{2 + o(1)})$.

The improvement comes from solving each multiple response regression problem (to obtain the low-rank factorization $UV$) approximately instead of exactly.

The authors main technical contribution is in analyzing the how the error introduced in solving for $U$ and $V$ approximately, propagates in the iterative process. Specifically they show, using a careful double induction argument and an incoherence bound on the perturbation of row norms of incoherent matrices that the incoherence of the approximate factors in the $t$-th iteration $\hat{U}_t, \hat{V}_t$ as well as the exact solutions they are approximating $U_t, V_t$ are incoherent as well as approach the true subspaces $U^*, V^*$.

\textbf{References}

Prateek Jain, Praneeth Netrapalli, and Sujay Sanghavi. Low-rank matrix completion using alternating
minimization. In Proceedings of the forty-fifth annual ACM symposium on Theory of computing,
pp. 665–674, 2013

Jonathan Kelner, Jerry Li, Allen Liu, Aaron Sidford, and Kevin Tian. Matrix completion in almostverification time. In 2023 IEEE 64th Annual Symposium on Foundations of Computer Science,
FOCS’23, 2023

**Strengths:**

Originality and Significance
The main contribution is the error analysis in the AM iterations, showing how the subspaces of the approximate solutions to the multiple response regressions $\hat{U}_t, \hat{V}_t$ converge to the true factors. The novelty comes from a double induction argument tying the incoherence and the closeness of the approximate solution to that of the exact solution in each iteration.

The technique sheds light on how AM algorithms for low-rank matrix completion can be sped-up (using approximate solvers). Since AM algorithms are popular in practice for this problem, this theoretical result can help substantiate the design of new more efficient algorithms.

Quality and Clarity
Overall the paper is well organized and written. The paper compares to relevant works sufficiently well and highlights the difference facets in which this result compares.

**Weaknesses:**

The main weakness might be in the significance of the final running time result in the context of the recent result by Kelner et. al (2023). Given that Kelner et. al achieve a significantly lower sample complexity of $\tilde{O}(n k^2)$ (with no dependence on $\kappa$), the novelty of this result could be questioned. Especially since the result is theoretical and no experiments have been provided to justify the efficiency of this approach.

**Questions:**

- Can you speak more to the significance of your result as compared to that of Kelner et. al (2023)? Specifically to the significance of the running time result given they achieve an asymptotically smaller running time result (please correct me if that is incorrect).

---

> ### Author Response · Authors · 2023-11-17
> **Thank you for your review!**
>
> We appreciate your insightful and valuable comments. We would like to highlight the significance of our results in comparison to Kelner et al., 2023b. Indeed, Kelner et al., 2023b offers a theoretically state-of-the-art sample complexity and runtime for a non-convex programming-based algorithm in matrix completion. Under standard incoherence assumptions, they achieve a sample complexity of $|\Omega|=O(nk^{2+o(1)})$ and a runtime of $\widetilde O(|\Omega|k)$. However, their approach significantly diverges from the algorithms commonly used in matrix completion practice.
>
> Their algorithm, broadly speaking, is grounded in the 'short-flat decomposition' developed in their earlier work (Kelner et al., 2023a) on semi-random sparse recovery. This decomposition separates the observation into a useful term and a noisy term with 'flat' singular values. The algorithm begins with initial matrices $U_0, V_0=0$, and processes a set of observations from M by first removing large rows and columns from $P_{\Omega_t}(M-U_tV_t^\top)$, followed by running a top-k SVD procedure from Musco and Musco, 2015. This procedure is iterated for ${\rm polylog}(1/\epsilon)$ rounds. Due to dropping rows and columns with large norms, they need to recover them back in the post-processing. This can be achieved through regression 'tests' to determine if a set of columns can linearly form a targeted column. The recovery algorithm then proceeds by sampling columns, tests for linear independence, and this process repeats for ${\rm polylog}(n)$ times. Ultimately, the selected columns help fill missing entries for a partially completed matrix for $U$, and the matrix $V$ is completed via Nesterov’s accelerated gradient descent (AGD). They employ Nesterov’s AGD for its good condition number dependence. Of course, they also need a bound on the initial error, which is done via estimating $\log(1/\epsilon)$ many spectral norms over independent samples of $M$, and outputting the median using Chernoff bound.
>
> From a practical standpoint, their algorithm is likely slower than alternating minimization or SGD due to the need for top-k SVD computation and linear independence testing for ${\rm polylog}(n, 1/\epsilon)$ rounds, along with an additional $\log(1/\epsilon)$ rounds for multiple response regression using Nesterov’s AGD. The randomized top-k SVD algorithm they employ, based on block Krylov subspace iteration, is known for numerical instability. While Nesterov’s AGD offers theoretical advantages, it is challenging to achieve fast practical implementation, often making Polyak’s momentum a preferred choice for its simplicity and practical efficacy.
>
> In contrast, alternating minimization and stochastic gradient descent are the most widely applied approaches in practice for matrix completion, despite theoretically inferior sample complexity. These methods are practically efficient, generally involving a sequence of least squares problems solvable by iterative methods. Our robust alternating minimization framework provides a theoretical justification for the practical success of alternating minimization, benefiting from faster solvers. We also believe that further tightening the sample complexity bound for alternating minimization could lead to direct runtime improvements for our algorithm.
>
> Kelner et al., 2023a. Kelner, Li, Liu, Sidford and Tian. Semi-random sparse recovery in nearly-linear time. COLT’2023.
>
> Kelner et al., 2023b. Kelner, Li, Liu, Sidford and Tian. Matrix completion in almost-verification time. FOCS’2023.
>
> Musco and Musco, 2015. Randomized block krylov methods for stronger and faster approximate singular value decomposition. NeurIPS’2015.

---

### Author Response · Authors · 2023-11-17
**Modifications to the draft**

Dear reviewers,

We have updated our pdf draft on the open review, by incorporating suggestions and comments from you. Thank you all for your valuable insights and comments! Below is a list of modifications we have made:

1. We have presented an informal version of our main theorem at the end of Section 1, so that readers have a clearer picture on the main result earlier in the paper.

2. In Section 4.1 - 4.3, technique overviews, we have added more details. For example, in Section 4.1, we demonstrate the necessity of developing a perturbation theory --- as our induction step assumes the exact matrices $U_{t+1}, V_{t+1}$ are incoherent, but to further bound the distance, we need $\widehat U_{t+1}, \widehat V_{t+1}$ are incoherent. The later is hard to enforce in the induction step, but we alternatively assume the exact matrices are incoherent, and use perturbation to show that the approximate matrices are also incoherent. We similarly expand on the proof for perturbation in Section 4.2.

3. We included a detailed comparison with Kelner et al. (2023b) at the end of Section 4. In short, both of our algorithms have a runtime behavior of $\widetilde O(|\Omega| k)$, but theirs has a better sample complexity in terms of dependency on $k$ and $\kappa$. However, we also note that their algorithm is much more complicated than widely-used alternating minimization or (stochastic) gradient descent. To implement their algorithm, one has to resort to approximate singular values and spectral norms, use Nesterov's accelerated gradient descent and a complicated post-processing phase. It is unclear when deployed in practice, their algorithm will actually outperform alternating minimization or (S)GD.

4. In Appendix, Lemma B.5, we modified the proof and dropped the assumptions the matrix $A$ has full column rank. Our new proof is based on SVD and works for any non-full rank matrix. We also elaborate on we can apply Pythagorean theorem. In short, we prove that the subspace $A(A^\top A)^\dagger A^\top-I$ is orthogonal to the column space of $A$, and $Ax_{{\rm OPT}}-b=(A(A^\top A)^\dagger A^\top-I)b$ while $Ax'-Ax_{{\rm OPT}}=A(x'-x_{{\rm OPT}})$. Thus we can apply Pythagorean theorem. For more details, we refer you to page 26 - 27 of the draft.

5. In Appendix, Lemma G.3, we additionally require $\epsilon_0\leq \frac{1}{2}\sigma_{\min}(A)$ to make the proof easier to understand, and we further simplify the proof by using an argument based on difference of squares formula. This makes the proof logic cleaner and easier to follow. We refer you to page 51 - 52 for more details.

---

### Author Response · Authors · 2023-11-21
**Reminder: Discussion period ends soon**

Dear Chairs and Reviewers,

We would like to kindly remind you that the discussion period is ending soon. We highly appreciate all the time and effort the reviewers have dedicated to our manuscript, along with their insightful suggestions. We have carefully considered these comments and believe we have addressed each of them. Should there be further concerns, we are eager to engage in further discussion with the reviewers.

Reviewer QBsW's primary concern revolves around the significance of our work in comparison to Kelner et al. (2023). We have provided a detailed explanation and discussion, not only in response to the reviewer but also within our manuscript (refer to page 9).

Reviewer CTEU has raised concerns about our proofs. We have provided detailed explanations addressing all of Reviewer CTEU's concerns and have defended the soundness of our proofs. Additionally, to enhance the presentation of our paper, we have made corresponding changes to the manuscript, aimed at ensuring future readers are more convinced of the solidity of our results.

Regarding Reviewer kPm7's suggestions, we have followed the advice to improve the manuscript's presentation by condensing Section 1 and enriching Sections 4.1 to 4.3 with more detailed discussions, thereby enhancing clarity in our technique overview. We also mainly focus on intuitions because the results are highly technical, and presenting intuition will enhance readability for our audience.

We believe that we have addressed all reviewers' comments and eagerly await their valuable responses. We are fully committed to revising our manuscript further to ensure its suitability in ICLR. Once again, we extend our gratitude to all the reviewers and the chair for considering our manuscript.

Best regards, Authors.

---

### Meta-Review · Area_Chair_obYj · 2023-12-12

**Metareview:**

The paper provides a nearly linear time algorithm for low rank matrix completion using alternate minimization. Previous algorithms for this problem using alternate minimization were known, but they required exact minimization. The paper shows that this approach is robust to approximate updates. The lack of experiments was noted in the review, but the paper appears theoretically sound, interesting to sections of the ICLR community and the authors have addressed the reviewer concerns during the discussion period.

**Justification For Why Not Higher Score:**

Lukewarm accept recommendations from reviewers.

**Justification For Why Not Lower Score:**

Seems theoretically sound. I don't find lack of experiments a particular problem.

---

### Decision · Program_Chairs · 2024-01-16

Accept (poster)